# Optimal Underdamped Langevin MCMC Method

**Zhengmian Hu, Feihu Huang, Heng Huang**
Department of Electrical and Computer Engineering
University of Pittsburgh, Pittsburgh, PA 15213, USA
{huzhengmian,huangfeihu2018}@gmail.com, heng.huang@pitt.edu

## Abstract

In the paper, we study the underdamped Langevin diffusion (ULD) with strongly-convex potential consisting of finite summation of $N$ smooth components, and propose an efficient discretization method, which requires $O(N + d^{\frac{1}{3}}N^{\frac{2}{3}}/\varepsilon^{\frac{2}{3}})$ gradient evaluations to achieve $\varepsilon$-error (in $\sqrt{\mathbb{E}\|\cdot\|_2^2}$ distance) for approximating $d$-dimensional ULD. Moreover, we prove a lower bound of gradient complexity as $\Omega(N + d^{\frac{1}{3}}N^{\frac{2}{3}}/\varepsilon^{\frac{2}{3}})$, which indicates that our method is optimal in dependence of $N$, $\varepsilon$, and $d$. In particular, we apply our method to sample the strongly-log-concave distribution and obtain gradient complexity better than all existing gradient based sampling algorithms. Experimental results on both synthetic and real-world data show that our new method consistently outperforms the existing ULD approaches.

## 1 Introduction

Sampling is an important research problem in statistics learning with many applications such as Bayesian inference [1], multi-arm bandit optimization [2], and reinforcement learning [3]. One of the fundamental problems in these applications is to sample from a high-dimensional strongly-log-concave distribution. Recently, several Markov chain Monte Carlo (MCMC) based methods were proposed to solve this problem based on underdamped Langevin diffusion (ULD). This continuous diffusion process converges to the target distribution exponentially fast. Thus, the methods approximating a ULD process could be used to sample from the target distribution within certain accuracy.

Multiple discretization methods have been proposed for approximating ULD. Among them, the Euler-Maruyama discretization [4] is the simplest one but generates the largest error. Recently the left point method (LPM) [1] [5] was introduced to fix the gradient term in ULD to be the gradient at $k$-th iteration, and then integrate the new linear stochastic differential equation (SDE) with a small time-interval. Subsequently, Shen and Lee [6] proposed randomized midpoint method (RMM) with smaller error. There are also discretization schemes based on splitting [7] or Runge-Kutta method [8, 9]. More recently, Cao et al. [10] derived an information-based complexity lower bound for simulating a $d$-dimensional ULD. Under the assumption that the full gradient oracle $\nabla f(\boldsymbol{x})$ is evaluated at most $n$ times, they show a lower bound for worst-case error by perturbation analysis, which matches the discretization error upper bound of RMM in the dependence of $d$ and $n$.

Although the ULD-MCMC methods with full gradient oracle are largely understood, many real-world applications involve summation form of potential function and large-scale data, which leads to the need of stochastic gradient methods. The vanilla stochastic gradient methods have been used to replace full gradient [5]. Albeit the computational cost for each iteration is reduced, the variance of

---

[1] Although this method is mostly just denoted as ULD-MCMC, we adopt the name LPM to distinguish it from other discretization methods. The name comes from the fact that gradient is evaluated at the left point of the time interval.

35th Conference on Neural Information Processing Systems (NeurIPS 2021).

Table 1: Summary of gradient complexity of sampling methods, which is defined as number of gradient evaluation of $\nabla f_i(\boldsymbol{x})$ needed to sample from $m$-strongly-log-concave distributions up to $\varepsilon\sqrt{d/m}$ accuracy in 2-Wasserstein distance where $\varepsilon \leq 1$ is the target accuracy, $d$ is the dimension. ULA, LPM, RMM, and ALUM are full gradient methods, therefore, gradient complexities for them are sample size $N$ times the iteration complexities. Only dependence on $d$, $\varepsilon$, and $N$ are shown below. The dependence of batch size $b$ is made clear in Table 3. The dependence of condition number $\kappa$ is discussed in Section 7.

| Algorithms | Gradient complexities |
|---|---|
| Unadjusted Langevin Algorithm (ULA) [15, 16] | $\widetilde{O}(N\varepsilon^{-2})$ |
| LPM [17] | $\widetilde{O}(N\varepsilon^{-1})$ |
| RMM [6] | $\widetilde{O}(N\varepsilon^{-\frac{2}{3}})$ |
| ALUM (Ours) | $\widetilde{O}(N\varepsilon^{-\frac{2}{3}})$ |
| Stochastic Gradient LPM (SG-LPM) [5] | $\widetilde{O}(\varepsilon^{-2})$ |
| SVRG-LPM [11][2] | $\widetilde{O}(N + \varepsilon^{-1} + N^{\frac{2}{3}}\varepsilon^{-\frac{2}{3}})$ |
| CV-ULD [18][3] | $\widetilde{O}(N + \varepsilon^{-3})$ |
| SVRG-ALUM (Ours) | $\widetilde{O}(N + N^{\frac{2}{3}}\varepsilon^{-\frac{2}{3}})$ |
| SAGA-ALUM (Ours) | $\widetilde{O}(N + N^{\frac{2}{3}}\varepsilon^{-\frac{2}{3}})$ |

stochastic gradient is much larger than the discretization error and therefore degenerates the overall performance. Previous works [11, 12] used stochastic variance reduced gradient (SVRG) [13] and SAGA [14] instead, but the gradient complexities of these methods are still worse than the full gradient RMM in terms of dependence on accuracy $\varepsilon$. Thus, there exists a natural question:

> *What is the optimal ULD-MCMC method with sum-decomposable potential?*

In this paper, we focus on optimal dependence of dimension $d$, components number $N$ and accuracy $\varepsilon$ in gradient complexity for estimating a ULD process. We answer this question by two parts. We first provide a novel ULD-MCMC method and derive the corresponding complexity upper bound in Sections 4 and 5. After that, we analyze the worse case error and show that the lower bound matches the upper bound in Section 6. The major contributions of our paper can be summarized as follows.

1. We propose a new full gradient ULD-MCMC method, called as AcceLerated ULD method (ALUM), whose discretization error has the same order dependence on dimension $d$, step size $h$ as RMM. Although RMM already has optimal asymptotic complexity in full gradient setting, ALUM is still of practical interest. Compared with RMM, which uses two gradient evaluations at each iteration, ALUM uses gradient less frequently and only requires one gradient at each iteration to achieve constant speedup.

2. We further propose VR-ALUM methods, including SVRG-ALUM and SAGA-ALUM, which utilize the unbiased variance reduction techniques in ALUM under sum-decomposable setting. We show that these methods achieve better gradient complexity than all existing gradient based MCMC approaches. These gradient complexities for sampling from a strongly-log-concave distribution are compared in Table 1.

3. We derive an information-based lower bound on worst-case error for estimating a ULD process with only gradient oracle and weighted Brownian motion oracle. We show that in order to achieve $\varepsilon$ approximation accuracy, $\Omega(N + d^{\frac{1}{3}}N^{\frac{2}{3}}\varepsilon^{-\frac{2}{3}})$ single component gradient evaluations are needed. This lower bound matches the upper bound for VR-ALUM in terms of dependence of dimension $d$, sample size $N$, and accuracy $\varepsilon$. Therefore, our VR-ALUM methods are indeed optimal for estimating a ULD process under sum-decomposable setting.

---

[2]In Zou et al. [11], the authors call their method SVR-HMC. However, their method is not based on Hamiltonian Monte Carlo (HMC), but based on ULD. Their method is just applying SVRG to replace full gradient in LPM.

[3]Further explanation and comparison is shown in Appendix A.2.

## 2 Related work

Though we mainly use the gradient oracle in this paper, there is also a large body of sampling algorithms which leverage zeroth order potential value oracle. For example, Metropolis-Hastings accept-reject step could be used to ensure an MCMC converges to a stationary distribution equal to the target distribution [19, 20, 21, 22, 23] and the linear convergence can be obtained. For example, MALA [24, 21, 25], as Metropolis-adjusted ULA, can achieve $\varepsilon$ error in total variation distance within $O(d^2 \log(1/\varepsilon))$ steps under certain initialization [26]. Note that purely gradient based sampling algorithms typically only converge sub-linearly because the stationary distribution is different from the target distribution.

Other diffusion processes could also be used for constructing MCMC. ULA, a discretization of Langevin diffusion (LD) [27, 24], is the first commonly used gradient-based MCMC. We will provide an overview for some LD-based and ULD-based methods in Appendix A.1. Later, higher order diffusion process is also discussed. Although the path is even smoother than ULD, currently it is not clear whether such higher order smoothness of path could be leveraged to accelerate the convergence without extra assumptions on potential function. Mou et al. [28] studied third-order diffusion, but their acceleration requires special structure or high order smoothness of potential. The more general diffusion process, which converges to the target distribution, was studied in Ma et al. [29].

## 3 Preliminary

The problem of sampling from a strongly-log-concave distribution involves a probability density function $p^*(\boldsymbol{x})$ defined on a real vector space $\mathbb{R}^d$. A corresponding potential function $f(\boldsymbol{x}) = -\log(p^*(\boldsymbol{x}))$ can be defined such that $f(\boldsymbol{x})$ is strongly convex. One way to solve this sampling problem is constructing a Markov chain that converges to a stationary distribution that is the same as or similar to the target distribution. In Appendix A.1, we introduce several such Markov chains as discretization of certain continuous stochastic processes, and we roughly analyze discretization error of these methods with pointing out the bottleneck.

We use $\widetilde{O}(f) = O(f) \log^{O(1)}(f)$ to omit logarithm factor. $\|\cdot\|_2$ means the Euclidean norm. We define a norm of random vector as $\|\cdot\|_{\mathbb{L}_2} = \sqrt{\mathbb{E}\|\cdot\|_2^2}$. Next, we list our assumptions on potential $f(\boldsymbol{x})$.

**Assumption 1** (Sum-decomposable). $f(\boldsymbol{x}) = \sum_{i=1}^N f_i(\boldsymbol{x})$, where integer $N$ is the sample size.

**Assumption 2** (Smoothness). *Each function $f_i$ is twice differentiable on $\mathbb{R}^d$ and there exists a constant $L > 0$, such that $\boldsymbol{\nabla}^2 f_i(\boldsymbol{x}) \preccurlyeq \frac{L}{N} I$ for any $\boldsymbol{x} \in \mathbb{R}^d$ where $I$ is the identity matrix. It can be easily verified that $f(\boldsymbol{x})$ is $L$-smooth.*

**Assumption 3** (Strong Convexity). *There exists a constant $m > 0$ such that: $f(\boldsymbol{x}) - f(\boldsymbol{y}) \geq \langle \nabla f(\boldsymbol{y}), \boldsymbol{x} - \boldsymbol{y} \rangle + \frac{m}{2} \|\boldsymbol{x} - \boldsymbol{y}\|_2^2$. We define the condition number $\kappa := L/m$.*

We finally define the 2-Wasserstein distance between distributions. For any pair of probability measures $\mu$ and $\nu$ on the same parameter space, a transference plan $\zeta$ between $\mu$ and $\nu$ is a joint distribution such that the marginal distributions on two sets of coordinates are $\mu$ and $\nu$, respectively. We denote $\Gamma(\mu, \nu)$ as the set of all transference plans, and define the 2-Wasserstein distance between $\mu$ and $\nu$ as follows:

$$W_2^2(\mu, \nu) = \inf_{\zeta \in \Gamma(\mu, \nu)} \int \|\boldsymbol{x} - \boldsymbol{y}\|_2^2 d\zeta(x, y).$$

## 4 AcceLerated ULD-MCMC (ALUM) methods

In this section, we propose a class of AcceLerated ULD-MCMC (ALUM) methods based on approximation of the continuous ULD process:

$$d\boldsymbol{X}_t = \boldsymbol{V}_t dt, \ d\boldsymbol{V}_t = -\nabla f(\boldsymbol{X}_t)dt - \gamma \boldsymbol{V}_t dt + \sqrt{2\gamma} d\boldsymbol{B}_t. \tag{1}$$

The gradient term in the above SDE could be highly non-linear such that closed form solution is not available. Thus, we propose an estimation of SDE solution at time point $h$ which only requires

gradient evaluation at one single point:

$$\boldsymbol{X}_h^{(o)} = \boldsymbol{X}_0 + \psi_1(h)\boldsymbol{V}_0 - h\psi_1(h - ah)\nabla f(\boldsymbol{X}_{ah}^{(e)}) + \boldsymbol{e}_{x,[0,h]},$$
$$\boldsymbol{V}_h^{(o)} = \psi_0(h)\boldsymbol{V}_0 - h\psi_0(h - ah)\nabla f(\boldsymbol{X}_{ah}^{(e)}) + \boldsymbol{e}_{v,[0,h]}, \tag{2}$$
$$\boldsymbol{X}_{ah}^{(e)} = \boldsymbol{X}_0 + \psi_1(ah)\boldsymbol{V}_0 + \boldsymbol{e}_{x,[0,ah]},$$

where the random variable $a$ is uniform random variable in $[0,1]$, $\psi_i(\cdot)$ and noise terms $\boldsymbol{e}_{x,[0,h]}$, $\boldsymbol{e}_{v,[0,h]}$, $\boldsymbol{e}_{x,[0,ah]}$ are defined as follows [4].

$$\psi_0(h) = e^{-\gamma h}, \psi_1(h) = \frac{1}{\gamma}(1 - e^{-\gamma h}),$$
$$\boldsymbol{e}_{x,[0,h]} = \sqrt{2\gamma} \int_0^h \psi_1(h - s)d\boldsymbol{B}_s, \quad \boldsymbol{e}_{v,[0,h]} = \sqrt{2\gamma} \int_0^h \psi_0(h - s)d\boldsymbol{B}_s. \tag{3}$$

## 4.1 Full gradient ALUM

In this subsection, we propose a full gradient ALUM method, which is shown in Algorithm 1. Specifically, the update rule is obtained by setting $\boldsymbol{x}_{k+1}^{(o)}, \boldsymbol{v}_{k+1}^{(o)}$ to $\boldsymbol{X}_h^{(o)}, \boldsymbol{V}_h^{(o)}$ where the ULD starts from $\boldsymbol{X}_0 = \boldsymbol{x}_k^{(o)}, \boldsymbol{V}_0 = \boldsymbol{v}_k^{(o)}$ and $h$ is a small step size.

Next, we compare our method with both RMM and Nesterov's accelerated gradient method (NAG) which has been shown optimal for gradient based optimization. **Compared to RMM:** By comparing the formula in (19) and (2), it is easy to see that ALUM and RMM differ in one single term $-\psi_2(ah)\nabla f(\boldsymbol{X}_0)$.

Our motivation of dropping this term is to reduce the computations. By deleting this term from the algorithm, only one gradient evaluation at a randomized midpoint is needed at each iteration. This compares favorably to the RMM, which requires two gradient evaluations at each iteration.

---

**Algorithm 1:** Full gradient ALUM Method

---

**Input:** Initial point $(\boldsymbol{x}_0^{(o)}, \boldsymbol{v}_0^{(o)})$, parameter $\gamma$, iteration number $K$, and step size $h > 0$.

**for** $k = 0$ **to** $K - 1$ **do**
    Randomly sample $a_k$ uniformly from $[0,1]$;
    Generate $\boldsymbol{e}_{x,[0,h],k}, \boldsymbol{e}_{v,[0,h],k}, \boldsymbol{e}_{x,[0,a_kh],k}$ according to Appendix A.5;
    $\boldsymbol{x}_k^{(e)} = \boldsymbol{x}_k^{(o)} + \psi_1(a_kh)\boldsymbol{v}_k^{(o)} + \boldsymbol{e}_{x,[0,a_kh],k}$;
    Calculate full gradient $\nabla f(\boldsymbol{x}_k^{(e)})$;
    $\boldsymbol{x}_{k+1}^{(o)} =$
    $\boldsymbol{x}_k^{(o)} + \psi_1(h)\boldsymbol{v}_k^{(o)} - h\psi_1(h - a_kh)\nabla f(\boldsymbol{x}_k^{(e)}) + \boldsymbol{e}_{x,[0,h],k}$;
    $\boldsymbol{v}_{k+1}^{(o)} = \psi_0(h)\boldsymbol{v}_k^{(o)} - h\psi_0(h - a_kh)\nabla f(\boldsymbol{x}_k^{(e)}) + \boldsymbol{e}_{v,[0,h],k}$;
**end for**
**Output:** $\boldsymbol{x}_K^{(o)}$.

---

**However, could this method still estimate ULD accurately with only half gradient evaluations?** We firmly answer this question with rigorous error analysis in Theorem 2. Roughly speaking, if we only consider the error's dependence on step size $h$, the answer is yes. In this section, we only give an intuition of how that is possible.

In Appendix A.1, we show that RMM has very low bias and high variance. Therefore, the bottleneck is the variance, and increasing the bias slightly would not degenerate the overall performance much if the bias is not larger than the error introduced by variance. The dropped term $-\psi_2(ah)\nabla f(\boldsymbol{X}_0)$ has norm $O(h^2)$, and the coefficient of gradient in $\boldsymbol{V}_h$ is $O(h)$. Thus, the bias introduced in single step is $O(h^3)$ when we use step size $h$. After accumulating for $T/h = O(h^{-1})$ iterations, the bias is $O(h^2)$, which still has better dependence on $h$ than the square root of variance $O(h^{3/2})$.

With increased bias, the complexities of ALUM and some other variants[5] still have the same dependence on $\varepsilon$ as RMM, the highest order dependence on $\kappa$ could deteriorate as discussed in Section 7.

**Compared to Nesterov's accelerated gradient (NAG) method:** NAG is a gradient based optimization method [30]. Based on [31], NAG could be formulated as the following momentum method:

---

[4]$\boldsymbol{e}_{x,[0,ah]}$ is obtained by simply substituting $h$ with $ah$ in the definition of $\boldsymbol{e}_{x,[0,h]}$.

[5]We can drop $\boldsymbol{e}_{x,[0,h],k}$ and $\boldsymbol{e}_{x,[0,a_kh],k}$ in (2) to derive other variants of ALUM. The bias of these variants increases to $O(h^{3/2})$, which is still no larger than the square root of variance. The maximum order dependence on $h$ is the same, thus the final iteration complexity has the same dependence on $\varepsilon$ as RMM.

$$\boldsymbol{x}_{k+1} = \boldsymbol{x}_k + c_k \boldsymbol{x}_k + c'_k \nabla f(\boldsymbol{x}_k^{(e)}), \quad \boldsymbol{v}_{k+1} = c_k \boldsymbol{x}_k + c'_k \nabla f(\boldsymbol{x}_k^{(e)}), \quad \boldsymbol{x}_k^{(e)} = \boldsymbol{x}_k + c_k \boldsymbol{x}_k. \quad (4)$$

We can see that the main differences between Algorithm 1 and (4) are coefficients and additional Gaussian noise terms. The coefficients in NAG are set in a deterministic way. However, the coefficients in ALUM could be random values from fixed distributions. The additional noise terms in ALUM come from the Brownian motion term in ULD and are necessary for sampling methods.

Despite the differences, both NAG and ALUM use one gradient at each iteration, and both of them take a big jump along current momentum direction to calculate the gradient instead of directly computing the gradient at current iterate.

### 4.2  Variance-reduced ALUM (VR-ALUM)

In the subsection, we propose variance-reduced stochastic ALUM (VR-ALUM) methods based on two common unbiased [6] variance-reduced techniques: stochastic variance reduced gradient (SVRG) [13] and SAGA [14]. VR-ALUM comes from simply replacing all full gradient in Algorithm 1 with a gradient estimation $\widetilde{\nabla}_k$. We show Algorithm 2 in the Appendix A.6 due to the limit of space.

Next, we briefly introduce these two variance reduction techniques. SVRG utilizes the following gradient estimation ($B_k$ is the batch of $k$-th iteration and $b$ is the batch size): $\widehat{\nabla}_k^{\text{SVRG}} = \frac{N}{b} \sum_{i \in B_k} \left( \nabla f_i(\boldsymbol{x}_k^{(e)\widetilde{\nabla}}) - \nabla f_i(\overline{\boldsymbol{x}}) \right) + \sum_{i=1}^{N} \nabla f_i(\overline{\boldsymbol{x}})$. The full gradient $\sum_{i=1}^{N} \nabla f_i(\overline{\boldsymbol{x}})$ and point $\overline{\boldsymbol{x}}$ are updated after every $\tau$ evaluations of $\widetilde{\nabla}_k^{\text{SVRG}}$. We call the hyperparameter $\tau$ as epoch length.

SAGA estimates the gradient in the following way, where $\boldsymbol{\phi}_{k+1}^i$ is set as $\boldsymbol{x}_k^{(e)\widetilde{\nabla}}$ if and only if $i \in B_k$, otherwise $\boldsymbol{\phi}_{k+1}^i = \boldsymbol{\phi}_k^i$: $\widetilde{\nabla}_k^{\text{SAGA}} = \frac{N}{b} \sum_{i \in B_k} \left( \nabla f_i(\boldsymbol{x}_k^{(e)\widetilde{\nabla}}) - \nabla f_i(\boldsymbol{\phi}_k^i) \right) + \sum_{i=1}^{N} \nabla f_i(\boldsymbol{\phi}_k^i)$. SAGA does not re-compute but stores the latest gradient information $\nabla f_i(\boldsymbol{\phi}_k^i)$ for each $f_i$. Therefore, SAGA does not introduce extra gradient evaluation except for initialization, but has much larger storage requirements.

Both SVRG and SAGA are unbiased, which means $\mathbb{E}_{B_k} \widetilde{\nabla}_k = \nabla f(\boldsymbol{x}_k^{(e)\widetilde{\nabla}})$, where $\mathbb{E}_{B_k}$ means expectation over random batch at $k$-th iteration. Moreover, both SVRG and SAGA reduce the mean-squared error of gradient estimation and satisfy bounded MSE property proposed in Appendix B.3.

## 5  Theoretical analysis

We provide non-asymptotic upper bounds on sampling error and discretization error for our methods, including full gradient ALUM and VR-ALUMs. The proof is shown in Appendix B. Throughout this section, we assume $W_2(p_0, p^*) = O(1)\sqrt{d/m}$ when deriving asymptotic results. This means the initialization is not too far away from the target distribution, and can be achieved under multiple setting as discussed in Appendix A.7.

### 5.1  Convergence analysis of full gradient ALUM

Recall that ALUM can be used for solving two different but related problems: strongly-log-concave sampling and approximating the ULD. We show the upper bound for sampling error in Theorem 1 and the upper bound for approximation error in Theorem 2 separately.

**Theorem 1.** *Suppose Assumptions 1 to 3 hold. Given an initial distribution $p_0(\boldsymbol{x})$, we initialize ALUM with random $\boldsymbol{x}_0^{(o)}$ based on probability $p_0$ and random $\boldsymbol{v}_0^{(o)}$ from standard Gaussian distribution. Assume $L = 1$ and let $\gamma = 2$, $p_k$ be the distribution of $\boldsymbol{x}_k^{(o)}$ and $p^*$ be the target distribution. Assume we use step size $h \leq \frac{m}{22}$. After running the ALUM for $k$ iterations, we have the following upper bound of sampling error in 2-Wasserstein distance:*

$$W_2(p_k, p^*) \leq 2(1 - \frac{mh}{4})^k W_2(p_0, p^*) + 12\sqrt{\frac{h^3 d}{m}}. \quad (5)$$

---

[6] The reason we choose unbiased variance reduction instead of biased one is that bias accumulates quicker than variance, therefore generates higher overall error.

Table 2: Iteration complexities for full gradient ALUM on both sampling and approximation problem.

| Problem | Accuracy | Step size $h$ | Iteration complexity |
|---|---|---|---|
| Sampling | $\varepsilon\sqrt{d/m}$ in $W_2$ | $h = O(\min(\varepsilon^{\frac{2}{3}}, m))$ | $\widetilde{O}(\max(\kappa/\varepsilon^{\frac{2}{3}}, \kappa^2))$ |
| Approximating | $\varepsilon$ in $\mathbb{L}_2$ | $h = O(\min(m^{\frac{2}{3}}\varepsilon^{\frac{2}{3}}d^{-\frac{1}{3}}, m))$ | $O(\max(T\kappa^{\frac{2}{3}}\varepsilon^{-\frac{2}{3}}d^{\frac{1}{3}}, T\kappa))$ |

**Theorem 2.** *With same assumptions and setup in Theorem 1, we consider $\boldsymbol{X}_{kh}$ which comes from a continuous ULD starting from the same position as $\boldsymbol{x}_0^{(o)}$ with probability 1. We have the following upper bound of discretization error:*

$$\|\boldsymbol{x}_k^{(o)} - \boldsymbol{X}_{kh}\|_{\mathbb{L}_2} \le 38\sqrt{\frac{h^3}{m}}W_2(p_0, p^*) + 31\sqrt{\frac{h^3 d}{m}}. \tag{6}$$

**Remark 1.** *The assumption $L = 1$ actually does not limit the availability of the algorithm. For any function $f(\boldsymbol{x})$ with $L \ne 1$, we can define another function $f'(\boldsymbol{x}') = f(\frac{1}{\sqrt{L}}\boldsymbol{x}')$ which satisfies $L = 1$. ALUM with step size $h \le \frac{m}{22L}$ and initial momentum from the standard Gaussian distribution can be used to sample $\boldsymbol{x}'$ from the distribution $p'(\boldsymbol{x}') \propto \exp(-f'(\boldsymbol{x}'))$. The sample $\boldsymbol{x}$ from distribution $p(\boldsymbol{x}) \propto \exp(-f(\boldsymbol{x}))$ could be obtained by a transform $\boldsymbol{x} = \frac{1}{\sqrt{L}}\boldsymbol{x}'$. This is essentially the same as directly incorporating $L$ into ULD process as in Appendix A.3.*

Thus, we can derive the iteration complexity for both problems. For sampling problem, we define the iteration complexity as the number of iterations $K$ needed to achieve $W_2(p_K, p^*) \le \varepsilon\sqrt{d/m}$ with certain step size $h$. For approximating the ULD at a given time point $T$, we define the iteration complexity as the number of iterations $K$ needed to achieve $\|\boldsymbol{x}_K^{(o)} - \boldsymbol{X}_T\|_{\mathbb{L}_2} \le \varepsilon$ with step size $h = T/K$. The results are shown in Table 2. Detailed derivations can be found at Appendix A.8.

## 5.2 Convergence analysis of variance-reduced ALUM (VR-ALUM)

We show the upper bound for sampling error in Theorem 3 and the upper bound for discretization error in Theorem 4 separately.

**Theorem 3.** *With same assumptions in Theorem 1, we use SVRG-ALUM with epoch length $\tau = \lceil N/b \rceil$ or SAGA-ALUM. We introduce an extra assumption $h^3 \le \frac{1}{2304c}b^3 m N^{-2}$. We have the following upper bound of sampling error in 2-Wasserstein distance.*

$$W_2(p_k, p^*) \le 2(1 - \frac{mh}{4})^k W_2(p_0, p^*) + 92\sqrt{\frac{h^3}{m}}\sqrt{c}\frac{N}{b^{\frac{3}{2}}}W_2(p_0, p^*) + (12 + 57\sqrt{c}\frac{N}{b^{\frac{3}{2}}}))\sqrt{\frac{h^3 d}{m}}. \tag{7}$$

*The constant $c$ is defined as $c = 1$ for SVRG-ALUM and $c = 2$ for SAGA-ALUM.*

**Theorem 4.** *With the same assumptions in Theorem 3, we consider $\boldsymbol{X}_{kh}$ which comes from a continuous ULD starting from the same position as $\boldsymbol{x}_0^{(o)\widetilde{\nabla}}$ with probability 1. We have the following upper bound of discretization error.*

$$\|\boldsymbol{x}_k^{(o)\widetilde{\nabla}} - \boldsymbol{X}_{kh}\|_{\mathbb{L}_2} \le (38 + 92\sqrt{c}\frac{N}{b^{\frac{3}{2}}})\sqrt{\frac{h^3}{m}}W_2(p_0, p^*) + (31 + 57\sqrt{c}\frac{N}{b^{\frac{3}{2}}})\sqrt{\frac{h^3 d}{m}}. \tag{8}$$

**Remark 2.** *Theorems 1 to 4 are specializations of more general results in Appendix B.1, where full gradient, SVRG and SAGA are unified under the framework of bounded MSE property that is defined in Appendix B.3. A unified approach not only simplifies the proof, but also indicates that our analysis could easily generalize to other gradient estimations that satisfy bounded MSE property.*

Similar to the full gradient case, we derive the iteration complexity $K$ in Appendix A.9. Moreover, we define gradient complexity as the number of single component gradient evaluation $\nabla f_i(\boldsymbol{x})$ needed to achieve certain accuracy. We show the results in Table 3 and add the derivations in Appendix A.9. We finally simplify the result by only considering the dependence of $d$, $N$, $b$, and $\varepsilon$.

**Corollary 1.** *When $b \le O(N^{\frac{2}{3}})$, the gradient complexity of SAGA-ALUM and SVRG-ALUM for sampling problem is $\widetilde{O}(N + N^{\frac{2}{3}}\varepsilon^{-\frac{2}{3}})$ and their gradient complexity for ULD approximation problem is $O(N + d^{\frac{1}{3}}N^{\frac{2}{3}}\varepsilon^{-\frac{2}{3}})$.*

Table 3: Gradient complexity for SAGA-ALUM and SVRG-ALUM.

| Problem | Accuracy | Gradient complexity |
|---|---|---|
| Sampling | $\varepsilon\sqrt{d/m}$ in $W_2$ | $\widetilde{O}(N + (b\kappa + N^{\frac{2}{3}}\kappa^{\frac{4}{3}})(1 + \varepsilon^{-\frac{2}{3}}) + b\kappa^2$ |
| Approximating | $\varepsilon$ in $\mathbb{L}_2$ | $O(N + T(\kappa b + \kappa^{\frac{1}{3}}N^{\frac{2}{3}}) + T\kappa^{\frac{2}{3}}d^{\frac{1}{3}}\varepsilon^{-\frac{2}{3}}(b + N^{\frac{2}{3}}))$ |

# 6 Information-based complexity

We first declare our setup for the problem class and accessible information. Then we show information-based lower bound for approximation error and oracle complexity. The proof is shown in Appendix C.

## 6.1 Setup

**Class of possible potential functions:** We denote the class of possible potential functions as $\mathcal{U} = \mathcal{U}(d, N, m, L) = \{(f_1, \ldots, f_N) | \boldsymbol{\nabla}^2 f_i(\boldsymbol{x}) \preccurlyeq (L/N)I, mI \preccurlyeq \boldsymbol{\nabla}^2 \sum_{i=1}^N f_i(\boldsymbol{x})$ and $\|\mathbb{E}_{p^*}[\boldsymbol{X}]\|_2 \leq \sqrt{d/m}\}$ where $0 < m < L$ and $p^*(\boldsymbol{X}) \propto \exp(-\sum_{i=1}^N f_i(\boldsymbol{X}))$.

Besides Assumptions 1 to 3 used in the previous sections, we introduce a new assumption that the target distributions have mean in a ball of constant radius around the origin. This is necessary because the lower bound could be arbitrarily large if the mean is far away from the initialization point.

**Solution mapping:** We denote the probability space for Brownian motion in the ULD process as $(\mathbb{M}, \Sigma, \mathbb{P})$. Then, the true solution of ULD process starting from the origin $\boldsymbol{0}$ at time $T$ can be denoted as the solution mapping $\boldsymbol{X}_T : (\omega, U) \in (\mathbb{M} \times \mathcal{U}) \mapsto \boldsymbol{X}_T(\omega, U) \in \mathbb{R}^d$.

**Gradient oracle:** For a given set of potential functions $U = (f_1, \ldots, f_N)$, the single component gradient oracle is $\Upsilon_U : (i, \boldsymbol{x}) \in [N] \times \mathbb{R}^d \mapsto \nabla f_i(\boldsymbol{x}) \in \mathbb{R}^d$.

**Brownian oracle:** We assume the Brownian motion at a given time $t > 0$ could be evaluated with oracle $\boldsymbol{B}_t(\omega) \in \mathbb{R}^d$ for any event $\omega \in \mathbb{M}$. We further assume that the weighted Brownian motion is also admissible. $\boldsymbol{B}_t^{(\theta)}(\omega) = \int_0^T e^{\theta s} d\boldsymbol{B}_s(\omega)$.

**Deterministic algorithm:** A deterministic algorithm starts from empty information $I_0 = ()$. At $i$-th step, one oracle and corresponding parameters are picked by certain procedure. If the gradient oracle is picked, the algorithm will generate an index $i$ and a point $\boldsymbol{x}$. If the weighted gradient oracle is picked, the parameters are order $\theta$ for weighted Brownian motion and time $t$. The picked oracle and parameters are represented by $\phi_i(I_i)$. The evaluation result is represented by $\Upsilon(\phi(I_i), \omega, U)$ where $\omega \in \mathbb{M}$ and $U \in \mathcal{U}$. The picked oracle, parameters, and result will be stored as new information, hence $I_{i+1} = (I_i, \phi(I_i), \Upsilon(\phi(I_i), \omega, U))$.

We consider deterministic algorithms that stop at $n$-th step. The final estimation is generated with a mapping $Y(I_n) \in \mathbb{R}^d$. We expect $Y(I_n)$ to be as close as possible to true solution $\boldsymbol{X}_T(\omega, U)$.

We denote a deterministic algorithm as a mapping $A$ from $\mathbb{M} \times \mathcal{U}$ to $\mathbb{R}^d$ and $A(\omega, U) = Y(I_n)$ for some $\phi$ and $Y$. The family of all such algorithms is denoted by $\mathcal{A}_n^{\mathrm{Det}}$.

**Randomized algorithm:** We consider another probability space $(\widetilde{\mathbb{M}}, \widetilde{\Sigma}, \widetilde{\mathbb{P}})$ as the source of randomness. A randomized algorithm $A$ with $n$ steps is a mapping from $\mathbb{M} \times \widetilde{\mathbb{M}} \times \mathcal{U}$ to $\mathbb{R}^d$ such that $A(\cdot, \widetilde{\omega}, \cdot) \in \mathcal{A}_n^{\mathrm{Det}}$ for any $\widetilde{\omega} \in \widetilde{\mathbb{M}}$. The family of all such randomized algorithms is denoted by $\mathcal{A}_n$.

**Worst error of algorithms:** We care about the following worst-case error for any possible algorithms:

$$e_{\mathcal{A}, \mathcal{U}}^2 := \inf_{A \in \mathcal{A}} \sup_{U \in \mathcal{U}} \mathbb{E}_{\omega \in \mathbb{P}} \mathbb{E}_{\widetilde{\omega} \in \widetilde{\mathbb{P}}} \|\boldsymbol{X}_T(\omega, U) - A(\omega, \widetilde{\omega}, U)\|_2^2 \tag{9}$$

We always assume $T > 0$ to avoid trivial cases.

## 6.2 Lower bounds

We first provide lower bounds on worst-case estimation error.

**Theorem 5.** *When $n < N$ which means that gradient evaluation number is less than components number, we have $e^2_{\mathcal{A}_n,\mathcal{U}} \geq dC_1$, where $C_1$ is positive and independent of $d$, $N$, and $n$.*

The above theorem is based on the fact that no algorithm can accurately estimate the global minimum point of the sum of quadratic potentials $\sum_{i=1}^N f_i(\boldsymbol{x})$ with information from only $N - 1$ components.

**Theorem 6.** *When the gradient evaluation number $n$ is a multiple of $N$, we have $e^2_{\mathcal{A}_n,\mathcal{U}} \geq dC_2 \frac{N^2}{n^3}$, where $C_2$ is positive and independent of $d$, $N$, and $n$.*

The method we use to prove the above theorem is to show that an algorithm that uses a limited number of gradient evaluations cannot distinguish a class of perturbed quadratic functions.

Next, we use the above lower bounds on error to derive a lower bound on gradient complexity.

**Corollary 2.** *For small enough target accuracy $\varepsilon$ such that $\varepsilon^2 < dC_1$, in order to achieve $e_{\mathcal{A}_n,\mathcal{U}} \leq \varepsilon$, the minimum number of single component gradient oracle evaluations is $\Omega(N + d^{\frac{1}{3}} N^{\frac{2}{3}} \varepsilon^{-\frac{2}{3}})$.*

This lower bound matches the upper bound in Corollary 1 which indicates that gradient complexity of variance-reduced ALUM for estimating a ULD is optimal in the dependence of $d$, components number $N$, and approximation accuracy $\varepsilon$.

# 7  Optimality

In this section, we discuss in what sense our ALUM is optimal (or not), and point out possible improvements for future work.

**Optimal for approximating problem:** ALUM can be used for two tasks: (1) estimating a ULD process, (2) sampling from a strongly-log-concave distribution. ALUM is only optimal for the first task, and is not necessarily optimal for the second one. For example, we see in Section 5 that there exists a logarithm factor in the gradient complexity for sampling problem. We believe some proper adaptive step size method could cancel that extra term.

**Dependency on $\kappa$:** ALUM is only optimal on the dependence of $d$, $N$, $\varepsilon$, but not optimal in $\kappa$ dependence. Actually both our method and our analysis may not be optimized for the dependence of $\kappa$. Currently, there is no information-based lower bound with clear dependence on condition number $\kappa$, therefore no matter how good the dependence on $\kappa$ is, it is not sufficient to say it is optimal. For sampling problem, it is believed that $O(\min(\kappa, d^2))$ is the "natural barrier" [6] for iterations needed for achieving $W_2 \leq \varepsilon\sqrt{d/m}$ with $\varepsilon = 1/2$. Currently, the best dependence on $\kappa$ is achieved by RMM with $O(\kappa^{7/6})$. For ALUM, the maximum dependence is $O(\kappa^2)$. This dependence is worse for two reasons. First, the analysis is not optimized for $\kappa$. Based on the analysis of bias and variance in Appendix A.10, we conjecture that the dependence could be improved to $O(\kappa^{3/2})$. Second, the method is not optimized for $\kappa$. We save one extra gradient compared to RMM, at the price of a slightly increased bias. This degenerates the dependence of $\kappa$.

Despite the max dependence order being larger than 1, in the high-precision regime, when $\varepsilon$ is small enough, the term $\kappa/\varepsilon^{\frac{2}{3}}$ is dominant, therefore, both full-gradient ALUM and full-gradient RMM achieve $O(\kappa)$ dependence.

We note the complexity for VR-ALUM increases the dependence of $\kappa$ to $\kappa^{4/3}$ in high-precision regime. We conjecture that this is an artifact of error-based analysis compared to momentum-based analysis, and could be improved by some tighter analysis.

**Assumptions:** Finally, we point out that our method and analysis are based on Assumptions 1 to 3 and the gradient oracle. It is natural to obtain better algorithm by introducing new assumptions or new oracle.

Many higher order integrators [8, 9] leverage higher order smoothness assumption to reduce discretization error. Assumptions on structure of potential [28] have also been shown to make acceleration possible together with high order diffusion process. Apart from the gradient oracle, many Metropolis-adjusted algorithms [24, 26] leverage zeroth-order oracle to achieve linear convergence. Second order oracle [17] has also been incorporated into estimating ULD.

# 8 Experiments

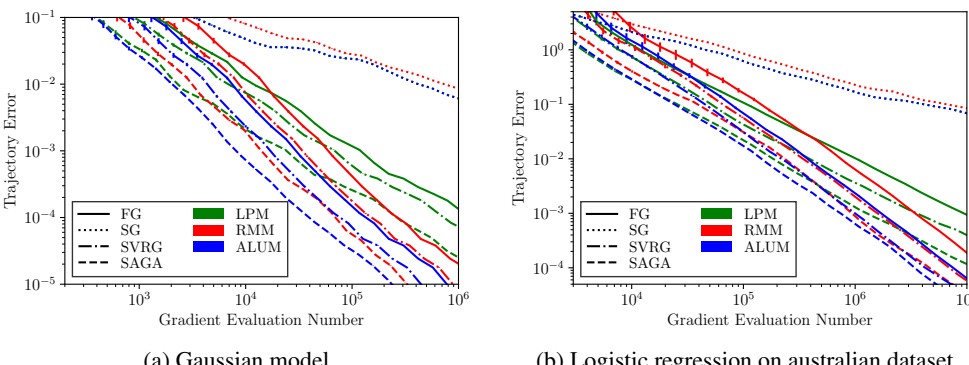

(a) Gaussian model  (b) Logistic regression on australian dataset

Figure 1: Trajectory error for different algorithms[7]. The vertical lines are the error bar. FG means full gradient, SG means stochastic gradient.

In this section, we compare different algorithms for estimating a ULD process on both Gaussian model and Bayesian logistic regression model. For the Gaussian model, the potential is defined as:

$$f_i(\boldsymbol{x}) = \frac{1}{2N}(\boldsymbol{d}_i - \boldsymbol{x})^\top \Sigma^{-1}(\boldsymbol{d}_i - \boldsymbol{x}), \tag{10}$$

where $\boldsymbol{d}_i$ and $\Sigma$ is generated randomly to satisfy $d = 5, N = 100, m = 1$ and $L = 10$. For logistic regression model, the potential is:

$$f_i(\boldsymbol{x}) = \frac{m}{2N}\|\boldsymbol{x}\|_2^2 + \sum_{i=1}^{N} \log(1 + \exp(-y_i\boldsymbol{a}_i^\top \boldsymbol{x})), \tag{11}$$

where $m$ is set such that $\kappa = 10^4$ and $y_i, \boldsymbol{a}_i$ are data points in australian dataset from LIBSVM [32].

We calculate the trajectory error which is defined as $\frac{1}{K}\sum_{i=1}^{K}\sqrt{\|\boldsymbol{x}_k - \boldsymbol{x}_k'\|_2^2 + \|\boldsymbol{v}_k - \boldsymbol{v}_k'\|_2^2}$, where $\boldsymbol{x}_k$ and $\boldsymbol{v}_k$ are generated by ALUM, VR-ALUMs or other algorithms. $\boldsymbol{x}_k'$ and $\boldsymbol{v}_k'$ come from a reference path that is very close to true solution. We specify how we generate this reference path in Appendix D.1. The reason for us to average the error along the path instead of just reporting error at the final iterate is that we need more data points to reduce the variance.

Figure 1 shows the error for ALUM, LPM, and RMM with full gradient, stochastic gradient, SVRG and SAGA.[8] The detailed setup can be found in Appendix D.2. We summarize the messages in Figure 1 as follows:

- SAGA-ALUM achieves the best efficiency in the sense that with same number evaluations of single component gradient $\nabla f_i(\boldsymbol{x})$, SAGA-ALUM has smaller discretization error than any other algorithms. Results on more dataset are shown in Appendix D.3.

- Variance reduced algorithms constantly outperform full gradient algorithm of the same type by a large margin.

- For full gradient method, the discretization error of ALUM and RMM has similar asymptotic dependence on gradient evaluation number, and they are better than LPM. This phenomenon will be shown clearer in Figure 2.

- ALUM achieves constant acceleration compared to RMM by saving one gradient evaluation per iteration.

---

[7]Most of the error bars are too small to be visible. The SG-ALUM and SG-LPM highly overlap.

[8]Currently there is no theoretical result for RMM with stochastic gradient, SVRG or SAGA. We also didn't provide a theoretical result for ALUM with stochastic gradient. However, this doesn't prevent us from evaluating them experimentally.

We also show the relationship between discretization error and step size in Figure 2. Only full gradient method are shown here due to the limit of space, and more results for VR-ALUMs are shown in Appendix D.4.

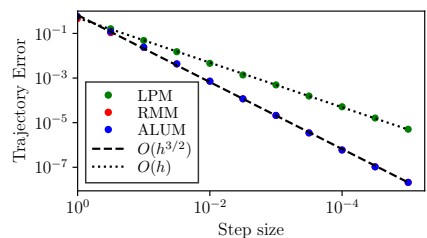

Our analysis in Theorem 2 gives upper bound of discretization error of ALUM as $O(h^{3/2})$. Figure 2 shows that our analysis is tight. Moreover, the discretization error of ALUM is almost the same as RMM. Due to the fact that ALUM requires only half gradient evaluations per step than RMM, ALUM achieves better efficiency.

Figure 2: Discretization error of full gradient methods on australian dataset.[9]

Next, we discuss the effect of batch size. According to our theory in Corollary 1, our method is not sensitive to this hyperparameter introduced by SAGA or SVRG when batch size is relatively small, as the gradient complexity remains the same for $b = O(N^{2/3})$. We verify that in Figure 3, where the sampling efficiency is almost same for small batch sizes, and only deteriorate for very large batch size. We give further discussion on why our method is not sensitive to step size in Appendix D.5.

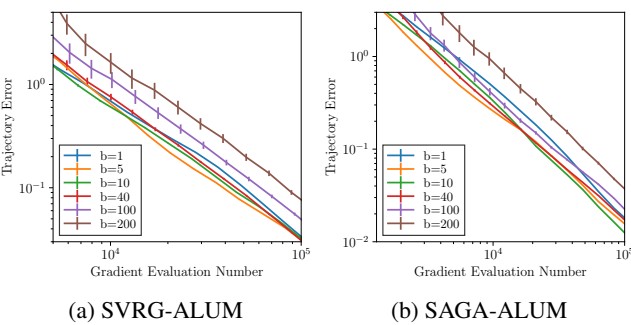

(a) SVRG-ALUM

(b) SAGA-ALUM

Figure 3: Discretization error for SVRG-ALUM and SAGA-ALUM with different batch sizes on australian dataset.

Finally, we apply ALUM and VR-ALUMs to sample from a target distribution in Appendix D.6.

## 9 Conclusion

In this paper, we propose a class of MCMC methods for finite sum form of strongly-convex potential. Our methods are proven to be optimal in the sense that the discretization error has the best possible asymptotic dependence on dimension, number of potential summands, and number of gradient evaluations. Experiments on both synthetic and real data verify the superior performance of our algorithm. We also discuss possible improvements for future work.

## Acknowledgments and Disclosure of Funding

This work was partially supported by NSF IIS 1845666, 1852606, 1838627, 1837956, 1956002, OIA 2040588.

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
