The simplest and most widely used process is the Langevin diffusion (LD) defined as following SDE:

$$dX_t = -\nabla f(X_t)dt + \sqrt{2}dB_t, \tag{12}$$

where $B_t$ is the standard Brownian motion. It is known that distribution of $X_t$ converges to the exact target distribution exponentially [24]. The unadjusted Langevin algorithm (ULA) comes from applying Euler-Maruyama discretization [4] directly to LD. The update rule is:

$$x_{k+1} = x_k + \nabla f(x_k)h + \sqrt{2}e_k, \; e_k = \int_0^t dB_t \sim \mathcal{N}(0, hI), \tag{13}$$

where $e_k$ is standard Gaussian vector, and $h$ is the step size. Compared with the ground-truth update $X_h = x_k - \int_0^h \nabla f(X_t)dt + \sqrt{2}\int_0^t dB_t$, where $X_t$ is an LD starting from the $x_k$, the discretization introduces an error $\sqrt{\mathbb{E}\|x_{k+1} - X_h\|_2^2} = O(h^{\frac{3}{2}})$ in each step. After accumulating this error in $\frac{T}{h}$ iterations, the final estimation error is $O(h^{\frac{1}{2}})$. Therefore, in order to sample from the distribution at accuracy $\varepsilon$, we need to select step size $h = O(\varepsilon^2)$, sampling time $T = O(\log(\varepsilon^{-1}))$ and the final iteration complexity is $T/h = \widetilde{O}(\varepsilon^{-2})$.

The analysis of ULA shows that the discretization error directly affects the sampling error, thus determines the sampling efficiency [15, 16]. More accurate approximation of the continuous diffusion process is desired.

The trajectory for LD is highly non-smooth, therefore is hard to approximate. In order to solve this problem, underdamped Langevin diffusion (ULD) [33] was proposed with using the following SDE:

$$dX_t = V_t dt, \; dV_t = -\nabla f(X_t)dt - \gamma V_t dt + \sqrt{2\gamma}dB_t. \tag{14}$$

We note that in previous works [5, 17, 34, 6], ULD shows up in multiple different forms which are essentially same up to a linear transformation. More detailed discussions can be found in Appendix A.3.

The ULD converges to extended distribution $p^*(x, v) \propto \exp(-f(x) - \frac{1}{2}\|v\|_2^2)$. The noise is injected into an additional momentum term $V_t$ and then propagates to $X_t$, therefore the trajectory of $X_t$ is much smoother and easier to approximate.

Cheng et at. [5] introduced the LPM discretization scheme to leverage this smoothness. In order to illustrate the method, we first give the integral form of ULD as follows. The detailed derivation is shown in Appendix A.4.

$$X_t = X_0 + \psi_1(s)V_0 - \int_0^t \psi_1(t-s)\nabla f(X_s)ds + e_{x,[0,t]}, e$$

$$V_t = \psi_0(t)V_0 - \int_0^t \psi_0(t-s)\nabla f(X_s)ds + e_{v,[0,t]}. \tag{15}$$

$$e_{x,[0,t]} = \sqrt{2\gamma}\int_0^t \psi_1(t-s)dB_s, \; e_{v,[0,t]} = \sqrt{2\gamma}\int_0^t \psi_0(t-s)dB_s. \tag{16}$$

The auxiliary function $\psi_i(x)$ is defined as:

$$\psi_0(t) = e^{-\gamma t}, \psi_i(t) = \int_0^t \psi_{i-1}(u)du, \forall i \geq 1. \tag{17}$$

The basic idea of LPM is to estimate $\nabla f(X_s)$ in (15) with gradient at initial point $\nabla f(X_0)$. This gives following approximation at time point $h$.

$$X_h^{(l)} = X_0 + \psi_1(h)V_0 - \psi_2(h)\nabla f(X_0) + e_{x,[0,h]}, \quad V_h^{(l)} = \psi_0(h)V_0 - \psi_1(h)\nabla f(X_0) + e_{v,[0,h]}. \tag{18}$$

This gives $O(h^2)$ error in single step which is lower than ULA. After accumulation of $T/h$ iterations, the discretization error is $O(h)$. With similar argument like ULA, the iteration complexity is $\widetilde{O}(\varepsilon^{-1})$.

Randomized midpoint method (RMM) [6], which enjoys best iteration complexity among all existing full gradient based sampling algorithms, tries to estimate the integration $\int_0^t \psi_i(t-s)\nabla f(\boldsymbol{X}_s)ds$ with a randomized midpoint $t\psi_i(t-at)\nabla f(\boldsymbol{X}_{at})$ where $a$ is a uniform random variable in $[0,1]$. This is actually an unbiased estimation itself. However, $\nabla f(\boldsymbol{X}_{at})$ still cannot be calculated explicitly. Therefore, RMM uses LPM to calculate $\nabla f(\boldsymbol{X}_{at}^{(l)})$ as a proxy, which introduces bias into estimation. The final formula of RMM is shown as below:

$$
\begin{aligned}
\boldsymbol{X}_h^{(r)} &= \boldsymbol{X}_0 + \psi_1(h)\boldsymbol{V}_0 - h\psi_1(h-ah)\nabla f(\boldsymbol{X}_{ah}^{(l)}) + \boldsymbol{e}_{x,[0,h]}, \\
\boldsymbol{V}_h^{(r)} &= \psi_0(h)\boldsymbol{V}_0 - h\psi_0(h-ah)\nabla f(\boldsymbol{X}_{ah}^{(l)}) + \boldsymbol{e}_{v,[0,h]}, \\
\boldsymbol{X}_{ah}^{(l)} &= \boldsymbol{X}_0 + \psi_1(ah)\boldsymbol{V}_0 - \psi_2(ah)\nabla f(\boldsymbol{X}_0) + \boldsymbol{e}_{x,[0,ah]}.
\end{aligned}
\tag{19}
$$

The bias and variance for single step can be controlled as $\sqrt{\mathbb{E}\|\mathbb{E}_a \boldsymbol{Z}_h^{(r)} - \boldsymbol{Z}_h\|_2^2} = O(h^4)$ and $\mathbb{E}\|\boldsymbol{Z}_h^{(r)} - \mathbb{E}_a \boldsymbol{Z}_h^{(r)}\|_2^2 = O(h^4)$ where $\boldsymbol{Z}$ is a vector combining both position $\boldsymbol{X}$ and momentum $\boldsymbol{V}$. After accumulation of $T/h$ iterations, the total bias is $O(h^3)$, total variance is $O(h^3)$, and the error coming from variance is $O(h^{\frac{3}{2}})$. The accumulated bias for RMM is much smaller than LPM and the newly introduced variance is the bottleneck. Therefore, the final discretization error is $O(h^{\frac{3}{2}})$ and iteration complexity is $\widetilde{O}(\varepsilon^{-\frac{2}{3}})$.

## A.2 Comparison with CV-ULD

Chatterji et al. [18] shows a gradient complexity for CV-ULD as $\widetilde{O}(N + d^{\frac{3}{2}}/(N^{\frac{3}{2}}\varepsilon^3))$, which is seemingly different from what is shown in Table 1. This is because (1) Chatterji et al. [18] defines mixing time as the number of steps to achieve $W_2(p_K, p^*) \le \varepsilon$ instead of $W_2(p_K, p^*) \le \varepsilon\sqrt{d/m}$, which is used throughout this paper. (2) Chatterji et al. [18] assumes gradient Lipschitz constant $L$ (it actually uses letter $M$) and strongly convex constant $m$ both scale linearly with the number of samples $N$.

In order to see that after adapting to the same setup, the $\widetilde{O}(N+\varepsilon^{-3})$ gradient complexity for CV-ULD is worse than $\widetilde{O}(N + N^{\frac{2}{3}}\varepsilon^{-\frac{2}{3}})$ gradient complexity for SVRG-ALUM and SAGA-ALUM, we just need to show that $\frac{7}{9}N + \frac{2}{9}\varepsilon^{-3} \ge N^{\frac{7}{9}}(\varepsilon^{-3})^{\frac{2}{9}} \ge N^{\frac{2}{3}}\varepsilon^{-\frac{2}{3}}$ and notice that the last inequality directly relaxes the order of $N$, therefore the inequality is asymptotically not tight.

## A.3 Equivalent forms of ULD

The underdamped Langevin dynamics has many commonly used forms, and they only differ with a linear transformation. We next give two examples and corresponding linear transforms to connect them with simplified form in (14).

### A.3.1 Form 1

$$
d\overline{\boldsymbol{X}}_{\bar{t}} = \xi\overline{\boldsymbol{V}}_{\bar{t}}d\bar{t}, \ d\overline{\boldsymbol{V}}_{\bar{t}} = -\boldsymbol{\nabla}\overline{f}(\overline{\boldsymbol{X}}_{\bar{t}})d\bar{t} - \gamma\xi\overline{\boldsymbol{V}}_{\bar{t}}d\bar{t} + \sqrt{2\gamma}d\overline{\boldsymbol{B}}_{\bar{t}}
\tag{20}
$$

$$
t = \xi\bar{t}, \boldsymbol{X}_t = \sqrt{\xi}\overline{\boldsymbol{X}}_{\bar{t}}, \boldsymbol{V}_t = \sqrt{\xi}\overline{\boldsymbol{V}}_{\bar{t}}, \boldsymbol{B}_t = \sqrt{\xi}\overline{\boldsymbol{B}}_{\bar{t}}, f(\sqrt{\xi}\overline{\boldsymbol{X}}) = \overline{f}(\overline{\boldsymbol{X}})
$$

$\overline{\boldsymbol{B}}_{\bar{t}}$ is still standard Brownian motion in the sense that $\mathbb{E}\|\overline{\boldsymbol{B}}_{\bar{t}}\|_2^2 = \bar{t}^2$. The stationary distribution is

$$
p^*(\overline{\boldsymbol{x}}, \overline{\boldsymbol{v}}) \propto \exp(-\overline{f}(\overline{\boldsymbol{x}}) - \frac{\xi}{2}\|\overline{\boldsymbol{v}}\|_2^2).
$$

### A.3.2 Form 2

$$
d\boldsymbol{X}_{t'}' = \boldsymbol{V}_{t'}'dt', \ d\boldsymbol{V}_{t'}' = -u\boldsymbol{\nabla}f'(\boldsymbol{X}_{t'}')dt' - \gamma\boldsymbol{V}_{t'}'dt' + \sqrt{2\gamma u}d\boldsymbol{B}_{t'}'
\tag{21}
$$

$$
t = t', \boldsymbol{X}_t = \frac{1}{\sqrt{u}}\boldsymbol{X}_{t'}', \boldsymbol{V}_t = \frac{1}{\sqrt{u}}\boldsymbol{V}_{t'}', \boldsymbol{B}_t = \boldsymbol{B}_{t'}', f(\frac{1}{\sqrt{u}}\boldsymbol{X}') = f'(\boldsymbol{X}'),
$$

$\boldsymbol{B}'_{t'}$ is still standard Brownian motion in the sense that $\mathbb{E}\|\boldsymbol{B}'_{t'}\|_2^2 = t'^2$. The stationary distribution is

$$p^*(\boldsymbol{x}', \boldsymbol{v}') \propto \exp(-f'(\boldsymbol{x}') - \frac{1}{2u}\|\boldsymbol{v}'\|_2^2).$$

## A.4 The integral form of ULD

$$
\begin{aligned}
\boldsymbol{X}_t &= \boldsymbol{X}_0 + \int_0^t \boldsymbol{V}_s ds \\
&= \boldsymbol{X}_0 + \int_0^t \psi_0(s)\boldsymbol{V}_0 ds - \int_0^t \int_0^u \psi_0(u-s)\nabla f(\boldsymbol{X}_s)dsdu + \sqrt{2\gamma}\int_0^t \int_0^u \psi_0(u-s)d\boldsymbol{B}_s du \\
&= \boldsymbol{X}_0 + \psi_1(s)\boldsymbol{V}_0 - \int_0^t \psi_1(t-s)\nabla f(\boldsymbol{X}_s)ds + \sqrt{2\gamma}\int_0^t \psi_1(t-s)d\boldsymbol{B}_s \\
\boldsymbol{V}_t &= e^{-\gamma t}\boldsymbol{V}_0 - \int_0^t e^{-\gamma(t-s)}\nabla f(\boldsymbol{X}_s)ds + \sqrt{2\gamma}\int_0^t e^{-\gamma(t-s)}d\boldsymbol{B}_s \\
&= \psi_0(t)\boldsymbol{V}_0 - \int_0^t \psi_0(t-s)\nabla f(\boldsymbol{X}_s)ds + \sqrt{2\gamma}\int_0^t \psi_0(t-s)d\boldsymbol{B}_s
\end{aligned}
$$

$$(22)$$

## A.5 Covariance of noise term in ALUM

For different $k$, $\boldsymbol{e}_{x,[0,h],k}$, $\boldsymbol{e}_{v,[0,h],k}$, $\boldsymbol{e}_{x,[0,a_kh],k}$ are independent with each other. For same $k$, the covarainces between these Gaussian random vector is as follows.

$$
\begin{aligned}
\mathbb{E}\left[\boldsymbol{e}_{x,[0,t]}\boldsymbol{e}_{x,[0,t]}^\top\right] &= 2\gamma I \int_0^t \psi_1(s)^2 ds = \frac{2\gamma t - 3 + 4\exp(-\gamma t) - \exp(-2\gamma t)}{\gamma^2}I \\
\mathbb{E}\left[\boldsymbol{e}_{x,[0,t]}\boldsymbol{e}_{v,[0,t]}^\top\right] &= 2\gamma I \int_0^t \psi_1(s)\psi_0(s) ds = \frac{4\sinh^2 \frac{\gamma t}{2}\exp(-\gamma t)}{\gamma}I \\
\mathbb{E}\left[\boldsymbol{e}_{v,[0,t]}\boldsymbol{e}_{v,[0,t]}^\top\right] &= 2\gamma I \int_0^t \psi_0(s)^2 ds = (1 - \exp(-2\gamma t))I \\
\mathbb{E}\left[\boldsymbol{e}_{x,[0,t]}\boldsymbol{e}_{x,[0,at]}^\top\right] &= 2\gamma I \int_0^{at} \psi_1(t-s)\psi_1(at-s) ds = \frac{2a\gamma t - 2 - 4\exp(-\gamma t)\sinh^2 \frac{a\gamma t}{2} + 2\exp(-a\gamma t)}{\gamma^2}I \\
\mathbb{E}\left[\boldsymbol{e}_{v,[0,t]}\boldsymbol{e}_{x,[0,at]}^\top\right] &= 2\gamma I \int_0^{at} \psi_0(t-s)\psi_1(at-s) ds = \frac{4\sinh^2 \frac{a\gamma t}{2}\exp(-\gamma t)}{\gamma}I \\
\mathbb{E}\left[\boldsymbol{e}_{x,[0,at]}\boldsymbol{e}_{x,[0,at]}^\top\right] &= 2\gamma I \int_0^{at} \psi_1(at-s)^2 ds = \frac{2a\gamma t - 3 + 4\exp(-a\gamma t) - \exp(-2a\gamma t)}{\gamma^2}I
\end{aligned}
$$

$$(23)$$

## A.6 Full description of VR-ALUM

We give full description of VR-ALUM in Algorithm 2.

## A.7 Initialization

In order to establish asymptotic convergence guarantee, we need to ensure that the initialization is not too far away from the target distribution. We introduce several popular initializations below that all can be summarized as $W_2(p_0, p^*) = O(1)\sqrt{d/m}$.

**Start from optimal** The global optimal could be chosen as initial point [6]. We could control the initial distance as $W_2(\delta_{\boldsymbol{x}^*}, p^*) \leq \sqrt{\frac{d}{m}}$.

---

[10]SAGA could be implemented in a cleverer way by caching the gradient, sum of gradient, and don't store $\phi_k^i$.

---

**Algorithm 2:** Variance reduced ALUM Method

---

**Input:** Initial point $(\boldsymbol{x}_0^{(o)\widetilde{\nabla}}, \boldsymbol{v}_0^{(o)\widetilde{\nabla}})$, parameter $\gamma$, iteration number $K$ batch size $b$, and step size $h > 0$.

**for** $k = 0$ **to** $K - 1$ **do**

    Randomly sample $a_k$ uniformly from $[0, 1]$;

    Generate $\boldsymbol{e}_{x,[0,h],k}$, $\boldsymbol{e}_{v,[0,h],k}$, $\boldsymbol{e}_{x,[0,a_kh],k}$ according to Appendix A.5;

    $\boldsymbol{x}_k^{(e)\widetilde{\nabla}} = \boldsymbol{x}_k^{(o)\widetilde{\nabla}} + \psi_1(a_kh)\boldsymbol{v}_k^{(o)\widetilde{\nabla}} + \boldsymbol{e}_{x,[0,a_kh],k}$;

    Randomly sample without replacement a batch $B_k$ of size $b$ from $[N]$;

    **if** SVRG is used **then**

        **if** $k \mod \tau = 0$ **then**

            $\overline{\boldsymbol{x}} \leftarrow \boldsymbol{x}_k^{(e)\widetilde{\nabla}}$;

        **end if**

        $\widetilde{\nabla}_k = \frac{N}{b} \sum_{i \in B_k} (\nabla f_i(\boldsymbol{x}_k^{(e)\widetilde{\nabla}}) - \nabla f_i(\overline{\boldsymbol{x}})) + \sum_{i=1}^{N} \nabla f_i(\overline{\boldsymbol{x}})$;

    **else if** SAGA is used[10] **then**

        $\widetilde{\nabla}_k = \frac{N}{b} \sum_{i \in B_k} (\nabla f_i(\boldsymbol{x}_k^{(e)\widetilde{\nabla}}) - \nabla f_i(\boldsymbol{\phi}_k^i)) + \sum_{i=1}^{N} \nabla f_i(\boldsymbol{\phi}_k^i)$;

        **for** $i \in B_k$ **do**

            $\boldsymbol{\phi}_{k+1}^i = \boldsymbol{x}_k^{(e)\widetilde{\nabla}}$;

        **end for**

        **for** $i \notin B_k$ **do**

            $\boldsymbol{\phi}_{k+1}^i = \boldsymbol{\phi}_k^i$;

        **end for**

    **end if**

    $\boldsymbol{x}_{k+1}^{(o)\widetilde{\nabla}} = \boldsymbol{x}_k^{(o)\widetilde{\nabla}} + \psi_1(h)\boldsymbol{v}_k^{(o)\widetilde{\nabla}} - h\psi_1(h - a_kh)\widetilde{\nabla}_k + \boldsymbol{e}_{x,[0,h],k}$;

    $\boldsymbol{v}_{k+1}^{(o)\widetilde{\nabla}} = \psi_0(h)\boldsymbol{v}_k^{(o)\widetilde{\nabla}} - h\psi_0(h - a_kh)\widetilde{\nabla}_k + \boldsymbol{e}_{v,[0,h],k}$;

**end for**

**Output:** $\boldsymbol{x}_K^{(o)\widetilde{\nabla}}$.

---

**Start from neighbor of optimal** A starting point $x_0$ could be a fixed point that is not far away from global optimal [5] such that $\|\boldsymbol{x}_0 - \boldsymbol{x}^*\|_2 \leq D$. We have $W_2(\delta_{\boldsymbol{x}_0}, p^*) \leq \sqrt{\frac{d}{m}} + \gamma D$, therefore, if we can find a point $\boldsymbol{x}_0$ close enough to $\boldsymbol{x}^*$, such that $D = O(1)\sqrt{\frac{d}{m}}$, the assumption $W_2(\delta_{\boldsymbol{x}^*}, p^*) \leq \sqrt{\frac{d}{m}}$ could be assured.

We emphasize that the assumption $W_2(p_0, p^*) = O(1)\sqrt{d/m}$ is only introduced to derive asymptotic results. Our non-asymptotic result in Theorems 1 to 4 doesn't use the assumption on the initialization at all. Therefore, our methods don't require a warm start, and non-asymptotic guarantees always hold regardless of what initialization we chose.

### A.8 Derivation of iteration complexity for full gradient ALUM

For sampling problem, we choose step size $h = O(\min(\varepsilon^{\frac{2}{3}}, m))$ and run ALUM to estimate ULD at time $T = Kh = O(m^{-1}\log(1/(\varepsilon)))$ where $K$ is the number of steps. According to theorem 1, we could achieve error in 2-Wasserstein distance as $O(\varepsilon\sqrt{\frac{d}{m}})$. Then we have the iteration complexity of full gradient ALUM as $K = \widetilde{O}(\kappa/\varepsilon^{\frac{2}{3}} + \kappa^2)$.

For ULD estimation problem, we just choose $h = O(\min(m^{\frac{2}{3}}\varepsilon^{\frac{2}{3}}d^{-\frac{1}{3}}, m))$, and run the algorithm for $K = T/h$ steps. According to theorem 2, we could achieve error in $\mathbb{L}_2$ distance as $O(\varepsilon)$.

## A.9 Derivation of iteration complexity for variance-reduced ALUM

**Sampling problem** We choose step size $h = O(\varepsilon^{\frac{2}{3}}, m, m^{\frac{1}{3}}\varepsilon^{\frac{2}{3}}bN^{-\frac{2}{3}}, m^{\frac{1}{3}}bN^{-\frac{2}{3}})$. Select $T = O(m^{-1}\log(1/(\varepsilon)))$, then the iteration complexity is $K = T/h = \widetilde{O}(\max(\kappa\varepsilon^{-\frac{2}{3}}, \kappa^2, \kappa^{\frac{4}{3}}\varepsilon^{-\frac{2}{3}}N^{\frac{2}{3}}/b, \kappa^{\frac{4}{3}}N^{\frac{2}{3}}/b))$.

Both SAGA and SVRG requires initialization, which evaluates the full gradient at the beginning. In each iteration, SAGA evaluates $b$ gradients, and SVRG uses $3b$ gradient in average. Therefore, the gradient complexity is

$$
\begin{aligned}
N + O(b)K &= \widetilde{O}(N + \max(b\kappa\varepsilon^{-\frac{2}{3}}, b\kappa^2, \kappa^{\frac{4}{3}}\varepsilon^{-\frac{2}{3}}N^{\frac{2}{3}}, \kappa^{\frac{4}{3}}N^{\frac{2}{3}})) \\
&= \widetilde{O}(N + \max(b\kappa\varepsilon^{-\frac{2}{3}}, b\kappa, b\kappa^2, \kappa^{\frac{4}{3}}\varepsilon^{-\frac{2}{3}}N^{\frac{2}{3}}, \kappa^{\frac{4}{3}}N^{\frac{2}{3}})) \\
&= \widetilde{O}(N + b\kappa\varepsilon^{-\frac{2}{3}} + b\kappa + b\kappa^2 + \kappa^{\frac{4}{3}}\varepsilon^{-\frac{2}{3}}N^{\frac{2}{3}} + \kappa^{\frac{4}{3}}N^{\frac{2}{3}}) \\
&= \widetilde{O}(N + (b\kappa + N^{\frac{2}{3}}\kappa^{\frac{4}{3}})(1 + \varepsilon^{-\frac{2}{3}}) + b\kappa^2).
\end{aligned}
\tag{24}
$$

If we only consider the dependence of $d$, $N$, $b$ and $\varepsilon$, we can see that gradient complexity is $\widetilde{O}(N + N^{\frac{2}{3}}\varepsilon^{-\frac{2}{3}})$ when $b \leq O(N^{\frac{2}{3}})$.

**Estimating ULD** We first derive the iteration complexity. Step size $h$ has two $\varepsilon$ independent upper bound in theorem 4, that is $h \leq \frac{m}{22}$ and $h^3 \leq \frac{1}{13824}b^3mN^{-2}$. We also require the right-hand side of inequality in theorem 4 to be smaller than $\varepsilon$. Therefore, we select $h = O(\min(m, bm^{\frac{1}{3}}N^{-\frac{2}{3}}, m^{\frac{2}{3}}d^{-\frac{1}{3}}\varepsilon^{\frac{2}{3}}, m^{\frac{2}{3}}d^{-\frac{1}{3}}\varepsilon^{\frac{2}{3}}bN^{-\frac{2}{3}}))$ and for a fixed $T$, we need iteration number as $K = T/h = O(\max(T\kappa, T\kappa^{\frac{1}{3}}N^{\frac{2}{3}}/b, T\kappa^{\frac{2}{3}}d^{\frac{1}{3}}\varepsilon^{-\frac{2}{3}}, \kappa^{\frac{2}{3}}d^{\frac{1}{3}}\varepsilon^{-\frac{2}{3}}N^{\frac{2}{3}}/b))$.

We next derive the gradient complexity.

$$
\begin{aligned}
N + O(b)K &= O(N + \max(T\kappa b, T\kappa^{\frac{1}{3}}N^{\frac{2}{3}}, T\kappa^{\frac{2}{3}}d^{\frac{1}{3}}\varepsilon^{-\frac{2}{3}}b, T\kappa^{\frac{2}{3}}d^{\frac{1}{3}}\varepsilon^{-\frac{2}{3}}N^{\frac{2}{3}})) \\
&= O(N + T\kappa b + T\kappa^{\frac{1}{3}}N^{\frac{2}{3}} + T\kappa^{\frac{2}{3}}d^{\frac{1}{3}}\varepsilon^{-\frac{2}{3}}b + T\kappa^{\frac{2}{3}}d^{\frac{1}{3}}\varepsilon^{-\frac{2}{3}}N^{\frac{2}{3}}) \\
&= O(N + T(\kappa b + \kappa^{\frac{1}{3}}N^{\frac{2}{3}}) + T\kappa^{\frac{2}{3}}d^{\frac{1}{3}}\varepsilon^{-\frac{2}{3}}(b + N^{\frac{2}{3}})).
\end{aligned}
\tag{25}
$$

If we only consider the dependence of $d$, $N$, $b$ and $\varepsilon$, we can see that gradient complexity is $O(N + N^{\frac{2}{3}}d^{\frac{1}{3}}\varepsilon^{-\frac{2}{3}})$ when $b \leq O(N^{\frac{2}{3}})$.

## A.10 Conjecture on the tighter $\kappa$ dependence for ALUM

The maximum upper bound of $\kappa$ for ALUM is $O(\kappa^2)$. We first provide intuition on how that result is derived.

According to inequality (102), in each step, the approximation error $A_k$ decreases with a multiplier $e^{-\frac{mh}{\gamma}}$. Meanwhile, we introduce $17h^4A_k^2$ amount of variance and $h^3A_k$ amount of bias. In order to balance the decrease and increase in each step to ensure a decrease of error, we first treat error introduced by variance as bias (ignore the useful fact that variance accumulates slower than bias). Then we just need to balance the decrease with multiplier $e^{-\frac{mh}{\gamma}}$ and increases with noise $5h^2A_k$. This indicates $h = O(m)$. In the proof, we require $h \leq \frac{m}{22}$ to establish inequality (103). The maximum step size $h \leq \frac{m}{22}$ shows up in Theorem 1 and produce $O(\kappa^2)$ dependence for ALUM.

If (102) was replaced with some other analysis that doesn't accumulate variance as bias, then we only need to balance decrease and increase as $h^3A_k$, the upper bound of step size becomes $h = O(m^{1/2})$, and the final iteration complexity has $O(\kappa^{3/2})$ dependence.

We believe a different analysis could achieve the above condition. However, it complicates the proof a lot, and has no effect on our main result, which is optimal dependence on $d,N,\varepsilon$. Therefore, we leave the rigorous theoretical analysis for future work.

## A.11 Piecewise noise terms

In this section, we show how to represent $e_{x,[0,nt]}, e_{v,[0,nt]}, e_{x,[0,ant]}$ as a combination of $e_{x,[it,(i+1)t]}$, $e_{v,[it,(i+1)t]}, e_{x,[\lfloor a \rfloor t, at]}$. This allows us to experimentally sample two ULD approximation with different step sizes, but the same realization of Brownian motion.

$$\boldsymbol{e}_{v,[0,nt]} = \sqrt{2\gamma} \int_0^{nt} \psi_0(nt-s)d\boldsymbol{B}_s$$

$$= \sqrt{2\gamma} \sum_{i=0}^{n-1} \int_{it}^{(i+1)t} \exp(-\gamma(nt-s))d\boldsymbol{B}_s$$

$$= \sum_{i=0}^{n-1} \exp(-\gamma(n-i-1)t) \left( \sqrt{2\gamma} \int_{it}^{(i+1)t} \exp(-\gamma((i+1)t-s))d\boldsymbol{B}_s \right) \tag{26}$$

$$= \sum_{i=0}^{n-1} \psi_0((n-i-1)t)\boldsymbol{e}_{v,[it,(i+1)t]}$$

$$\boldsymbol{e}_{x,[0,nt]} = \sqrt{2\gamma} \int_0^{nt} \psi_1(nt-s)d\boldsymbol{B}_s$$

$$= \sqrt{2\gamma} \sum_{i=0}^{n-1} \int_{it}^{(i+1)t} \psi_1(nt-s)d\boldsymbol{B}_s$$

$$= \sqrt{2\gamma} \sum_{i=0}^{n-1} \int_{it}^{(i+1)t} \left( \psi_1((i+1)t-s) + \psi_1((n-i-1)t)\psi_0((i+1)t-s) \right) d\boldsymbol{B}_s \tag{27}$$

$$= \sum_{i=0}^{n-1} \left( \boldsymbol{e}_{x,[it,(i+1)t]} + \psi_1((n-i-1)t)\boldsymbol{e}_{v,[it,(i+1)t]} \right)$$

We assume $0 \le a \le n$.

$$\boldsymbol{e}_{x,[0,at]} = \sqrt{2\gamma} \int_0^{at} \psi_1(at-s)d\boldsymbol{B}_s$$

$$= \sqrt{2\gamma} \sum_{i=0}^{\lceil a \rceil-1} \int_{it}^{\min(i+1,a)t} \psi_1(nt-s)d\boldsymbol{B}_s$$

$$= \sqrt{2\gamma} \sum_{i=0}^{\lceil a \rceil-1} \begin{cases} \int_{it}^{at} \psi_1(at-s)d\boldsymbol{B}_s & i = \lceil a \rceil - 1 \\ \int_{it}^{(i+1)t} \left( \psi_1((i+1)t-s) + \psi_1((a-i-1)t)\psi_0((i+1)t-s) \right) d\boldsymbol{B}_s & \text{otherwise} \end{cases}$$

$$= \sum_{i=0}^{\lceil a \rceil-1} \begin{cases} \boldsymbol{e}_{x,[\lfloor a \rfloor t,at]} & i = \lceil a \rceil - 1 \\ \boldsymbol{e}_{x,[it,(i+1)t]} + \psi_1((a-i-1)t)\boldsymbol{e}_{v,[it,(i+1)t]} & \text{otherwise} \end{cases}$$

$$= \boldsymbol{e}_{x,[\lfloor a \rfloor t,at]} + \sum_{i=0}^{\lfloor a \rfloor-1} \left( \boldsymbol{e}_{x,[it,(i+1)t]} + \psi_1((a-i-1)t)\boldsymbol{e}_{v,[it,(i+1)t]} \right)$$

$$\tag{28}$$

## B  Proof of convergence result and discretization error

We first unify the full gradient version and variance-reduced version in appendix B.1, then introduce some basic concepts in appendices B.2 to B.4, then we analyze the sampling error and approximation error in appendices B.5 and B.6.

### B.1  Statement of general result

We first formulate a general version for theorem 1 and theorem 3, which uses the bounded MSE property defined in appendix B.3.

**Theorem 7.** *Suppose we use some unbiased gradient estimation method that satisfies bounded MSE property with parameter $\Theta$. Suppose assumptions 1 to 3 holds. Given an initial distribution $p_0(\boldsymbol{x})$, we initialize an ALUM with random $\boldsymbol{x}_0^{(o)\widetilde{\nabla}}$ based on probability $p_0$ and random $\boldsymbol{v}_0^{(o)\widetilde{\nabla}}$ from standard*

*Gaussian distribution. Assume $L = 1$ and let $\gamma = 2$, $p_k$ be the distribution of $\boldsymbol{x}_k^{(o)\widetilde{\nabla}}$ and $p^*$ be the target distribution. Assume we use step size $h \le \frac{m}{11\gamma}$ and $\frac{h^3\Theta}{m} \le \frac{1}{2304}$. After running the ALUM for $k$ iterations, for some $A_0 \le \gamma W_2(p_0, p^*)$, we have the following upper bound of sampling error in 2-Wasserstein distance.*

$$W_2(p_k, p^*) \le (1 - \frac{mh}{2\gamma})^k A_0 + 46\sqrt{\frac{h^3}{m}}\sqrt{\Theta}A_0 + (12 + 57\sqrt{\Theta})\sqrt{\frac{h^3}{m}}\sqrt{d}. \tag{29}$$

The next theorem is general version result for theorem 2 and theorem 4.

**Theorem 8.** *With same assumptions and setup in theorem 7, we consider $\boldsymbol{X}_{kh}$ which comes from a continuous ULD starting from same position as $\boldsymbol{x}_0^{(o)\widetilde{\nabla}}$ with probability 1. We have following upper bound of discretization error*

$$\|\boldsymbol{x}_k^{(o)} - \boldsymbol{X}_{kh}\|_{\mathbb{L}_2} \le (19 + 46\sqrt{\Theta})\sqrt{\frac{h^3}{m}}A_0 + (31 + 57\sqrt{\Theta})\sqrt{\frac{h^3 d}{m}}. \tag{30}$$

In order to get all theorems in Section 5, we just need to use the fact that $\Theta = 0$ for full gradient ALUM, $\Theta \le \frac{N^2}{b^3}$ for SVRG-ALUM and $\Theta \le 2\frac{N^2}{b^3}$ for SAGA-ALUM.

## B.2 Preliminary

We first introduce several notations.

$$\|\boldsymbol{v}\|_{\mathbb{L}_2} = \sqrt{\mathbb{E}[\|\boldsymbol{v}\|_2^2]}$$
$$\|\boldsymbol{v}\|_{\mathbb{L}_2,B} = \sqrt{\mathbb{E}_B[\|\boldsymbol{v}\|_2^2]} \tag{31}$$
$$\|\boldsymbol{v}\|_{\mathbb{L}_2,a} = \sqrt{\mathbb{E}_a[\|\boldsymbol{v}\|_2^2]}$$

$\mathbb{E}_B[\cdot]$ takes expectation with regard to random Brownian motion. $\mathbb{E}_a[\cdot]$ means average for random variable $a$ which determines randomized midpoint in RMM or ALUM.

Next we introduce another assumption to simplify the analysis.

**Assumption 4** (Optimal at Zero). *Without loss of generality, we assume $\boldsymbol{x}^* = 0$ and $f(\boldsymbol{x}^*) = 0$ where $\boldsymbol{x}^*$ is the global minimum for the strongly convex potential energy function.*

**Remark 3.** *Assumption 4 doesn't mean that we have to calculate the global optimal of function $f(\boldsymbol{x})$ and shift the function. ULD is invariant under shifting the coordinate. More specifically, the stochastic process on a shifted potential function and shifted start point should be exactly the same as shifted process obtained based on original potential function and start point. The Wasserstein distance is also invariant under this shifting if we translate both target distribution and obtained distribution. Therefore, we can safely introduce Assumption 4 to simplify the analysis without really translate the potential.*

## B.3 Bounded MSE property

Given $\boldsymbol{x}_k^{(e)}$ for $k \ge 0$ as a series of points provided sequentially for estimating the gradient. A gradient estimation method generates $\widetilde{\nabla}_k$ as random vectors for estimating $\nabla f(\boldsymbol{x}_k^{(e)})$. We let $\mathbb{E}_k[\cdot]$ means taking expectation for randomness only after $k$-th iteration in the gradient estimator. For example, if vanilla SGD, SVRG or SAGA is used, the randomness comes from the random batch and $\mathbb{E}_k[\widetilde{\nabla}_{k+1}]$ is same as $\mathbb{E}_{B_{k+1}}[\widetilde{\nabla}_{k+1}]$ that is expectation with respect to random batch in $(k+1)$-th iteration.

In this section, we propose bounded MSE property to control the mean-squared error (MSE) of variance reduction methods.

**Definition 1** (Bounded MSE property). *We say a gradient estimation method satisfies the bounded MSE property if (1) it is unbiased, which means $\mathbb{E}_k[\widetilde{\nabla}_{k+1}] = \nabla f(\boldsymbol{x}_{k+1}^{(e)}) = \sum_{i=1}^N \nabla f_i(\boldsymbol{x}_{k+1}^{(e)})$. (2) it*

*has the following upper bound for some parameter $\Theta$ and all $k \geq 0$.*

$$\mathbb{E}[\|\widetilde{\nabla}_{k+1} - \nabla f(\boldsymbol{x}_{k+1}^{(e)})\|_2^2] \leq \Theta \max_{0 \leq i \leq k} Q_i,$$

$$\|\widetilde{\nabla}_0 - \nabla f(\boldsymbol{x}_0^{(e)})\|_2^2 = 0, \tag{32}$$

$$Q_k = N \sum_{i=1}^{N} \|\nabla f_i(\boldsymbol{x}_{k+1}^{(e)}) - \nabla f_i(\boldsymbol{x}_k^{(e)})\|_2^2.$$

Clearly, the full gradient estimator $\widetilde{\nabla}_k = \nabla f(\boldsymbol{x}_k^{(e)})$ satisfies bounded MSE property with $\Theta = 0$ because there is no gradient error. We next show both SVRG and SAGA satisfy bounded MSE property.

**Lemma 1.** *SVRG with epoch length $\tau$ satisfies bounded MSE property with $\Theta = \frac{N-b}{N-1} \frac{(\tau-1)^2}{b} \leq \frac{\tau^2}{b}$.*

**Lemma 2.** *SAGA satisfies bounded MSE property with $\Theta = \frac{N(N-b)(2N-b)}{(N-1)b^3} \leq \frac{2N^2}{b^3}$.*

We show the proof in appendix B.3.1 and appendix B.3.2 separately.

### B.3.1 Proof of lemma 1 for SVRG

Recall the definition of SVRG as follows.

$$\widetilde{\nabla}_k^{\text{SVRG}} = \frac{N}{b} \sum_{i \in B_k} (\nabla f_i(\boldsymbol{x}_k^{(e)}) - \nabla f_i(\overline{\boldsymbol{x}})) + \sum_{i=1}^{N} \nabla f_i(\overline{\boldsymbol{x}}) \tag{33}$$

For $k = 0$, there is no gradient error because the full gradient is just calculated. For $k \geq 1$, we define $\boldsymbol{X}_i = \nabla f_i(\boldsymbol{x}_k^{(e)}) - \nabla f_i(\overline{\boldsymbol{x}})$, $\boldsymbol{Y}_i = N\boldsymbol{X}_i - \sum_{i=1}^{N} \boldsymbol{X}_i$. We then have

$$\sum_{i=1}^{N} \boldsymbol{Y}_i = 0 \tag{34}$$

$$\widetilde{\nabla}_k^{\text{SVRG}} - \sum_{i=1}^{N} \nabla f_i(\boldsymbol{x}_k^{(e)}) = \frac{1}{b} \sum_{i \in B_k} \boldsymbol{Y}_i \tag{35}$$

**Lemma 3.**

$$\mathbb{E}_{B_k} \left\| \frac{1}{b} \sum_{i \in B_k} \boldsymbol{Y}_i \right\|_2^2 = \frac{N-b}{N-1} \frac{1}{Nb} \sum_{i=1}^{N} \|\boldsymbol{Y}_i\|_2^2 \tag{36}$$

Due to the fact that batch $B_k$ is sampling without replacement, controlling the norm is not that straightforward, therefore we show the proof separately in appendix B.3.3.

$$\sum_{i=1}^{N} \|\boldsymbol{Y}_i\|_2^2 = N^2 \sum_{i=1}^{N} \|\boldsymbol{X}_i\|_2^2 - N \left\| \sum_{i=1}^{N} \boldsymbol{X}_i \right\|_2^2 \leq N^2 \sum_{i=1}^{N} \|\boldsymbol{X}_i\|_2^2 \tag{37}$$

Therefore, we reach following inequality

$$\mathbb{E}_{B_k} \|\widetilde{\nabla}_k^{\text{SVRG}} - \nabla f_i(\boldsymbol{x}_k^{(e)})\|_2^2 \leq \frac{N-b}{N-1} \frac{N}{b} \sum_{i=1}^{N} \|\boldsymbol{X}_i\|_2^2 \tag{38}$$

We next give upper bound on $\sum_{i=1}^{N} \|\boldsymbol{X}_i\|_2^2$.

According to definition of SVRG, we know $\overline{x} = x_{k'}^{(e)}$ where $k'$ is the last iteration that the algorithm update the full gradient. Therefore, we have $k - \tau + 1 \le k' \le k$.

$$
\begin{aligned}
\sum_{i=1}^{N} \|\boldsymbol{X}_i\|_2^2 &= \sum_{i=1}^{N} \left\| \nabla f_i(\boldsymbol{x}_k^{(e)}) - \nabla f_i(\overline{\boldsymbol{x}}) \right\|_2^2 \\
&= \sum_{i=1}^{N} \left\| \nabla f_i(\boldsymbol{x}_k^{(e)}) - \nabla f_i(\boldsymbol{x}_{k'}^{(e)}) \right\|_2^2 \\
&\le \sum_{i=1}^{N} (k - k') \sum_{t=k'}^{k-1} \left\| \nabla f_i(\boldsymbol{x}_{t+1}^{(e)}) - \nabla f_i(\boldsymbol{x}_t^{(e)}) \right\|_2^2 \\
&= (k - k') \sum_{t=k'}^{k-1} \sum_{i=1}^{N} \left\| \nabla f_i(\boldsymbol{x}_{t+1}^{(e)}) - \nabla f_i(\boldsymbol{x}_t^{(e)}) \right\|_2^2 \\
&= (k - k') \sum_{t=k'}^{k-1} \frac{1}{N} Q_t \\
&\le \frac{(k - k')^2}{N} \max_{i<k} Q_i \\
&\le \frac{(\tau - 1)^2}{N} \max_{i<k} Q_i
\end{aligned}
\tag{39}
$$

Finally, we have MSE upper bound as follows.

$$
\mathbb{E}_{B_k} \| \widetilde{\nabla}_k^{\text{SVRG}} - \sum_{i=1}^{N} \nabla f_i(\boldsymbol{x}_k^{(e)}) \|_2^2 \le \frac{N - b}{N - 1} \frac{(\tau - 1)^2}{b} \max_{i<k} Q_i.
\tag{40}
$$

### B.3.2 Proof of lemma 2 for SAGA

Recall the definition of SAGA as follows.

$$
\widetilde{\nabla}_k^{\text{SAGA}} = \frac{N}{b} \sum_{i \in B_k} (\nabla f_i(\boldsymbol{x}_k^{(e)}) - \nabla f_i(\boldsymbol{\phi}_k^i)) + \sum_{i=1}^{N} \nabla f_i(\boldsymbol{\phi}_k^i)
\tag{41}
$$

Again, for $k = 0$, there is no gradient error, and for $k \ge 1$, we define $\boldsymbol{X}_i = \nabla f_i(\boldsymbol{x}_k^{(e)}) - \nabla f_i(\boldsymbol{\phi}_k^i)$. Similar to (38) in appendix B.3.1, we can reach the following result.

$$
\mathbb{E}_{B_k} \| \widetilde{\nabla}_k^{\text{SAGA}} - \nabla f_i(\boldsymbol{x}_k^{(e)}) \|_2^2 \le \frac{N - b}{N - 1} \frac{N}{b} \sum_{i=1}^{N} \|\boldsymbol{X}_i\|_2^2
\tag{42}
$$

We next give upper bound on $\sum_{i=1}^{N} \|\boldsymbol{X}_i\|_2^2$.

We define $M_u = \sum_{i=1}^{N} \|\nabla f_i(\boldsymbol{x}_k^{(e)}) - \nabla f_i(\boldsymbol{\phi}_u^i)\|_2^2$ for $k \ge u \ge 0$. We can see that $\sum_{i=1}^{N} \|\boldsymbol{X}_i\|_2^2 = M_k$.

$$
\begin{aligned}
M_0 &= \sum_{i=1}^{N} \|\nabla f_i(\boldsymbol{x}_k^{(e)}) - \nabla f_i(\boldsymbol{x}_0^{(e)})\|_2^2 \\
&\le \sum_{i=1}^{N} k \sum_{t=0}^{k-1} \left\| \nabla f_i(\boldsymbol{x}_{t+1}^{(e)}) - \nabla f_i(\boldsymbol{x}_t^{(e)}) \right\|_2^2 \\
&= k \sum_{t=0}^{k-1} \sum_{i=1}^{N} \left\| \nabla f_i(\boldsymbol{x}_{t+1}^{(e)}) - \nabla f_i(\boldsymbol{x}_t^{(e)}) \right\|_2^2 \\
&= k \sum_{t=0}^{k-1} \frac{1}{N} Q_t \\
&\le \frac{k^2}{N} \max_{i<k} Q_i
\end{aligned}
\tag{43}
$$

We next define indicator $\mathbb{I}_{i,u} = 1$ if and only if $i \in B_u$ and $\mathbb{I}_{i,u} = 0$ otherwise.

$$
\begin{aligned}
\mathbb{E}_{B_{u-1}}[M_u] =& \mathbb{E}_{B_{u-1}} \sum_{i=1}^{N} \left( \mathbb{I}_{i,u-1} \| \nabla f_i(\boldsymbol{x}_k^{(e)}) - \nabla f_i(\boldsymbol{x}_{u-1}^{(e)}) \|_2^2 + (1 - \mathbb{I}_{i,u-1}) \| \nabla f_i(\boldsymbol{x}_k^{(e)}) - \nabla f_i(\boldsymbol{\phi}_{u-1}^i) \|_2^2 \right) \\
=& \frac{b}{N} \sum_{i=1}^{N} \| \nabla f_i(\boldsymbol{x}_k^{(e)}) - \nabla f_i(\boldsymbol{x}_{u-1}^{(e)}) \|_2^2 + \frac{N-b}{N} M_{u-1} \\
\leq& \frac{b}{N} \sum_{i=1}^{N} (k - u + 1) \sum_{t=u-1}^{k-1} \| \nabla f_i(\boldsymbol{x}_{t+1}^{(e)}) - \nabla f_i(\boldsymbol{x}_t^{(e)}) \|_2^2 + \frac{N-b}{N} M_{u-1} \\
=& \frac{b}{N} (k - u + 1) \sum_{t=u-1}^{k-1} \sum_{i=1}^{N} \| \nabla f_i(\boldsymbol{x}_{t+1}^{(e)}) - \nabla f_i(\boldsymbol{x}_t^{(e)}) \|_2^2 + \frac{N-b}{N} M_{u-1} \\
=& \frac{b}{N} (k - u + 1) \sum_{t=u-1}^{k-1} \frac{1}{N} Q_t + \frac{N-b}{N} M_{u-1} \\
\leq& \frac{b}{N^2} (k - u + 1)^2 \max_{i<k} Q_i + \frac{N-b}{N} M_{u-1}
\end{aligned}
$$
(44)

Combining the above two inequalities, we have

$$
\begin{aligned}
\mathbb{E}[M_u] \leq& \left( (\frac{N-b}{N})^k \frac{k^2}{N} + \sum_{u=1}^{k} (\frac{N-b}{N})^{k-u} \frac{b(k-u+1)^2}{N^2} \right) \max_{i<k} Q_i \\
=& S(k) \max_{i<k} Q_i
\end{aligned}
$$
(45)

The function $S(k)$ increases monotonically for $k$ because $S(k+1) - S(k) = (\frac{N-b}{N})^k \frac{2k+1}{N} \geq 0$. We then have $S(k) \leq \lim_{k \to \infty} S(k) = \frac{2N-b}{b^2}$. Finally, we have MSE upper bound as follows.

$$
\mathbb{E} \| \widetilde{\nabla}_k^{\text{SAGA}} - \nabla f_i(\boldsymbol{x}_k^{(e)})) \|_2^2 \leq \frac{N(N-b)(2N-b)}{(N-1)b^3} \max_{i<k} Q_i.
$$
(46)

### B.3.3 Other

*Proof of lemma 3.* We define indicator $\mathbb{I}_i$ such that $\mathbb{I}_i = 1$ if and only if $i \in B_k$ and $\mathbb{I}_0 = 0$ otherwise. Since $B_k$ is a simple random sample of size $b$ from a population of $N$ elements, we have

$$
\mathbb{E}_{B_k}[\mathbb{I}_i] = \frac{b}{N}, \mathbb{E}_{B_k}[\mathbb{I}_i \mathbb{I}_j] = \frac{b}{N} \frac{b-1}{N-1}, \forall i \neq j
$$
(47)

$$
\begin{aligned}
\mathbb{E}_{B_k} \left\| \sum_{i \in B_k} \boldsymbol{Y}_i \right\|_2^2 =& \mathbb{E}_{B_k} \left( \sum_{i \in B_k} \| \boldsymbol{Y}_i \|_2^2 + \sum_{i \neq j \in B_k} \langle \boldsymbol{Y}_i, \boldsymbol{Y}_j \rangle^2 \right) \\
=& \mathbb{E}_{B_k} \left( \sum_{i=1}^{N} \mathbb{I}_i \| \boldsymbol{Y}_i \|_2^2 + \sum_{i \neq j} \mathbb{I}_i \mathbb{I}_j \langle \boldsymbol{Y}_i, \boldsymbol{Y}_j \rangle^2 \right) \\
=& \sum_{i=1}^{N} \frac{b}{N} \| \boldsymbol{Y}_i \|_2^2 + \sum_{i \neq j} \frac{b}{N} \frac{b-1}{N-1} \langle \boldsymbol{Y}_i, \boldsymbol{Y}_j \rangle^2 \\
=& \frac{N-b}{N-1} \frac{b}{N} \sum_{i=1}^{N} \| \boldsymbol{Y}_i \|_2^2 + \frac{b}{N} \frac{b-1}{N-1} \left\langle \sum_{i=1}^{N} \boldsymbol{Y}_i, \sum_{j=1}^{N} \boldsymbol{Y}_j \right\rangle^2 \\
=& \frac{N-b}{N-1} \frac{b}{N} \sum_{i=1}^{N} \| \boldsymbol{Y}_i \|_2^2
\end{aligned}
$$
(48)

The last equality comes from the fact that $\sum_{i=1}^{N} \boldsymbol{Y}_i = 0$. $\qquad\square$

## B.4 Continuous Contraction

We first define several vectors useful in the analysis by linear combination or direct sum of $\boldsymbol{X}$ and $\boldsymbol{V}$.

$$\boldsymbol{R}_t = \gamma \boldsymbol{X}_t + \boldsymbol{V}_t, \boldsymbol{S}_t = \boldsymbol{V}_t, \boldsymbol{Z}_t = \begin{bmatrix} \boldsymbol{R}_t \\ \boldsymbol{S}_t \end{bmatrix} \tag{49}$$

$$\boldsymbol{Z}_t = \begin{bmatrix} \boldsymbol{R}_t \\ \boldsymbol{S}_t \end{bmatrix} = \begin{bmatrix} \gamma & 1 \\ 0 & 1 \end{bmatrix} \begin{bmatrix} \boldsymbol{X}_t \\ \boldsymbol{V}_t \end{bmatrix} \tag{50}$$

$$\begin{bmatrix} \boldsymbol{X}_t \\ \boldsymbol{V}_t \end{bmatrix} = \frac{1}{\gamma} \begin{bmatrix} 1 & -1 \\ 0 & \gamma \end{bmatrix} \begin{bmatrix} \boldsymbol{R}_t \\ \boldsymbol{S}_t \end{bmatrix} \tag{51}$$

The following contraction result for continuous process is essentially the same as analysis in [17]. We rephrase them here in a form that is more coherent with the other part of this paper.

**Lemma 4.** *For any two ULD processes* $\begin{bmatrix} \boldsymbol{X}_t \\ \boldsymbol{Z}_t \end{bmatrix}$ *and* $\begin{bmatrix} \boldsymbol{X}'_t \\ \boldsymbol{Z}'_t \end{bmatrix}$ *with the same Brownian motion* $\boldsymbol{B}_t$, *we define* $\Delta \boldsymbol{X}_t = \boldsymbol{X}_t - \boldsymbol{X}'_t$ *and similarly for* $\Delta \boldsymbol{V}_t$, $\Delta \boldsymbol{R}_t$, $\Delta \boldsymbol{S}_t$ *and* $\Delta \boldsymbol{Z}_t$. *If* $\gamma \geq \sqrt{m + L}$, *then we have*

$$\|\Delta \boldsymbol{Z}_t\|_2 \leq e^{-\frac{m}{\gamma}t} \|\Delta \boldsymbol{Z}_0\|_2 \tag{52}$$

*Proof of Lemma 4.*

$$
\begin{aligned}
d\Delta \boldsymbol{Z}_t &= \begin{bmatrix} \gamma & 1 \\ 0 & 1 \end{bmatrix} \begin{bmatrix} d\Delta \boldsymbol{X}_t \\ d\Delta \boldsymbol{V}_t \end{bmatrix} \\
&= \begin{bmatrix} \gamma & 1 \\ 0 & 1 \end{bmatrix} \begin{bmatrix} \Delta \boldsymbol{V}_t \\ \nabla f(\boldsymbol{X}'_t) - \nabla f(\boldsymbol{X}_t) - \gamma \Delta \boldsymbol{V}_t \end{bmatrix} dt \\
&\quad \nabla f(\boldsymbol{X}'_t) - \nabla f(\boldsymbol{X}_t) = -H_t \Delta \boldsymbol{X}_t \\
&\quad H_t = \int_0^1 \nabla^2 f(\boldsymbol{X}'_t + u \Delta \boldsymbol{X}_t) du \\
&\quad mI \preccurlyeq H_t \preccurlyeq LI
\end{aligned}
\tag{53}
$$

$$
\begin{aligned}
d\Delta \boldsymbol{Z}_t &= \begin{bmatrix} \gamma & 1 \\ 0 & 1 \end{bmatrix} \begin{bmatrix} 0 & 1 \\ -H_t & -\gamma \end{bmatrix} \begin{bmatrix} \Delta \boldsymbol{X}_t \\ \Delta \boldsymbol{V}_t \end{bmatrix} \\
&= \frac{1}{\gamma} \begin{bmatrix} \gamma & 1 \\ 0 & 1 \end{bmatrix} \begin{bmatrix} 0 & 1 \\ -H_t & -\gamma \end{bmatrix} \begin{bmatrix} 1 & -1 \\ 0 & \gamma \end{bmatrix} \begin{bmatrix} \Delta \boldsymbol{R}_t \\ \Delta \boldsymbol{S}_t \end{bmatrix} \\
&= \frac{1}{\gamma} \begin{bmatrix} -H_t & H_t \\ -H_t & H_t - \gamma^2 \end{bmatrix} \begin{bmatrix} \Delta \boldsymbol{R}_t \\ \Delta \boldsymbol{S}_t \end{bmatrix}
\end{aligned}
\tag{54}
$$

$$
\begin{aligned}
d\|\Delta \boldsymbol{Z}_t\|_2^2 &= \frac{2}{\gamma} \begin{bmatrix} \Delta \boldsymbol{R}_t & \Delta \boldsymbol{S}_t \end{bmatrix} \begin{bmatrix} -H_t & 0 \\ 0 & H_t - \gamma^2 \end{bmatrix} \begin{bmatrix} \Delta \boldsymbol{R}_t \\ \Delta \boldsymbol{S}_t \end{bmatrix} \\
&\leq -\frac{2m}{\gamma} \|\Delta \boldsymbol{Z}_t\|_2^2
\end{aligned}
\tag{55}
$$

$$\|\Delta \boldsymbol{Z}_t\|_2^2 \leq e^{-\frac{2m}{\gamma}t} \|\Delta \boldsymbol{Z}_0\|_2^2 \tag{56}$$

$\square$

## B.5 Sampling error

We first define a ground truth path. The initial point $\boldsymbol{x}_0^*, \boldsymbol{v}_0^*$ is drawn from target distribution $p^*(\boldsymbol{x}^*, \boldsymbol{v}^*) \propto \exp(-f(\boldsymbol{x}^*) - \frac{\|\boldsymbol{v}^*\|_2^2}{2})$. Then we generate $\boldsymbol{x}_{k+1}^*, \boldsymbol{v}_{k+1}^*$ from $\boldsymbol{x}_k^*, \boldsymbol{v}_k^*$ by taking repeatedly applying a ULD process. More specifically, we let $\boldsymbol{x}_{k+1}^* = \boldsymbol{X}_h, \boldsymbol{v}_{k+1}^* = \boldsymbol{V}_h$ where $\boldsymbol{X}_t, \boldsymbol{V}_t$ is the ULD process starting from $\boldsymbol{x}_k^*, \boldsymbol{v}_k^*$. Clearly, the distribution of $\boldsymbol{x}_k^*, \boldsymbol{v}_k^*$ is still target distribution. Finally, we can define $\boldsymbol{z}_k^* = \begin{bmatrix} \gamma \boldsymbol{x}^* + \boldsymbol{v}^* \\ \boldsymbol{v}^* \end{bmatrix}$.

We then first define $A_k = \left\| \boldsymbol{z}_k^{(o)\widetilde{\nabla}} - \boldsymbol{z}_k^* \right\|_{\mathbb{L}_2}$ as the difference between the path $\boldsymbol{z}_k^{(o)\widetilde{\nabla}} = \begin{bmatrix} \gamma \boldsymbol{x}^{(o)\widetilde{\nabla}} + \boldsymbol{v}^{(o)\widetilde{\nabla}} \\ \boldsymbol{v}^{(o)\widetilde{\nabla}} \end{bmatrix}$ obtained by our algorithm and the ground truth path. Based on (50), we have $\|\Delta Z\|_{\mathbb{L}_2} \geq \frac{\gamma}{\sqrt{2}} \|\Delta X\|_{\mathbb{L}_2} \geq \|\Delta X\|_{\mathbb{L}_2}$. Therefore, $A_k$ is an upper bound of $W_2(p_k, p^*)$.

At initialization, based on (51), we have $\left\| \begin{bmatrix} \Delta \boldsymbol{X} \\ \Delta \boldsymbol{V} \end{bmatrix} \right\|_{\mathbb{L}_2} \geq \frac{1}{\gamma} \|\Delta Z\|_{\mathbb{L}_2}$. Moreover, let distribution of $\boldsymbol{x}_0$ be $p_0(\boldsymbol{x})$, and distribution of $\boldsymbol{v}_0$ be a Gaussian distribution that is the same as $\boldsymbol{v}_0^*$. Let the coupling between $\begin{bmatrix} \boldsymbol{x}_0 \\ \boldsymbol{v}_0 \end{bmatrix}$ and $\begin{bmatrix} \boldsymbol{x}_0^* \\ \boldsymbol{v}_0^* \end{bmatrix}$ follows that $\boldsymbol{v}_0$ and $\boldsymbol{v}_0^*$ are exactly the same, and $\boldsymbol{x}_0$ and $\boldsymbol{x}_0^*$ are coupled according to the optimal transport between $p_0(\boldsymbol{x})$ and $p^*(\boldsymbol{x})$. In this way of initialization, we can get $\left\| \begin{bmatrix} \Delta \boldsymbol{x}_0 \\ \Delta \boldsymbol{v}_0 \end{bmatrix} \right\|_{\mathbb{L}_2} = W_2(p_0, p^*)$. Therefore, we have $A_0 \leq \gamma W_2(p_0, p^*)$.

The next ten pages of numerical analysis is a little bit long, so we provide an overview as follows.

### B.5.1  Overview of proof

The main idea of controlling $A_k$ is decomposing error and contraction as follows.

$$
\begin{aligned}
A_{k+1} =& \mathbb{E}\left[ \left\| \boldsymbol{Z}_t^{(o)\widetilde{\nabla}} - \boldsymbol{Z}_t^* \right\|_2^2 \right] \\
\overset{\text{\textcircled{1}}}{=}& \mathbb{E}\left[ \left\| \boldsymbol{Z}_t^{(o)\widetilde{\nabla}} - \mathbb{E}_a \mathbb{E}_k \boldsymbol{Z}_t^{(o)\widetilde{\nabla}} \right\|_2^2 \right] \\
& + \mathbb{E}\left[ \left\| \mathbb{E}_a \mathbb{E}_k \boldsymbol{Z}_t^{(o)\widetilde{\nabla}} - \boldsymbol{Z}_t^* \right\|_2^2 \right] \\
\overset{\text{\textcircled{2}}}{\leq}& \mathbb{E}\left[ \left\| \boldsymbol{Z}_t^{(o)\widetilde{\nabla}} - \mathbb{E}_a \boldsymbol{Z}_t^{(r)} \right\|_2^2 \right] \\
& + \mathbb{E}\left[ \left\| \mathbb{E}_a \mathbb{E}_k \boldsymbol{Z}_t^{(o)\widetilde{\nabla}} - \boldsymbol{Z}_t^* \right\|_2^2 \right] \\
\overset{\text{\textcircled{3}}}{=}& \mathbb{E}\left[ \left\| \boldsymbol{Z}_t^{(o)\widetilde{\nabla}} - \mathbb{E}_a \boldsymbol{Z}_t^{(r)} \right\|_2^2 \right] \\
& + \mathbb{E}\left[ \left\| \mathbb{E}_a \boldsymbol{Z}_t^{(o)} - \boldsymbol{Z}_t^* \right\|_2^2 \right] \\
\overset{\text{\textcircled{4}}}{\leq}& 3\mathbb{E}\left[ \left\| \boldsymbol{Z}_t^{(o)\widetilde{\nabla}} - \boldsymbol{Z}_t^{(o)} \right\|_2^2 \right] + 3\mathbb{E}\left[ \left\| \boldsymbol{Z}_t^{(o)} - \boldsymbol{Z}_t^{(r)} \right\|_2^2 \right] \\
& + 3\mathbb{E}\left[ \left\| \boldsymbol{Z}_t^{(r)} - \mathbb{E}_a \boldsymbol{Z}_t^{(r)} \right\|_2^2 \right] \\
& + \left( \left\| \mathbb{E}_a \boldsymbol{Z}_t^{(o)} - \mathbb{E}_a \boldsymbol{Z}_t^{(r)} \right\|_{\mathbb{L}_2} + \left\| \mathbb{E}_a \boldsymbol{Z}_t^{(r)} - \boldsymbol{Z}_t \right\|_{\mathbb{L}_2} + \left\| \boldsymbol{Z}_t - \boldsymbol{Z}_t^* \right\|_{\mathbb{L}_2} \right)^2
\end{aligned}
\tag{57}
$$

The vector $\boldsymbol{Z}^{(r)}$ is based on $\boldsymbol{X}^{(r)}, \boldsymbol{V}^{(r)}$ generated by RMM, which is defined in (19).

Inequality ① splits bias and variance and is based on the fact that $\mathbb{E}_a \mathbb{E}_k \boldsymbol{Z}_t^* = \boldsymbol{Z}_t^*$.

Inequality ② comes from the fact that $\mathbb{E}\left[ \left\| \boldsymbol{Z}_t^{(o)\widetilde{\nabla}} - \mathbb{E}_a \boldsymbol{Z}_t^{(r)} \right\|_2^2 \right] = \mathbb{E}\left[ \left\| \boldsymbol{Z}_t^{(o)\widetilde{\nabla}} - \mathbb{E}_a \mathbb{E}_k \boldsymbol{Z}_t^{(o)\widetilde{\nabla}} \right\|_2^2 \right] + \mathbb{E}\left[ \left\| \mathbb{E}_a \mathbb{E}_k \boldsymbol{Z}_t^{(o)\widetilde{\nabla}} - \mathbb{E}_a \boldsymbol{Z}_t^{(r)} \right\|_2^2 \right]$.

Inequality ③ comes from the fact that variance reduction method we used are unbiased.

Inequality ④ is simply applying Young's inequality and triangle inequality.

The last term is $\|\boldsymbol{Z}_t - \boldsymbol{Z}_t^*\|_{\mathbb{L}_2} = e^{-\frac{m}{\gamma}t}\|\boldsymbol{Z}_0 - \boldsymbol{Z}_0^*\|_{\mathbb{L}_2} = e^{-\frac{m}{\gamma}t}A_k$. Therefore, we just need to control all other error term by term. We do that in Appendices B.5.3 to B.5.6.

During the analysis, we give upper bound of errors for $A_k$ by using another value $Q_k = N\sum_{i=1}^N \mathbb{E}[\|\nabla f_i(\boldsymbol{x}_{k+1}^{(e)\widetilde{\nabla}}) - \nabla f_i(\boldsymbol{x}_k^{(e)\widetilde{\nabla}})\|_2^2]$. $Q_k$ decides the magnitudes for gradient error. In Appendix B.5.8, we give the upper bound of $Q_k$ by using $A_k$. Finally, we can give upper bound for both $Q_k$ and $A_k$ in Appendix B.5.9.

### B.5.2 Some basic upper bounds for continuous process

**Moments** First, we derive the moments for equilibrium distribution.

Due to the fact that $\boldsymbol{V}_t^*$ follows standard Gaussian distribution,

$$\mathbb{E}\left[\|\boldsymbol{V}_t^*\|_2^2\right] = d. \tag{58}$$

Next we control the moment of $\nabla f(\boldsymbol{x})$.

$$
\begin{aligned}
Zd &= \int de^{-f(\boldsymbol{x})}d\boldsymbol{x} \\
&= \int \operatorname{div}(\boldsymbol{x})e^{-f(\boldsymbol{x})}d\boldsymbol{x} \\
&= \int \boldsymbol{x}\cdot\nabla f(\boldsymbol{x})e^{-f(\boldsymbol{x})}d\boldsymbol{x} \\
&\geq \frac{1}{L}\int \|\nabla f(\boldsymbol{x})\|_2^2 e^{-f(\boldsymbol{x})}d\boldsymbol{x}
\end{aligned}
\tag{59}
$$

$$\mathbb{E}_{\boldsymbol{x}\sim p^*}\left[\|\nabla f(\boldsymbol{x})\|_2^2\right] = \frac{1}{Z}\int \|\nabla f(\boldsymbol{x})\|_2^2 e^{-f(\boldsymbol{x})}d\boldsymbol{x} \leq Ld \tag{60}$$

Then we control the moments for ULD process.

$$
\begin{aligned}
\|\boldsymbol{V}_t\|_{\mathbb{L}_2} &\leq \|\boldsymbol{V}_t^*\|_{\mathbb{L}_2} + \|\boldsymbol{V}_t - \boldsymbol{V}_t^*\|_{\mathbb{L}_2} \\
&\leq \sqrt{d} + \|\boldsymbol{Z}_t - \boldsymbol{Z}_t^*\|_{\mathbb{L}_2} \\
&\leq \sqrt{d} + \|\boldsymbol{Z}_0 - \boldsymbol{Z}_0^*\|_{\mathbb{L}_2} \\
&= \sqrt{d} + \left\|\boldsymbol{z}_k^{(o)\widetilde{\nabla}} - \boldsymbol{z}_k^*\right\|_{\mathbb{L}_2} \\
&= \sqrt{d} + A_k
\end{aligned}
\tag{61}
$$

$$
\begin{aligned}
\|\nabla f(\boldsymbol{X}_t)\|_{\mathbb{L}_2} &\leq \|\nabla f(\boldsymbol{X}_t^*)\|_{\mathbb{L}_2} + \|\nabla f(\boldsymbol{X}_t) - \nabla f(\boldsymbol{X}_t^*)\|_{\mathbb{L}_2} \\
&\leq \sqrt{Ld} + L\|\boldsymbol{X}_t - \boldsymbol{X}_t^*\|_{\mathbb{L}_2} \\
&\leq \sqrt{Ld} + L\frac{\sqrt{2}}{\gamma}\|\boldsymbol{Z}_t - \boldsymbol{Z}_t^*\|_{\mathbb{L}_2} \\
&\leq \sqrt{Ld} + \frac{\sqrt{2}L}{\gamma}\|\boldsymbol{Z}_0 - \boldsymbol{Z}_0^*\|_{\mathbb{L}_2} \\
&= \sqrt{Ld} + \frac{\sqrt{2}L}{\gamma}\left\|\boldsymbol{z}_k^{(o)\widetilde{\nabla}} - \boldsymbol{z}_k^*\right\|_{\mathbb{L}_2} \\
&= \sqrt{Ld} + \frac{\sqrt{2}L}{\gamma}A_k
\end{aligned}
\tag{62}
$$

**Change**     Assume $t_2 \geq t_1 \geq 0$.

$$\begin{aligned}
\|\boldsymbol{X}_{t_1} - \boldsymbol{X}_{t_2}\|_{\mathbb{L}_2} &\leq \left\|\int_{t_1}^{t_2} \boldsymbol{V}_s ds\right\|_{\mathbb{L}_2} \\
&\leq \int_{t_1}^{t_2} \|\boldsymbol{V}_s\|_{\mathbb{L}_2} ds \\
&\leq (t_2 - t_1) \max_{t_1 \leq u \leq t_2} \|\boldsymbol{V}_u\|_{\mathbb{L}_2}
\end{aligned} \tag{63}$$

### B.5.3   Error for LPM

$$\begin{aligned}
\left\|\boldsymbol{V}_t^{(l)} - \boldsymbol{V}_t\right\|_{\mathbb{L}_2} &= \left\|\int_0^t \psi_0(t-s)(\nabla f(\boldsymbol{X}_s) - \nabla f(\boldsymbol{X}_0)) ds\right\|_{\mathbb{L}_2} \\
&\leq \int_0^t \|\nabla f(\boldsymbol{X}_s) - \nabla f(\boldsymbol{X}_0)\|_{\mathbb{L}_2} ds \\
&\leq L \int_0^t \|\boldsymbol{X}_s - \boldsymbol{X}_0\|_{\mathbb{L}_2} ds \\
&\leq L \int_0^t s \max_{0 \leq u \leq t} \|\boldsymbol{V}_u\|_{\mathbb{L}_2} ds \\
&\leq \frac{1}{2} L t^2 \max_{0 \leq u \leq t} \|\boldsymbol{V}_u\|_{\mathbb{L}_2}
\end{aligned} \tag{64}$$

$$\begin{aligned}
\left\|\boldsymbol{X}_t^{(l)} - \boldsymbol{X}_t\right\|_{\mathbb{L}_2} &= \left\|\int_0^t (\boldsymbol{V}_t^{(l)} - \boldsymbol{V}_t) ds\right\|_{\mathbb{L}_2} \\
&\leq \int_0^t \left\|\boldsymbol{V}_t^{(l)} - \boldsymbol{V}_t\right\|_{\mathbb{L}_2} ds \\
&\leq \frac{1}{2} L \int_0^t s^2 \max_{0 \leq u \leq s} \|\boldsymbol{V}_u\|_{\mathbb{L}_2} ds \\
&\leq \frac{1}{6} L t^3 \max_{0 \leq u \leq t} \|\boldsymbol{V}_u\|_{\mathbb{L}_2}
\end{aligned} \tag{65}$$

### B.5.4   Error for RMM

**First term - Variance**     We aim to control the variance $\mathbb{E}\left[\left\|\boldsymbol{Z}_t^{(r)} - \mathbb{E}_a \boldsymbol{Z}_t^{(r)}\right\|_2^2\right]$.

$$\begin{aligned}
\mathbb{E}\left[\left\|\boldsymbol{Z}_t^{(r)} - \mathbb{E}_a \boldsymbol{Z}_t^{(r)}\right\|_2^2\right] &\leq \mathbb{E}\left[\left\|\boldsymbol{Z}_t^{(r)} - \boldsymbol{Z}_t\right\|_2^2\right] \\
&= \mathbb{E}\left[\left\|\begin{bmatrix} \boldsymbol{R}_t^{(r)} \\ \boldsymbol{S}_t^{(r)} \end{bmatrix} - \begin{bmatrix} \boldsymbol{R}_t \\ \boldsymbol{S}_t \end{bmatrix}\right\|_2^2\right] \\
&= \mathbb{E}\left[\left\|\begin{bmatrix} \gamma & 1 \\ 0 & 1 \end{bmatrix}\begin{bmatrix} \boldsymbol{X}_t^{(r)} - \boldsymbol{X}_t \\ \boldsymbol{V}_t^{(r)} - \boldsymbol{V}_t \end{bmatrix}\right\|_2^2\right] \\
&\leq 2\gamma^2 \mathbb{E}\left[\left\|\boldsymbol{X}_t^{(r)} - \boldsymbol{X}_t\right\|_2^2\right] + 3\mathbb{E}\left[\left\|\boldsymbol{V}_t^{(r)} - \boldsymbol{V}_t\right\|_2^2\right] \\
&= 2\gamma^2 \left\|\boldsymbol{X}_t^{(r)} - \boldsymbol{X}_t\right\|_{\mathbb{L}_2}^2 + 3\left\|\boldsymbol{V}_t^{(r)} - \boldsymbol{V}_t\right\|_{\mathbb{L}_2}^2
\end{aligned} \tag{66}$$

$$\left\|\boldsymbol{X}_t^{(r)} - \boldsymbol{X}_t\right\|_{\mathbb{L}_2} \leq \left\|t\psi_1(t-at)\nabla f(\boldsymbol{X}_{at}^{(l)}) - t\psi_1(t-at)\nabla f(\boldsymbol{X}_{at})\right\|_{\mathbb{L}_2}$$

$$+ \left\|t\psi_1(t-at)\nabla f(\boldsymbol{X}_{at}) - \int_0^t \psi_1(t-at)\nabla f(\boldsymbol{X}_s)ds\right\|_{\mathbb{L}_2} \quad (67)$$

$$+ \left\|\int_0^t \psi_1(t-at)\nabla f(\boldsymbol{X}_s)ds - \int_0^t \psi_1(t-s)\nabla f(\boldsymbol{X}_s)ds\right\|_{\mathbb{L}_2}$$

We denote above three terms as $D_1, D_2$ and $D_3$.

$$D_1 = \left\|t\psi_1(t-at)\nabla f(\boldsymbol{X}_{at}^{(l)}) - t\psi_1(t-at)\nabla f(\boldsymbol{X}_{at})\right\|_{\mathbb{L}_2}$$

$$\leq \sqrt{\mathbb{E}_a\left[(t\psi_1(t-at)\|\nabla f(\boldsymbol{X}_{at}^{(l)}) - \nabla f(\boldsymbol{X}_{at})\|_{\mathbb{L}_2,B})^2\right]}$$

$$\leq \sqrt{\mathbb{E}_a\left[(t\psi_1(t-at)L\|\boldsymbol{X}_{at}^{(l)} - \boldsymbol{X}_{at}\|_{\mathbb{L}_2,B})^2\right]} \quad (68)$$

$$\leq \sqrt{\mathbb{E}_a\left[\left(t\psi_1(t-at)L(\frac{1}{6}L(at)^3 \max_{0\leq u\leq t}\|\boldsymbol{V}_u\|_{\mathbb{L}_2})\right)^2\right]}$$

$$\leq \frac{\sqrt{7}}{252}L^2 t^5 \max_{0\leq u\leq t}\|\boldsymbol{V}_u\|_{\mathbb{L}_2}$$

$$D_2 = \left\|t\psi_1(t-at)\nabla f(\boldsymbol{X}_{at}) - \int_0^t \psi_1(t-at)\nabla f(\boldsymbol{X}_s)ds\right\|_{\mathbb{L}_2}$$

$$= \sqrt{\mathbb{E}_a\left[\psi_1(t-at)^2\mathbb{E}_B\left(\int_0^t(\nabla f(\boldsymbol{X}_{at}) - \nabla f(\boldsymbol{X}_s))ds\right)^2\right]}$$

$$= \sqrt{\mathbb{E}_a\left[\psi_1(t-at)^2\left\|\int_0^t(\nabla f(\boldsymbol{X}_{at}) - \nabla f(\boldsymbol{X}_s))ds\right\|_{\mathbb{L}_2,B}^2\right]}$$

$$\leq \sqrt{\mathbb{E}_a\left[\psi_1(t-at)^2\left(\int_0^t\|(\nabla f(\boldsymbol{X}_{at}) - \nabla f(\boldsymbol{X}_s))\|_{\mathbb{L}_2,B}ds\right)^2\right]} \quad (69)$$

$$\leq \sqrt{\mathbb{E}_a\left[\psi_1(t-at)^2L^2\left(\int_0^t\|(\boldsymbol{X}_{at} - \boldsymbol{X}_s)\|_{\mathbb{L}_2,B}ds\right)^2\right]}$$

$$\leq \sqrt{\mathbb{E}_a\left[\psi_1(t-at)^2L^2\left(\int_0^t|at-s|\max_{0\leq u\leq t}\|\boldsymbol{V}_u\|_{\mathbb{L}_2}ds\right)^2\right]}$$

$$\leq \frac{\sqrt{210}}{70}Lt^3 \max_{0\leq u\leq t}\|\boldsymbol{V}_u\|_{\mathbb{L}_2}$$

$$D_3 = \left\| \int_0^t \psi_1(t-at)\nabla f(\boldsymbol{X}_s)ds - \int_0^t \psi_1(t-s)\nabla f(\boldsymbol{X}_s)ds \right\|_{\mathbb{L}_2}$$

$$= \sqrt{\mathbb{E}_a \left\| \int_0^t (\psi_1(t-at) - \psi_1(t-s))\nabla f(\boldsymbol{X}_s)ds \right\|_{\mathbb{L}_2,B}^2}$$

$$\leq \sqrt{\mathbb{E}_a \left( \int_0^t |\psi_1(t-at) - \psi_1(t-s)| \|\nabla f(\boldsymbol{X}_s)\|_{\mathbb{L}_2,B} ds \right)^2} \tag{70}$$

$$\leq \sqrt{\mathbb{E}_a \left( \int_0^t |\psi_1(t-at) - \psi_1(t-s)| ds \right)^2} \max_{0 \leq u \leq t} \|\nabla f(\boldsymbol{X}_u)\|_{\mathbb{L}_2}$$

$$\leq \frac{\sqrt{105}}{30} t^2 \max_{0 \leq u \leq t} \|\nabla f(\boldsymbol{X}_u)\|_{\mathbb{L}_2}$$

Therefore, we conclude the variance of $\boldsymbol{X}$ as following.

$$\left\| \boldsymbol{X}_t^{(r)} - \boldsymbol{X}_t \right\|_{\mathbb{L}_2} \leq \left( \frac{\sqrt{210}}{70} L t^3 + \frac{\sqrt{7}}{252} L^2 t^5 \right) \max_{0 \leq u \leq t} \|\boldsymbol{V}_u\|_{\mathbb{L}_2}$$

$$+ \frac{\sqrt{105}}{30} t^2 \max_{0 \leq u \leq t} \|\nabla f(\boldsymbol{X}_u)\|_{\mathbb{L}_2} \tag{71}$$

$$\left\| \boldsymbol{V}_t^{(r)} - \boldsymbol{V}_t \right\|_{\mathbb{L}_2} \leq \left\| t\psi_0(t-at)\nabla f(\boldsymbol{X}_{at}^{(l)}) - t\psi_0(t-at)\nabla f(\boldsymbol{X}_{at}) \right\|_{\mathbb{L}_2}$$

$$+ \left\| t\psi_0(t-at)\nabla f(\boldsymbol{X}_{at}) - \int_0^t \psi_0(t-at)\nabla f(\boldsymbol{X}_s)ds \right\|_{\mathbb{L}_2} \tag{72}$$

$$+ \left\| \int_0^t \psi_0(t-at)\nabla f(\boldsymbol{X}_s)ds - \int_0^t \psi_0(t-s)\nabla f(\boldsymbol{X}_s)ds \right\|_{\mathbb{L}_2}$$

We denote above three terms as $E_1, E_2$ and $E_3$.

$$E_1 = \left\| t\psi_0(t-at)\nabla f(\boldsymbol{X}_{at}^{(l)}) - t\psi_0(t-at)\nabla f(\boldsymbol{X}_{at}) \right\|_{\mathbb{L}_2}$$

$$\leq \sqrt{\mathbb{E}_a \left[ (t\psi_0(t-at)\|\nabla f(\boldsymbol{X}_{at}^{(l)}) - \nabla f(\boldsymbol{X}_{at})\|_{\mathbb{L}_2,B})^2 \right]}$$

$$\leq \sqrt{\mathbb{E}_a \left[ (t\psi_0(t-at)L\|\boldsymbol{X}_{at}^{(l)} - \boldsymbol{X}_{at}\|_{\mathbb{L}_2,B})^2 \right]} \tag{73}$$

$$\leq \sqrt{\mathbb{E}_a \left[ \left( t\psi_0(t-at)L(\frac{1}{6}L(at)^3 \max_{0 \leq u \leq t} \|\boldsymbol{V}_u\|_{\mathbb{L}_2}) \right)^2 \right]}$$

$$\leq \frac{\sqrt{7}}{42} L^2 t^4 \max_{0 \leq u \leq t} \|\boldsymbol{V}_u\|_{\mathbb{L}_2}$$

$$E_2 = \left\| t\psi_0(t-at)\nabla f(\boldsymbol{X}_{at}) - \int_0^t \psi_0(t-at)\nabla f(\boldsymbol{X}_s)ds \right\|_{\mathbb{L}_2}$$

$$= \sqrt{\mathbb{E}_a\left[\psi_0(t-at)^2 \mathbb{E}_B\left(\int_0^t (\nabla f(\boldsymbol{X}_{at}) - \nabla f(\boldsymbol{X}_s))ds\right)^2\right]}$$

$$= \sqrt{\mathbb{E}_a\left[\psi_0(t-at)^2 \left\| \int_0^t (\nabla f(\boldsymbol{X}_{at}) - \nabla f(\boldsymbol{X}_s))ds \right\|_{\mathbb{L}_2,B}^2\right]}$$

$$\leq \sqrt{\mathbb{E}_a\left[\psi_0(t-at)^2 \left(\int_0^t \|(\nabla f(\boldsymbol{X}_{at}) - \nabla f(\boldsymbol{X}_s))\|_{\mathbb{L}_2,B}ds\right)^2\right]} \tag{74}$$

$$\leq \sqrt{\mathbb{E}_a\left[\psi_0(t-at)^2 L^2 \left(\int_0^t \|(\boldsymbol{X}_{at} - \boldsymbol{X}_s)\|_{\mathbb{L}_2,B}ds\right)^2\right]}$$

$$\leq \sqrt{\mathbb{E}_a\left[\psi_0(t-at)^2 L^2 \left(\int_0^t |at - s| \max_{0\leq u\leq t}\|\boldsymbol{V}_u\|_{\mathbb{L}_2}ds\right)^2\right]}$$

$$\leq \frac{\sqrt{105}}{30}Lt^2 \max_{0\leq u\leq t}\|\boldsymbol{V}_u\|_{\mathbb{L}_2}$$

$$E_3 = \left\| \int_0^t \psi_0(t-at)\nabla f(\boldsymbol{X}_s)ds - \int_0^t \psi_0(t-s)\nabla f(\boldsymbol{X}_s)ds \right\|_{\mathbb{L}_2}$$

$$= \sqrt{\mathbb{E}_a\left\| \int_0^t (\psi_0(t-at) - \psi_0(t-s))\nabla f(\boldsymbol{X}_s)ds \right\|_{\mathbb{L}_2,B}^2}$$

$$\leq \sqrt{\mathbb{E}_a(\int_0^t |\psi_0(t-at) - \psi_0(t-s)|\|\nabla f(\boldsymbol{X}_s)\|_{\mathbb{L}_2,B}ds)^2} \tag{75}$$

$$\leq \sqrt{\mathbb{E}_a(\int_0^t |\psi_0(t-at) - \psi_0(t-s)|ds)^2} \max_{0\leq u\leq t}\|\nabla f(\boldsymbol{X}_u)\|_{\mathbb{L}_2}$$

$$\leq \frac{\sqrt{105}}{30}\gamma t^2 \max_{0\leq u\leq t}\|\nabla f(\boldsymbol{X}_u)\|_{\mathbb{L}_2}$$

Therefore, we conclude the variance of $\boldsymbol{X}$ as following.

$$\left\|\boldsymbol{V}_t^{(r)} - \boldsymbol{V}_t\right\|_{\mathbb{L}_2} \leq (\frac{\sqrt{105}}{30}Lt^2 + \frac{\sqrt{7}}{42}L^2t^4) \max_{0\leq u\leq t}\|\boldsymbol{V}_u\|_{\mathbb{L}_2}$$
$$+ \frac{\sqrt{105}}{30}\gamma t^2 \max_{0\leq u\leq t}\|\nabla f(\boldsymbol{X}_u)\|_{\mathbb{L}_2} \tag{76}$$

Finally, we can give the upper bound of variance as follows.

$$\mathbb{E}\left[\left\|\boldsymbol{Z}_t^{(r)} - \mathbb{E}_a\boldsymbol{Z}_t^{(r)}\right\|_2^2\right] \leq \frac{7}{6}\gamma^2 t^4 \max_{0\leq u\leq t}\|\nabla f(\boldsymbol{X}_u)\|_{\mathbb{L}_2}^2 + \frac{7}{20}t^4 \max_{0\leq u\leq t}\|\boldsymbol{V}_u\|_{\mathbb{L}_2}^2$$
$$+ RL^2t^6 \max_{0\leq u\leq t}\|\boldsymbol{V}_u\|_{\mathbb{L}_2}^2 \tag{77}$$

$$R = \frac{\left(5\gamma^2 L^2t^4 + 36\,L\sqrt{2}\gamma^2\sqrt{15}t^2 + 270L^2t^2 + 756\,L\sqrt{15} + 1944\,\gamma^2\right)}{22680}$$

**Second term - Bias** We aim to control the bias $\left\|\mathbb{E}_a\boldsymbol{Z}_t^{(r)} - \boldsymbol{Z}_t\right\|_2$.

$$\left\|\mathbb{E}_a\boldsymbol{Z}_t^{(r)} - \boldsymbol{Z}_t\right\|_{\mathbb{L}_2} \leq \gamma\left\|\mathbb{E}_a\boldsymbol{X}_t^{(r)} - \boldsymbol{X}_t\right\|_{\mathbb{L}_2} + 2\left\|\mathbb{E}_a\boldsymbol{V}_t^{(r)} - \boldsymbol{V}_t\right\|_{\mathbb{L}_2} \tag{78}$$

$$\left\|\mathbb{E}_a\boldsymbol{X}_t^{(r)} - \boldsymbol{X}_t\right\|_{\mathbb{L}_2} = \left\|\int_0^t \psi_1(t-s)\nabla f(\boldsymbol{X}_s^{(l)})ds - \int_0^t \psi_1(t-s)\nabla f(\boldsymbol{X}_s)ds\right\|_{\mathbb{L}_2}$$

$$= \left\|\int_0^t \psi_1(t-s)(\nabla f(\boldsymbol{X}_s^{(l)}) - \nabla f(\boldsymbol{X}_s))ds\right\|_{\mathbb{L}_2}$$

$$\leq \int_0^t \psi_1(t-s)\left\|\nabla f(\boldsymbol{X}_s^{(l)}) - \nabla f(\boldsymbol{X}_s)\right\|_{\mathbb{L}_2}ds$$

$$\leq \int_0^t \psi_1(t-s)L\left\|\boldsymbol{X}_s^{(l)} - \boldsymbol{X}_s\right\|_{\mathbb{L}_2}ds \tag{79}$$

$$\leq \int_0^t \psi_1(t-s)L(\frac{1}{6}Ls^3 \max_{0\leq u\leq t}\|\boldsymbol{V}_u\|_{\mathbb{L}_2})ds$$

$$\leq \frac{1}{120}L^2t^5 \max_{0\leq u\leq t}\|\boldsymbol{V}_u\|_{\mathbb{L}_2}$$

$$\left\|\mathbb{E}_a\boldsymbol{V}_t^{(r)} - \boldsymbol{V}_t\right\|_{\mathbb{L}_2} = \left\|\int_0^t \psi_0(t-s)\nabla f(\boldsymbol{X}_s^{(l)})ds - \int_0^t \psi_0(t-s)\nabla f(\boldsymbol{X}_s)ds\right\|_{\mathbb{L}_2}$$

$$= \left\|\int_0^t \psi_0(t-s)(\nabla f(\boldsymbol{X}_s^{(l)}) - \nabla f(\boldsymbol{X}_s))ds\right\|_{\mathbb{L}_2}$$

$$\leq \int_0^t \psi_0(t-s)\left\|\nabla f(\boldsymbol{X}_s^{(l)}) - \nabla f(\boldsymbol{X}_s)\right\|_{\mathbb{L}_2}ds$$

$$\leq \int_0^t \psi_0(t-s)L\left\|\boldsymbol{X}_s^{(l)} - \boldsymbol{X}_s\right\|_{\mathbb{L}_2}ds \tag{80}$$

$$\leq \int_0^t \psi_0(t-s)L(\frac{1}{6}Ls^3 \max_{0\leq u\leq t}\|\boldsymbol{V}_u\|_{\mathbb{L}_2})ds$$

$$\leq \frac{1}{24}L^2t^4 \max_{0\leq u\leq t}\|\boldsymbol{V}_u\|_{\mathbb{L}_2}$$

$$\left\|\mathbb{E}_a\boldsymbol{Z}_t^{(r)} - \boldsymbol{Z}_t\right\|_{\mathbb{L}_2} \leq \frac{1}{12}L^2t^4(1 + \frac{1}{10}\gamma Lt) \max_{0\leq u\leq t}\|\boldsymbol{V}_u\|_{\mathbb{L}_2} \tag{81}$$

### B.5.5 Error for ALUM

**Variance**

$$
\begin{aligned}
&\mathbb{E}\left[\left\|\boldsymbol{Z}_t^{(o)} - \boldsymbol{Z}_t^{(r)}\right\|_2^2\right]\\
=&\mathbb{E}\left[\left\|\begin{bmatrix}\gamma & 1\\ 0 & 1\end{bmatrix}\begin{bmatrix}\boldsymbol{X}_t^{(o)} - \boldsymbol{X}_t^{(r)}\\ \boldsymbol{V}_t^{(o)} - \boldsymbol{V}_t^{(r)}\end{bmatrix}\right\|_2^2\right]\\
\leq&\mathbb{E}\left[\left\|\begin{bmatrix}\gamma & 1\\ 0 & 1\end{bmatrix}\begin{bmatrix}t\psi_1(t-at)(\nabla f(\boldsymbol{X}_{at}^{(e)\widetilde{\nabla}}) - \nabla f(\boldsymbol{X}_{at}^{(l)}))\\ t\psi_0(t-at)(\nabla f(\boldsymbol{X}_{at}^{(e)\widetilde{\nabla}}) - \nabla f(\boldsymbol{X}_{at}^{(l)}))\end{bmatrix}\right\|_2^2\right]\\
\leq&\mathbb{E}\left[t^2((\gamma\psi_1(t-at) + \psi_0(t-at))^2 + \psi_0(t-at)^2)\left\|\nabla f(\boldsymbol{X}_{at}^{(e)\widetilde{\nabla}}) - \nabla f(\boldsymbol{X}_{at}^{(l)})\right\|_2^2\right]\\
=&\mathbb{E}_a\left[t^2((\gamma\psi_1(t-at) + \psi_0(t-at))^2 + \psi_0(t-at)^2)\left\|\nabla f(\boldsymbol{X}_{at}^{(e)\widetilde{\nabla}}) - \nabla f(\boldsymbol{X}_{at}^{(l)})\right\|_{\mathbb{L}_2,B}^2\right]\\
\leq&\mathbb{E}_a\left[t^2((\gamma\psi_1(t-at) + \psi_0(t-at))^2 + \psi_0(t-at)^2)L^2\left\|\boldsymbol{X}_{at}^{(e)\widetilde{\nabla}} - \boldsymbol{X}_{at}^{(l)}\right\|_{\mathbb{L}_2,B}^2\right]\\
=&\mathbb{E}_a\left[t^2((\gamma\psi_1(t-at) + \psi_0(t-at))^2 + \psi_0(t-at)^2)L^2\|\psi_2(at)\nabla f(\boldsymbol{X}_0)\|_{\mathbb{L}_2,B}^2\right]\\
\leq&\mathbb{E}_a\left[t^2\psi_2(at)^2((\gamma\psi_1(t-at) + \psi_0(t-at))^2 + \psi_0(t-at)^2)\right]L^2 \max_{0\leq u\leq t}\|\nabla f(\boldsymbol{X}_t)\|_{\mathbb{L}_2}^2\\
\leq&\frac{1}{10}L^2t^6 \max_{0\leq u\leq t}\|\nabla f(\boldsymbol{X}_t)\|_{\mathbb{L}_2}^2
\end{aligned}
\tag{82}
$$

$$
\begin{aligned}
&\mathbb{E}\left[\left\|\boldsymbol{X}_t^{(o)} - \boldsymbol{X}_t^{(r)}\right\|_2^2\right]\\
\leq&\mathbb{E}_a\left[t^2\psi_1(t-at)^2\psi_2(at)^2\right]L^2 \max_{0\leq u\leq t}\|\nabla f(\boldsymbol{X}_t)\|_{\mathbb{L}_2}^2\\
\leq&\frac{1}{420}L^2t^8 \max_{0\leq u\leq t}\|\nabla f(\boldsymbol{X}_t)\|_{\mathbb{L}_2}^2
\end{aligned}
\tag{83}
$$

$$
\begin{aligned}
&\mathbb{E}\left[\left\|\boldsymbol{V}_t^{(o)} - \boldsymbol{V}_t^{(r)}\right\|_2^2\right]\\
\leq&\mathbb{E}_a\left[t^2\psi_0(t-at)^2\psi_2(at)^2\right]L^2 \max_{0\leq u\leq t}\|\nabla f(\boldsymbol{X}_t)\|_{\mathbb{L}_2}^2\\
\leq&\frac{1}{20}L^2t^6 \max_{0\leq u\leq t}\|\nabla f(\boldsymbol{X}_t)\|_{\mathbb{L}_2}^2
\end{aligned}
\tag{84}
$$

**Bias**

$$
\begin{aligned}
&\left\|\mathbb{E}_a\boldsymbol{Z}_t^{(o)} - \mathbb{E}_a\boldsymbol{Z}_t^{(r)}\right\|_{\mathbb{L}_2}\\
=&\sqrt{\mathbb{E}\left\|\mathbb{E}_a\left[\boldsymbol{Z}_t^{(o)} - \boldsymbol{Z}_t^{(r)}\right]\right\|_2^2}\\
\leq&\sqrt{\mathbb{E}\left\|\boldsymbol{Z}_t^{(o)} - \boldsymbol{Z}_t^{(r)}\right\|_2^2}\\
\leq&\frac{1}{\sqrt{10}}Lt^3 \max_{0\leq u\leq t}\|\nabla f(\boldsymbol{X}_t)\|_{\mathbb{L}_2}
\end{aligned}
\tag{85}
$$

### B.5.6 Error introduced by gradient estimation

$$\mathbb{E}\left[\left\|\boldsymbol{Z}_t^{(o)\widetilde{\nabla}} - \boldsymbol{Z}_t^{(o)}\right\|_2^2\right]$$

$$=\mathbb{E}\left[\left\|\begin{bmatrix}\gamma & 1\\ 0 & 1\end{bmatrix}\begin{bmatrix}\boldsymbol{X}_t^{(o)\widetilde{\nabla}} - \boldsymbol{X}_t^{(o)}\\ \boldsymbol{V}_t^{(o)\widetilde{\nabla}} - \boldsymbol{V}_t^{(o)}\end{bmatrix}\right\|_2^2\right]$$

$$=\mathbb{E}\left[\left\|\begin{bmatrix}\gamma & 1\\ 0 & 1\end{bmatrix}\begin{bmatrix}t\psi_1(t-at)(\widetilde{\nabla}_k - \nabla f(\boldsymbol{X}_{at}^{(e)\widetilde{\nabla}}))\\ t\psi_0(t-at)(\widetilde{\nabla}_k - \nabla f(\boldsymbol{X}_{at}^{(e)\widetilde{\nabla}}))\end{bmatrix}\right\|_2^2\right] \tag{86}$$

$$=\mathbb{E}\left[t^2((\gamma\psi_1(t-at) + \psi_0(t-at))^2 + \psi_0(t-at)^2)\left\|\widetilde{\nabla}_k - \mathbb{E}_k\widetilde{\nabla}_k\right\|_2^2\right]$$

$$\leq\mathbb{E}\left[t^2((\gamma t+1)^2 + 1^2)\left\|\widetilde{\nabla}_k - \mathbb{E}_k\widetilde{\nabla}_k\right\|_2^2\right]$$

$$=t^2((\gamma t+1)^2 + 1^2)\mathbb{E}\left[\left\|\widetilde{\nabla}_k - \mathbb{E}_k\widetilde{\nabla}_k\right\|_2^2\right]$$

$$\leq t^2((\gamma t+1)^2 + 1)\Theta\max_{i<k}Q_i$$

The last inequality follows definition 1. We use $Q_k = N\sum_{i=1}^{N}\mathbb{E}[\|\nabla f_i(\boldsymbol{x}_{k+1}^{(e)\widetilde{\nabla}}) - \nabla f_i(\boldsymbol{x}_k^{(e)\widetilde{\nabla}})\|_2^2]$. This has an extra expectation compared to $Q_k$ in definition 1 because we need to consider the extra randomness coming from the Brownian motion instead of just random batch.

$$\mathbb{E}\left[\left\|\boldsymbol{X}_t^{(o)\widetilde{\nabla}} - \boldsymbol{X}_t^{(o)}\right\|_2^2\right] \leq\mathbb{E}\left[\left\|t\psi_1(t-at)(\widetilde{\nabla}_k - \nabla f(\boldsymbol{X}_{at}^{(e)\widetilde{\nabla}}))\right\|_2^2\right] \tag{87}$$
$$\leq t^4\Theta\max_{i<k}Q_i$$

$$\mathbb{E}\left[\left\|\boldsymbol{V}_t^{(o)\widetilde{\nabla}} - \boldsymbol{V}_t^{(o)}\right\|_2^2\right] \leq\mathbb{E}\left[\left\|t\psi_0(t-at)(\widetilde{\nabla}_k - \nabla f(\boldsymbol{X}_{at}^{(e)\widetilde{\nabla}}))\right\|_2^2\right] \tag{88}$$
$$\leq t^2\Theta\max_{i<k}Q_i$$

### B.5.7 Summary of error in single step

By combining above error terms, we get follow error change in single step.

$$\mathbb{E}\left[\langle\boldsymbol{Z}_t^{(o)\widetilde{\nabla}} - \mathbb{E}_a\mathbb{E}_k\boldsymbol{Z}_t^{(o)\widetilde{\nabla}}, \mathbb{E}_a\mathbb{E}_k\boldsymbol{Z}_t^{(o)\widetilde{\nabla}} - \boldsymbol{Z}_t^*\rangle\right]$$

$$=\mathbb{E}_B\left[\langle\mathbb{E}_a\mathbb{E}_k\boldsymbol{Z}_t^{(o)\widetilde{\nabla}} - \mathbb{E}_a\mathbb{E}_k\boldsymbol{Z}_t^{(o)\widetilde{\nabla}}, \mathbb{E}_a\mathbb{E}_k\boldsymbol{Z}_t^{(o)\widetilde{\nabla}} - \boldsymbol{Z}_t^*\rangle\right] \tag{89}$$

$$=0$$

$$\mathbb{E}\left[\left\|\boldsymbol{Z}_t^{(o)\widetilde{\nabla}} - \boldsymbol{Z}_t^*\right\|_2^2\right] = \mathbb{E}\left[\left\|\boldsymbol{Z}_t^{(o)\widetilde{\nabla}} - \mathbb{E}_a\mathbb{E}_k\boldsymbol{Z}_t^{(o)\widetilde{\nabla}}\right\|_2^2\right]$$
$$+ \mathbb{E}\left[\left\|\mathbb{E}_a\mathbb{E}_k\boldsymbol{Z}_t^{(o)\widetilde{\nabla}} - \boldsymbol{Z}_t^*\right\|_2^2\right]$$
$$\leq \mathbb{E}\left[\left\|\boldsymbol{Z}_t^{(o)\widetilde{\nabla}} - \mathbb{E}_a\boldsymbol{Z}_t^{(r)}\right\|_2^2\right]$$
$$+ \mathbb{E}\left[\left\|\mathbb{E}_a\boldsymbol{Z}_t^{(o)} - \boldsymbol{Z}_t^*\right\|_2^2\right]$$
$$\leq 3\mathbb{E}\left[\left\|\boldsymbol{Z}_t^{(o)\widetilde{\nabla}} - \boldsymbol{Z}_t^{(o)}\right\|_2^2\right] + 3\mathbb{E}\left[\left\|\boldsymbol{Z}_t^{(o)} - \boldsymbol{Z}_t^{(r)}\right\|_2^2\right]$$
$$+ 3\mathbb{E}\left[\left\|\boldsymbol{Z}_t^{(r)} - \mathbb{E}_a\boldsymbol{Z}_t^{(r)}\right\|_2^2\right]$$
$$+ \left(\left\|\mathbb{E}_a\boldsymbol{Z}_t^{(o)} - \mathbb{E}_a\boldsymbol{Z}_t^{(r)}\right\|_{\mathbb{L}_2} + \left\|\mathbb{E}_a\boldsymbol{Z}_t^{(r)} - \boldsymbol{Z}_t\right\|_{\mathbb{L}_2} + \left\|\boldsymbol{Z}_t - \boldsymbol{Z}_t^*\right\|_{\mathbb{L}_2}\right)^2$$
$$\leq 3t^2((\gamma t + 1)^2 + 1)\Theta \max_{i<k} Q_i$$
$$+ \frac{7}{2}\gamma^2 t^4 \max_{0 \leq u \leq t}\|\nabla f(\boldsymbol{X}_u)\|_{\mathbb{L}_2}^2 + \frac{21}{20}t^4 \max_{0 \leq u \leq t}\|\boldsymbol{V}_u\|_{\mathbb{L}_2}^2$$
$$+ L^2 t^6 \left(\frac{3}{10} \max_{0 \leq u \leq t}\|\nabla f(\boldsymbol{X}_t)\|_{\mathbb{L}_2}^2 + 3R \max_{0 \leq u \leq t}\|\boldsymbol{V}_u\|_{\mathbb{L}_2}^2\right)$$
$$+ \left(\frac{1}{\sqrt{10}}Lt^3 \max_{0 \leq u \leq t}\|\nabla f(\boldsymbol{X}_t)\|_{\mathbb{L}_2}\right.$$
$$+ \frac{1}{12}L^2 t^4 (1 + \frac{1}{10}\gamma Lt) \max_{0 \leq u \leq t}\|\boldsymbol{V}_u\|_{\mathbb{L}_2}$$
$$\left. + e^{-\frac{m}{\gamma}t}\|\boldsymbol{Z}_0 - \boldsymbol{Z}_0^*\|_{\mathbb{L}_2}\right)^2$$
$$\tag{90}$$

$$R = \frac{\left(5\gamma^2 L^2 t^4 + 36\,L\sqrt{2}\gamma^2\sqrt{15}t^2 + 270L^2t^2 + 756\,L\sqrt{15} + 1944\,\gamma^2\right)}{22680}$$

### B.5.8 Control $Q_k$ with $A_k$

From here, we assume that a fixed step size $h$ is used. In this section we introduce temporary notation $a_k$ and $\boldsymbol{e}_{x,[0,a_k h],k}$, where $k$ is inserted to specify the iteration.

We know that at each iteration, we calculate gradient estimation on point:

$$\boldsymbol{x}_k^{(e)\widetilde{\nabla}} = \boldsymbol{x}_k^{(o)\widetilde{\nabla}} + \psi_1(a_k h)\boldsymbol{v}_k^{(o)\widetilde{\nabla}} + \boldsymbol{e}_{x,[0,a_k h],k}.$$

Recall that value $Q_k$ is introduced in Appendices B.3 and B.5.6 and is used for controlling gradient estimation error. We next give the upper bound of value $Q_k$.

$$Q_k = N \sum_{i=1}^{N} \mathbb{E}[\|\nabla f_i(\boldsymbol{x}_{k+1}^{(e)\widetilde{\nabla}}) - \nabla f_i(\boldsymbol{x}_k^{(e)\widetilde{\nabla}})\|_2^2]$$
$$\leq N \sum_{i=1}^{N} (\frac{L}{N})^2 \mathbb{E}[\|\boldsymbol{x}_{k+1}^{(e)\widetilde{\nabla}} - \boldsymbol{x}_k^{(e)\widetilde{\nabla}}\|_2^2] \tag{91}$$
$$= L^2 \|\boldsymbol{x}_{k+1}^{(e)\widetilde{\nabla}} - \boldsymbol{x}_k^{(e)\widetilde{\nabla}}\|_{\mathbb{L}_2}^2$$

$\boldsymbol{X}_h, \boldsymbol{V}_h$ starts from $\boldsymbol{x}_k^{(o)\widetilde{\nabla}}, \boldsymbol{v}_k^{(o)\widetilde{\nabla}}$.

$$
\begin{aligned}
&\|\boldsymbol{x}_{k+1}^{(e)\widetilde{\nabla}} - \boldsymbol{x}_k^{(e)\widetilde{\nabla}}\|_{\mathbb{L}_2} \\
=&\|(\boldsymbol{x}_{k+1}^{(o)\widetilde{\nabla}} - \boldsymbol{x}_k^{(o)\widetilde{\nabla}}) + \psi_1(a_{k+1}h)\boldsymbol{v}_{k+1}^{(o)\widetilde{\nabla}} - \psi_1(a_k h)\boldsymbol{v}_k^{(o)\widetilde{\nabla}} + \boldsymbol{e}_{x,[0,a_{k+1}h],k+1} - \boldsymbol{e}_{x,[0,a_k h],k}\|_{\mathbb{L}_2} \\
\leq&\|\boldsymbol{x}_{k+1}^{(o)\widetilde{\nabla}} - \boldsymbol{x}_k^{(o)\widetilde{\nabla}}\|_{\mathbb{L}_2} + \|\psi_1(a_{k+1}h)\boldsymbol{v}_{k+1}^{(o)\widetilde{\nabla}}\|_{\mathbb{L}_2} + \|\psi_1(a_k h)\boldsymbol{v}_k^{(o)\widetilde{\nabla}}\|_{\mathbb{L}_2} \\
&+ \|\boldsymbol{e}_{x,[0,a_{k+1}h],k+1}\|_{\mathbb{L}_2} + \|\boldsymbol{e}_{x,[0,a_k h],k}\|_{\mathbb{L}_2}
\end{aligned}
\tag{92}
$$

$$
\begin{aligned}
&\|\boldsymbol{x}_{k+1}^{(o)\widetilde{\nabla}} - \boldsymbol{x}_k^{(o)\widetilde{\nabla}}\|_{\mathbb{L}_2} \\
=&\|\boldsymbol{X}_h^{(o)\widetilde{\nabla}} - \boldsymbol{X}_0\|_{\mathbb{L}_2} \\
\leq&\|\boldsymbol{X}_h^{(o)\widetilde{\nabla}} - \boldsymbol{X}_h^{(o)}\|_{\mathbb{L}_2} + \|\boldsymbol{X}_h^{(o)} - \boldsymbol{X}_h^{(r)}\|_{\mathbb{L}_2} + \|\boldsymbol{X}_h^{(r)} - \boldsymbol{X}_h\|_{\mathbb{L}_2} + \|\boldsymbol{X}_h - \boldsymbol{X}_0\|_{\mathbb{L}_2} \\
\leq& h^2\sqrt{\Theta}\max_{i<k}\sqrt{Q_i} + h\max_{0\leq u\leq h}\|\boldsymbol{V}_u\|_{\mathbb{L}_2} \\
&+ (\frac{\sqrt{210}}{70}Lh^3 + \frac{\sqrt{7}}{252}L^2h^5)\max_{0\leq u\leq t}\|\boldsymbol{V}_u\|_{\mathbb{L}_2} \\
&+ (\frac{\sqrt{105}}{30}h^2 + \frac{1}{\sqrt{420}}Lh^4)\max_{0\leq u\leq t}\|\nabla f(\boldsymbol{X}_u)\|_{\mathbb{L}_2}
\end{aligned}
\tag{93}
$$

$$
\|\psi_1(a_{k+1}h)\boldsymbol{v}_{k+1}^{(o)\widetilde{\nabla}}\|_{\mathbb{L}_2} \leq h\|\boldsymbol{v}_{k+1}^{(o)\widetilde{\nabla}}\|_{\mathbb{L}_2} = h\|\boldsymbol{V}_h^{(o)\widetilde{\nabla}}\|_{\mathbb{L}_2}
\tag{94}
$$

$$
\begin{aligned}
&\|\boldsymbol{V}_h^{(o)\widetilde{\nabla}}\|_{\mathbb{L}_2} \\
\leq&\|\boldsymbol{V}_h^{(o)\widetilde{\nabla}} - \boldsymbol{V}_h^{(o)}\|_{\mathbb{L}_2} + \|\boldsymbol{V}_h^{(o)} - \boldsymbol{V}_h^{(r)}\|_{\mathbb{L}_2} + \|\boldsymbol{V}_h^{(r)} - \boldsymbol{V}_h\|_{\mathbb{L}_2} + \|\boldsymbol{V}_h\|_{\mathbb{L}_2} \\
\leq& h\sqrt{\Theta}\max_{i<k}\sqrt{Q_i} + \max_{0\leq u\leq h}\|\boldsymbol{V}_u\|_{\mathbb{L}_2} \\
&+ (\frac{\sqrt{105}}{30}Lh^2 + \frac{\sqrt{7}}{42}L^2h^4)\max_{0\leq u\leq t}\|\boldsymbol{V}_u\|_{\mathbb{L}_2} \\
&+ (\frac{\sqrt{105}}{30}\gamma h^2 + \frac{1}{\sqrt{20}}Lh^3)\max_{0\leq u\leq t}\|\nabla f(\boldsymbol{X}_u)\|_{\mathbb{L}_2}
\end{aligned}
\tag{95}
$$

$$
\|\psi_1(a_k h)\boldsymbol{v}_k^{(o)\widetilde{\nabla}}\|_{\mathbb{L}_2} \leq h\|\boldsymbol{v}_k^{(o)\widetilde{\nabla}}\|_{\mathbb{L}_2} = h\|\boldsymbol{V}_0\|_{\mathbb{L}_2} \leq h\max_{0\leq u\leq t}\|\boldsymbol{V}_u\|_{\mathbb{L}_2}
\tag{96}
$$

$$
\begin{aligned}
&\|\boldsymbol{e}_{x,[0,a_{k+1}h],k+1}\|_{\mathbb{L}_2} = \|\boldsymbol{e}_{x,[0,a_k h],k}\|_{\mathbb{L}_2} \\
=&\sqrt{\mathbb{E}\big[\|\boldsymbol{e}_{x,[0,a_k h],k}\|_2^2\big]} \\
=&\sqrt{\mathbb{E}_a\left[(2\gamma d\int_0^{a_k h}\psi_1(u)^2 du)^2\right]} \\
\leq&\frac{1}{\sqrt{6}}\sqrt{\gamma d h^3}
\end{aligned}
\tag{97}
$$

$$
\begin{aligned}
\sqrt{Q_k} \leq& 2Lh^2\sqrt{\Theta}\max_{i<k}\sqrt{Q_i} + 3Lh\max_{0\leq u\leq t}\|\boldsymbol{V}_u\|_{\mathbb{L}_2} + \frac{2}{\sqrt{6}}\sqrt{\gamma d h^3} \\
&+ ((\frac{\sqrt{210}}{70} + \frac{\sqrt{105}}{30})L^2h^3 + (\frac{\sqrt{7}}{252} + \frac{\sqrt{7}}{40})L^3h^5)\max_{0\leq u\leq t}\|\boldsymbol{V}_u\|_{\mathbb{L}_2} \\
&+ (\frac{\sqrt{105}}{30}(1+\gamma h)Lh^2 + (\frac{1}{\sqrt{420}} + \frac{1}{\sqrt{20}})L^2h^4)\max_{0\leq u\leq t}\|\nabla f(\boldsymbol{X}_u)\|_{\mathbb{L}_2}
\end{aligned}
\tag{98}
$$

### B.5.9 Upper bound of $Q_k$ and $A_k$

Starting from this section, we assume $L = 1$ and $\gamma = 2$. We further assume $h < \frac{1}{10}$ to simplify upper bounds in previous sections.

$$\sqrt{Q_k} \leq 2h^2\sqrt{\Theta}\max_{i<k}\sqrt{Q_i} + 4hA_k + 4h\sqrt{d} \tag{99}$$

In order to give a tractable upper bound of $A_{k+1}$, we further apply Young's inequalities to $\|\mathbf{V}_t\|_{\mathbb{L}_2}^2$ and $\|\nabla f(\mathbf{X}_u)\|_{\mathbb{L}_2}^2$.

$$\|\mathbf{V}_t\|_{\mathbb{L}_2}^2 \leq 2d + 2A_k^2 \tag{100}$$

$$\|\nabla f(\mathbf{X}_u)\|_{\mathbb{L}_2}^2 \leq 2Ld + 4\frac{L^2}{\gamma^2}A_k^2 \tag{101}$$

$$\begin{aligned}
A_{k+1}^2 &= \mathbb{E}\left[\left\|\mathbf{Z}_t^{(o)\widetilde{\nabla}} - \mathbf{Z}_t^*\right\|_2^2\right] \\
&\leq 8h^2\Theta\max_{i<k}Q_i + 17h^4A_k^2 + 31h^4d \\
&\quad + \left(h^3A_k + h^3\sqrt{d} + e^{-\frac{m}{\gamma}t}A_k\right)^2 \\
&\leq 8h^2\Theta\max_{i<k}Q_i + 31h^4d \\
&\quad + \left(\sqrt{17}h^2A_k + h^3A_k + h^3\sqrt{d} + e^{-\frac{m}{\gamma}h}A_k\right)^2 \\
&\leq 8h^2\Theta\max_{i<k}Q_i + 31h^4d \\
&\quad + \left(5h^2A_k + h^3\sqrt{d} + e^{-\frac{m}{\gamma}h}A_k\right)^2
\end{aligned} \tag{102}$$

We further assume $h \leq \frac{1}{11}\frac{m}{\gamma}$, which implies $h \leq \frac{\frac{m}{\gamma}}{(\frac{m}{\gamma})^2+10}$ and $h \leq \frac{1}{22} \leq \frac{1}{10}$.

$$e^{-\frac{mh}{\gamma}} + 5h^2 = 1 - \frac{mh}{2\gamma} + h(-\frac{m}{2\gamma} + (\frac{m^2}{2\gamma^2} + 5)h) \leq 1 - \frac{mh}{2\gamma} \tag{103}$$

By [35, Lemma 7], if $x_{k+1}^2 \leq ((1-\alpha)x_k + B)^2 + A$, then

$$x_k \leq (1-\alpha)^k x_0 + \frac{B}{\alpha} + \frac{A}{B + \sqrt{\alpha(2-\alpha)A}} \leq (1-\alpha)^k x_0 + \frac{B}{\alpha} + \frac{\sqrt{A}}{\sqrt{\alpha}} \tag{104}$$

Therefore, we have

$$\begin{aligned}
A_k &= (1 - \frac{mh}{2\gamma})^k A_0 + \frac{2\gamma\sqrt{h^4d}}{m} + \sqrt{\frac{16h\gamma}{m}\Theta\max_{i<k}Q_i + \frac{62\gamma h^4 d}{m}} \\
&\leq (1 - \frac{mh}{2\gamma})^k A_0 + 4\sqrt{\frac{h\gamma}{m}}\sqrt{\Theta}\max_{i<k}\sqrt{Q_i} + (\frac{4\sqrt{h}}{\sqrt{m}} + \sqrt{62\gamma})\sqrt{\frac{h^3 d}{m}} \\
&\leq (1 - \frac{mh}{2\gamma})^k A_0 + 4\sqrt{\frac{h\gamma}{m}}\sqrt{\Theta}\max_{i<k}\sqrt{Q_i} + 12\sqrt{\frac{h^3 d}{m}}
\end{aligned} \tag{105}$$

We can then control $\max_{i<k}\sqrt{Q_i}$ as follows.

$$\begin{aligned}
\sqrt{Q_k} &\leq 24\sqrt{\frac{h^3}{m}}\sqrt{\Theta}\max_{i<k}\sqrt{Q_i} + 5h\sqrt{d} + 4h(1 - \frac{mh}{2\gamma})^k A_0 \\
&\leq 24\sqrt{\frac{h^3}{m}}\sqrt{\Theta}\max_{i<k}\sqrt{Q_i} + 5h\sqrt{d} + 4hA_0
\end{aligned} \tag{106}$$

Assume $\frac{h^3\Theta}{m} \leq \frac{1}{2304}$.

$$\sqrt{Q_k} \leq \frac{1}{2}\max_{i<k}\sqrt{Q_i} + 5h\sqrt{d} + 4hA_0. \tag{107}$$

$$\max_{i<k} \sqrt{Q_i} \leq 10h\sqrt{d} + 8hA_0 \tag{108}$$

Finally we can give upper bound of $A_k$.

$$A_k \leq (1 - \frac{mh}{2\gamma})^k A_0 + 46\sqrt{\frac{h^3}{m}}\sqrt{\Theta}A_0 + (12 + 57\sqrt{\Theta})\sqrt{\frac{h^3}{m}}\sqrt{d} \tag{109}$$
$$\leq 2A_0 + 2\sqrt{d}$$

## B.6 Discretization error

Consider a ULD process $\boldsymbol{Z}_t$ starting from the same initial point as the algorithm. Let $\boldsymbol{z}_k = \boldsymbol{Z}_{kh}$ and $B_k = \left\|\boldsymbol{z}_k^{(e)\widetilde{\nabla}} - \boldsymbol{z}_k\right\|_{\mathbb{L}_2}$. Based on (50), we have $B_k \geq \|\boldsymbol{x}_k^{(o)\widetilde{\nabla}} - \boldsymbol{X}_{kh}\|_{\mathbb{L}_2}$.

We can control the error $B_k$ similarly to what we have done for $A_k$. Actually we don't need to analyze the error from scratch again. The following result can be obtained directly by modifying (102). This is because the error in variance and bias terms is the same, and the only difference is that we are controlling difference between $\boldsymbol{z}_k^{(e)\widetilde{\nabla}}$ and $\boldsymbol{z}_k$ instead of between $\boldsymbol{z}_k^{(e)\widetilde{\nabla}}$ and $\boldsymbol{z}_k^*$.

$$B_{k+1}^2 \leq 8h^2\Theta\max_{i<k} Q_i + 17h^4 A_k^2 + 31h^4 d \tag{110}$$
$$+ \left(h^3 A_k + h^3\sqrt{d} + e^{-\frac{m}{\gamma}h}B_k\right)^2.$$

We use simplified upper bounds for $A_k$ and $Q_k$ in eqs. (108) and (109):

$$A_k \leq 2A_0 + 2\sqrt{d}$$
$$\max_{i<k} \sqrt{Q_i} \leq 10h\sqrt{d} + 8hA_0. \tag{111}$$

With $B_0 = 0$ and inequality [35, Lemma 7] or (104), we can derive the following result

$$B_k \leq (19 + 46\sqrt{\Theta})\sqrt{\frac{h^3}{m}}A_0 + (31 + 57\sqrt{\Theta})\sqrt{\frac{h^3 d}{m}}. \tag{112}$$

# C Proof of information-based complexity lower bound

We first introduce some notations in appendix C.1. We prove theorems 5 and 6 for $d = 1$ case in appendices C.2 and C.3. Then we generalize these results to high dimension in appendix C.4. Finally, we prove corollary 2 in appendix C.5.

## C.1 Preliminary

We let $[N]$ means $\{i \in \mathbb{N}|1 \leq i \leq N\}$. For set $\Theta$, $|\Theta|$ means the cardinality or number of elements if $\Theta$ is finite.

## C.2 Flat error when $n < N$

We first show the intuition behind Theorem 5 as follows. Then we give rigorous analysis.

In our definition of problem class $\mathcal{U}$, the mean of the distribution is restricted to a small ball. However, the minimum of each component of the potential is not restricted. Therefore, we can construct a family of adversarial models, such that each component potential function has a minimum being very far away from the origin, in the meantime, the sum potential has a minimum close to the origin. In this case, every component could affect the mean of the final distribution arbitrarily, and the lack of information for even just one component makes the prediction very uninformative.

### C.2.1 Proof of theorem 5 for 1-d Case

We consider the class of randomized algorithms $\mathcal{A}_n$ with step number $n < N$. We first assume $N$ to be positive odd integer in the following proof. At the end of this subsection, we extend the method to even integer $N$.

A $N-1$-steps algorithm could ignore the last $N-1-n$ oracle evaluation and be essentially the same as a $n$-steps algorithms. Therefore, $e_{\mathcal{A}_{N-1},\mathcal{U}} \leq e_{\mathcal{A}_n,\mathcal{U}}$ and we can only consider $n = N-1$ without loss of generality.

**Parameterized model** We consider a class of parameterized model as a subset of $\mathcal{U}$ to give a lower bound of $e_{\mathcal{A},\mathcal{U}}$.

Let parameter be $\theta \in \Theta = \{(a_0,\ldots,a_N) \in \{-1,1\}^{N+1} | \sum_0^N a_i = 0\}$. We define potentials $f_i(x) = \frac{u}{2N}x^2 - \sqrt{u}a_i x$, such that $\sum_i^N f_i(x) = \frac{u}{2}x^2 - \sqrt{u}(\sum_1^N a_i)x$ and global minimum is $x^* = \frac{1}{\sqrt{u}}\sum_1^N a_i = -\frac{1}{\sqrt{u}}a_0 \in \{-\frac{1}{\sqrt{u}}, \frac{1}{\sqrt{u}}\}$. We assume $m \leq u \leq L$, and it is easy to see that $U = (f_1,\ldots,f_N)$ is inside $\mathcal{U}$.

We extend the notation to use $\boldsymbol{X}_T(\omega,\theta)$ and $A(\omega,\widetilde{\omega},\theta)$ to mean $\boldsymbol{X}_T(\omega,U)$ and $A(\omega,\widetilde{\omega},U)$ where $U$ is above model with parameter $\theta$.

$$
\begin{aligned}
e^2_{\mathcal{A}_n,\mathcal{U}} &= \inf_{A\in\mathcal{A}_n} \sup_{U\in\mathcal{U}} \mathbb{E}_{\omega\in\mathbb{P}}\mathbb{E}_{\widetilde{\omega}\in\widetilde{\mathbb{P}}}[\boldsymbol{X}_T(\omega,U) - A(\omega,\widetilde{\omega},U)]^2 \\
&\geq \inf_{A\in\mathcal{A}_n} \sup_{\theta\in\Theta} \mathbb{E}_{\omega\in\mathbb{P}}\mathbb{E}_{\widetilde{\omega}\in\widetilde{\mathbb{P}}}[\boldsymbol{X}_T(\omega,\theta) - A(\omega,\widetilde{\omega},\theta)]^2 \\
&\geq \inf_{A\in\mathcal{A}_n} \frac{1}{|\Theta|}\sum_{\theta\in\Theta} \mathbb{E}_{\omega\in\mathbb{P}}\mathbb{E}_{\widetilde{\omega}\in\widetilde{\mathbb{P}}}[\boldsymbol{X}_T(\omega,\theta) - A(\omega,\widetilde{\omega},\theta)]^2 \\
&\geq \inf_{A\in\mathcal{A}_n} \mathbb{E}_{\omega\in\mathbb{P}}\mathbb{E}_{\widetilde{\omega}\in\widetilde{\mathbb{P}}} \frac{1}{|\Theta|}\sum_{\theta\in\Theta}[\boldsymbol{X}_T(\omega,\theta) - A(\omega,\widetilde{\omega},\theta)]^2
\end{aligned}
\tag{113}
$$

We next give a lower bound on the quantity $\frac{1}{|\Theta|}\sum_{\theta\in\Theta}[\boldsymbol{X}_T(\omega,\theta) - A(\omega,\widetilde{\omega},\theta)]^2$ under a fixed $\omega$ and $\widetilde{\omega}$. The basic idea is that due to lack of information, multiple $\theta$ could generate same $A(\omega,\widetilde{\omega},\theta)$, therefore the error could be lower-bounded by differences between $\boldsymbol{X}_T(\omega,\theta)$ with equivalence $\theta$.

**Mask** We use a mask on parameters to represent how much information could be accessed by the algorithm.

The algorithm takes $N-1$ steps to calculate $A(\omega,\widetilde{\omega},\theta)$. During this process, $N-1$ oracle are evaluated. Therefore, there is at most $N-1$ gradient oracle evaluation $\Upsilon_U(i,x)$. Those indexes $i$ that appear at least once in these gradient evaluations are collected into a mask.

Let $\mathcal{J} = \{m \subseteq [N]\} = 2^{[N]}$, we define a mapping $\beta : \theta \in \Theta \mapsto \beta(\theta) \in \mathcal{J}$ to represent the mask. More specifically, $\beta(\theta)$ contains index $i$ if and only if, at a certain step during calculating $A(\omega,\widetilde{\omega},\theta)$, the algorithm evaluates the gradient oracle $\Upsilon_U(i,x)$ with some $x$.

**Expanded mask** We have $|\beta(\theta)| \leq N-1$, since there is at most $N-1$ gradient evaluation. We next expand this mask into $\beta'(\theta)$ to make sure $|\beta'(\theta)| = N-1$. This is done by simply adding the largest element in $[N]$ that is still not inside $\beta(\theta)$ until the size reaches $N-1$. More specifically, we can always find $k$, such that $|\beta(\theta) \cup \{i \in \mathbb{N}^+ | k \leq i \leq N\}| = N-1$ and we let $\beta'(\theta) = \beta(\theta) \cup \{i \in \mathbb{N}^+ | k \leq i \leq N\}$.

**Complementary mask** We define $\widetilde{\beta'}(\theta) = \{i \in \mathbb{N} | i \leq N\} \setminus \beta'(\theta)$ as complementary mask to represent how much information is not relevant to the algorithm output. It is easy to see that $|\widetilde{\beta'}(\theta)| = 2$ and $0 \in \widetilde{\beta'}(\theta)$.

**Equivalent parameters** For each $i \in \widetilde{\beta'}(\theta)$, the gradient of $f_i(x)$ is not used in the algorithm, and therefore changing the function $f_i$ or parameter $a_i$ will not affect algorithm output $A(\omega,\widetilde{\omega},\theta)$.

Therefore, we say two parameters to be equivalent if they have the same mask, and they are the same under this mask. More specifically, two parameters $\theta = (a_0, \ldots, a_N)$ and $\theta' = (a'_0, \ldots, a'_N)$ are equivalent if and only if $\beta'(\theta) = \beta'(\theta')$ and for any $i \in \beta'(\theta)$, $a_i = a'_i$. For any two equivalent parameters $\theta$ and $\theta'$, the algorithm gives the same output $A(\omega, \widetilde{\omega}, \theta) = A(\omega, \widetilde{\omega}, \theta')$.

**Equivalent class** We divide the parameter space $\Theta$ into a family of disjoint equivalent class $\Theta_i$, such that $\Theta = \cup_i \Theta_i$.

**Lemma 5.** *For any equivalent class, the size is either $1$ or $2$. Moreover, the number of equivalent classes with $2$ elements is at least $\frac{|\Theta|}{4}$.*

*Proof of lemma 5.* We first show that the size of each equivalent class is either $1$ or $2$. For each parameter $\theta = (a_0, \ldots, a_N)$, we have $|\widetilde{\beta}'(\theta)| = 2$ and $0 \in \widetilde{\beta}'(\theta)$. We denote the other element in $\widetilde{\beta}'(\theta)$ to be $u$ such that $\widetilde{\beta}'(\theta) = \{0, u\}$. Then we have $\sum_{i \in \beta(\theta)} a_i = -\sum_{i \in \beta'(\theta)} a_i = -a_0 - a_u \in \{-2, 0, 2\}$. If $\sum_{i \in \beta(\theta)} a_i = -2$, then it is mandatory that $a_0 = -1$ and $a_u = -1$, therefore, there is no other parameter to be equivalent to $\theta$ and the size of equivalent class is $1$. The situation is similar if $\sum_{i \in \beta(\theta)} a_i = 2$. When $\sum_{i \in \beta(\theta)} a_i = 0$, there are two possible value for $a_0, a_u$. $a_0 = 1, a_u = -1$ and $a_0 = -1, a_u = 1$ are both valid, so the size of the equivalent class is $2$.

Before we control the number of equivalent classes, we extend the notation to represent the "path" of the mask and construct a bijection between parameters $\Theta$ and the path. Recall that $\beta'(\theta)$ can be constructed sequentially. We can start from an empty mask $\beta'_0(\theta)$ and $i = 0$. At each step of the algorithm, if gradient oracle is evaluated, we denote the index parameter for gradient oracle as $t_{i+1}$, we add it into the mask as $\beta'_{i+1}(\theta) = \beta'_i(\theta) \cup \{t_{i+1}\}$ and set $i \leftarrow i + 1$. At the end of the algorithm, we still have $i \leq N + 1$. We then repeatedly select largest index from $\{c \in \mathbb{N} | c \leq N, c \notin \beta'_i(\theta)\}$ as $t_{i+1}$, then append it to mask to form $\beta'_{i+1}(\theta)$ and increase $i$ by $1$. The above process is repeated until $i = N + 1$. It is easy to see that $|\beta'_i(\theta)| = i$ and $\beta'(\theta) = \beta'_{N-1}(\theta)$.

We define the mapping $p : \theta \in \Theta \mapsto (a_{t_1}, \ldots, a_{t_{N+1}}) \in \Theta$. This mapping is injective, as two different $\theta$ will give $a_{t_i}$ at certain step in the construction. In the meantime, $p$ is also a self-map. Therefore, $p$ is a bijection, and every path $(a_{t_1}, \ldots, a_{t_{N+1}}) \in \Theta$ can be realized by exactly one $\theta \in \Theta$.

We next show that the number of equivalent classes with $2$ elements is at least $\frac{|\Theta|}{4}$. First, we have $|\Theta| = \binom{N+1}{\frac{N+1}{2}}$. This is because choosing a $\theta$ is equivalent to selecting $\frac{N+1}{2}$ indexes from all $N + 1$ indexes to set $1$ and set $-1$ for all other indexes. Next, we consider all $\theta$ that has distinct equivalent parameter. $\exists! \theta' \in \Theta, \theta \neq \theta', \beta'(\theta) = \beta'(\theta') \iff \sum_{i \in \beta'(\theta)} a_i = 0 \iff \sum_{i=1}^{N-1} a_{t_i} = 0$. Selecting a $\theta$ that has distinct equivalent parameter is equivalent to selecting half indexes for previous $N - 1$ elements in $(a_{t_1}, \ldots, a_{t_{N+1}})$ to be $-1$, one of $a_N$ and $a_{N+1}$ to be $-1$, and all others to be $1$, and finally mapped back to $\theta$ with $p^{-1}$. Therefore, the number of $\theta$ that has distinct equivalent parameter is $2\binom{N-1}{\frac{N-1}{2}}$. Finally, the number of equivalent classes with $2$ elements is $\binom{N-1}{\frac{N-1}{2}} = \frac{1}{4}(1 + \frac{1}{N})|\Theta| \geq \frac{1}{4}|\Theta|$. $\qquad\square$

We only consider equivalent classes with 2 elements to derive a lower bound as follows.

$$\frac{1}{|\Theta|}\sum_{\theta\in\Theta}[\boldsymbol{X}_T(\omega,\theta)-A(\omega,\widetilde{\omega},\theta)]^2$$

$$\geq\frac{1}{|\Theta|}\sum_{\substack{\Theta_i\\|\Theta_i|=2}}\sum_{\theta\in\Theta_i}[\boldsymbol{X}_T(\omega,\theta)-A(\omega,\widetilde{\omega},\theta)]^2$$

$$\geq\frac{1}{|\Theta|}\sum_{\substack{\Theta_i\\|\Theta_i|=2}}([\boldsymbol{X}_T(\omega,\theta)-A(\omega,\widetilde{\omega},\theta)]^2+[\boldsymbol{X}_T(\omega,\theta')-A(\omega,\widetilde{\omega},\theta)]^2)\big|_{\theta,\theta_i\in\Theta_i,\theta\neq\theta_i}$$

$$\geq\frac{1}{|\Theta|}\sum_{\substack{\Theta_i\\|\Theta_i|=2}}\frac{1}{2}[\boldsymbol{X}_T(\omega,\theta)-\boldsymbol{X}_T(\omega,\theta')]^2\big|_{\theta,\theta_i\in\Theta_i,\theta\neq\theta_i} \tag{114}$$

$$\overset{①}{\geq}\frac{1}{|\Theta|}\sum_{\substack{\Theta_i\\|\Theta_i|=2}}\frac{1}{2}C(T)^2\frac{1}{u}$$

$$\geq\frac{1}{|\Theta|}\frac{1}{4}|\Theta|\frac{1}{2}C(T)^2\frac{1}{u}$$

$$=\frac{1}{8}C(T)^2\frac{1}{u}$$

Inequality ① comes from lemma 6. We can just set $u=m$ for largest error.

If $N$ is even integer, we just need to modify our construction of parameterized model. We define $\theta\in\Theta=\{(a_0,\dots,a_N)\in\{-1,1\}^{N+1}|\sum_0^N a_i=1\}$, and $f_i(x)=\frac{u}{2N}x^2-\sqrt{u}(a_i-1/N)x$ such that the global minimum still falls in $\{-\frac{1}{\sqrt{u}},\frac{1}{\sqrt{u}}\}$. Similarly, we can still define masks, equivalent parameters, equivalent class, and we can still have the number of equivalent classes with exactly two elements being larger than $\frac{1}{4}$ times total number of parameters. Finally, we can still give a lower bound in the same form.

### C.2.2 Explicit ULD solution for 1-d quadratic potential

We consider ULD process on potential function $f(x)=\frac{u}{2}x^2$ starting from $x_0,v_0$.

$$d\begin{bmatrix}x_t\\v_t\end{bmatrix}=\begin{bmatrix}0&1\\-u&-\gamma\end{bmatrix}\begin{bmatrix}x_t\\v_t\end{bmatrix}dt+\begin{bmatrix}0\\\sqrt{2\gamma}\end{bmatrix}dB_t \tag{115}$$

We define $H=\begin{bmatrix}0&1\\-u&-\gamma\end{bmatrix}$. It is easy to verify that the following solution solves the SDE.

$$\begin{bmatrix}x_t\\v_t\end{bmatrix}=e^{Ht}\begin{bmatrix}x_0\\v_0\end{bmatrix}+\int_0^t e^{H(t-s)}\begin{bmatrix}0\\\sqrt{2\gamma}\end{bmatrix}dB_t \tag{116}$$

We can also derive $e^{Ht}$ explicitly.

$$e^{Ht}=\frac{1}{\lambda_+-\lambda_-}\left(e^{\lambda_-t}\begin{bmatrix}\lambda_+&1\\-u&-\lambda_-\end{bmatrix}-(e^{\lambda_+t}\begin{bmatrix}\lambda_-&1\\-u&-\lambda_+\end{bmatrix})\right) \tag{117}$$

$$\lambda_\pm=\frac{\gamma\pm\sqrt{\gamma^2-4u}}{2} \tag{118}$$

### C.2.3 Difference between two ULD processes with same Brownian motion but different quadratic potential

**Lemma 6.** *Given two quadratic potential $f(x)=\frac{u}{2}x^2-\sqrt{u}x$ and $f'(x)=\frac{u}{2}x^2+\sqrt{u}x$, consider two ULD process $X_T(\omega,f)$ and $X_T(\omega,f')$ starting from same initialization $\tilde{X}_0=0,V_0=0$. Then, we have*

$$0\leq X_T(\omega,f)-X_T(\omega,f')\leq C(T)\frac{1}{\sqrt{u}},$$

*where $C(T)$ only depends on $T$, $\gamma$, $u$.*

*Proof.* First consider a quadratic potential $\widetilde{f}(x) = \frac{u}{2}x^2$. Then we have $X_T(\omega, f) = \frac{1}{\sqrt{u}} + \widetilde{X}_T(\omega, \widetilde{f})$ where $\widetilde{X}_T(\omega, \widetilde{f})$ is ULD process starting from $X_0 = -\frac{1}{\sqrt{u}}$. Similarly, $X_T(\omega, f') = -\frac{1}{\sqrt{u}} + \widetilde{X}'_T(\omega, \widetilde{f})$ where $\widetilde{X}'_T(\omega, \widetilde{f})$ is ULD process starting from $X_0 = \frac{1}{\sqrt{u}}$.

$\widetilde{X}_T(\omega, \widetilde{f})$ and $\widetilde{X}'_T(\omega, \widetilde{f})$ and be calculated explicitly by result in appendix C.2.2. Therefore, we have the following result.

$$0 \leq X_T(\omega, f) - X_T(\omega, f') \leq (1 - \frac{e^{-\lambda_- T}\lambda_+ - e^{\lambda_+ T}\lambda_-}{\lambda_+ - \lambda_-})\frac{2}{\sqrt{u}} \tag{119}$$

$\square$

## C.3 Error by perturbation

With similar tools in previous section, we can derive a lower bound by applying perturbation on a quadratic potential function.

### C.3.1 Proof of theorem 6 for 1-d Case

**Parameterized model**  For positive values $m < u < L$, $C_x$ and $\varepsilon$, we define $x_i = \frac{NC_x}{2n}(i - \frac{n}{N})$ for $0 \leq i \leq \frac{2n}{N}$, $I_i = [x_i, x_{i+1}]$ for $0 \leq i < \frac{2n}{N}$, and $f_0(x) = \frac{u}{2N}x^2$.

Let parameter be $\theta \in \Theta = \{0, 1\}^{2n}$, we define $f_i(x)$ by it's gradient and global optimal. We require that $\nabla f_i(x) = \nabla f_0(x) + \sum_{j=0}^{\frac{2n}{N}-1} \theta_{jN+i}\, g(x - x_j)$ and $0 = \operatorname{argmin}_x f_i(x)$ for $1 \leq i \leq N$.

The function $g$ represents a local perturbation.

$$g(x) := \begin{cases} \frac{4n\xi}{N^2 C_x}x^2 & x \in [0, \frac{NC_x}{8n}] \\ \frac{4n\xi}{N^2 C_x}[-(x - \frac{NC_x}{4n})^2 + 2(\frac{NC_x}{8n})^2] & x \in [\frac{NC_x}{8n}, \frac{3NC_x}{8n}] \\ \frac{4n\xi}{N^2 C_x}(x - \frac{NC_x}{2n})^2 & x \in [\frac{3NC_x}{8n}, \frac{NC_x}{2n}] \\ 0 & x \notin [0, \frac{NC_x}{2n}] \end{cases} \tag{120}$$

It can be easily verify that $\|g'\|_\infty = \frac{\xi}{N}$, $\|g\|_\infty = \frac{C_x \xi}{8n}$ and when $u - m \leq \xi \leq L - u$, we have $U = (f_1, \ldots, f_N) \in \mathcal{U}$ for every parameter $\theta$.

Similar to the flat error case, we extend the notation to use $\boldsymbol{X}_T(\omega, \theta)$ and $A(\omega, \widetilde{\omega}, \theta)$ to mean $\boldsymbol{X}_T(\omega, U)$ and $A(\omega, \widetilde{\omega}, U)$ where $U$ is above model with parameter $\theta$.

$$e^2_{\mathcal{A}_n, \mathcal{U}} \geq \inf_{A \in \mathcal{A}_n} \mathbb{E}_{\omega \in \mathbb{P}} \mathbb{E}_{\widetilde{\omega} \in \widetilde{\mathbb{P}}} \frac{1}{|\Theta|} \sum_{\theta \in \Theta} [\boldsymbol{X}_T(\omega, \theta) - A(\omega, \widetilde{\omega}, \theta)]^2 \tag{121}$$

**Mask**  The algorithm takes $n$ steps to calculate $A(\omega, \widetilde{\omega}, \theta)$. During this process, $n$ oracle are evaluated. Therefore, there is at most $n$ gradient oracle evaluation $\Upsilon_U(i, x)$. For each gradient oracle evaluation, if $x \in [-\frac{C_x}{2}, \frac{C_x}{2})$, then there exists exactly one $j$ such that $x \in I_j$. In this case, we add $jN + i$ into the mask.

Let $\mathcal{J} = \{m \subseteq [2n]\} = 2^{[2n]}$, we define a mapping $\beta : \theta \in \Theta \mapsto \beta(\theta) \in \mathcal{J}$ to represent the mask. More specifically, $\beta(\theta)$ contains index $jN + i$ if and only if, at a certain step during calculating $A(\omega, \widetilde{\omega}, \theta)$, the algorithm evaluates the gradient oracle $\Upsilon_U(i, x)$ with $x \in I_j$.

**Expanded mask**  We have $|\beta(\theta)| \leq n$, since there is at most $n$ gradient evaluation. We expand this mask into $\beta'(\theta)$ to make sure $|\beta'(\theta)| = n$ by repeatedly adding the largest element in $[2n]$ that is still not inside $\beta(\theta)$ until the size reaches $n$. More specifically, we can always find $k$, such that $|\beta(\theta) \cup \{i \in \mathbb{N}^+ | k \leq i \leq 2n\}| = n$ and we let $\beta'(\theta) = \beta(\theta) \cup \{i \in \mathbb{N}^+ | k \leq i \leq 2n\}$.

**Complementary mask**  We define $\widetilde{\beta}'(\theta) = \{i \in \mathbb{N}^+ | i \leq 2n\} \setminus \beta'(\theta)$ as complementary mask to represent how much information is not relevant to the algorithm output. It is easy to see that $|\widetilde{\beta}'(\theta)| = n$.

**Equivalent parameters** We say two parameters to be equivalent if they have the same mask, and they are the same under this mask. More specifically, two parameters $\theta$ and $\theta'$ are equivalent if and only if $\beta'(\theta) = \beta'(\theta')$ and for any $i \in \beta'(\theta)$, $\theta_i = \theta'_i$. For any two equivalent parameters $\theta$ and $\theta'$, the algorithm gives the same output $A(\omega, \widetilde{\omega}, \theta) = A(\omega, \widetilde{\omega}, \theta')$.

**Equivalent class** We divide the parameter space $\Theta$ into a family of disjoint equivalent class $\Theta_i$, such that $\Theta = \cup_i \Theta_i$.

For each parameter $\theta$, by altering different digits at $\widetilde{\beta'}(\theta)$, we can construct $2^n$ different equivalent parameter, therefore, the size of any equivalent class is always $2^n$. Then the number of equivalent classes is just $2^{2n}/2^n = 2^n$.

We next construct a pairing between parameters in each equivalent class $\Theta_i$ with a self-map $S$ on $\Theta_i$. For any $\theta, \theta' \in \Theta_i$, we construct an order between them $\theta \succ \theta' \iff \forall i \in \widetilde{\beta'}(\theta), \theta_i \geq \theta'_i$. Based on this order, all $2^n$ elements in $\Theta_i$ forms a Boolean lattice. The rank of $\theta$ in this lattice is just $\rho(\theta) = |\{i \in \widetilde{\beta'}(\theta)|\theta_i = 1\}|$. We select a symmetric chain decomposition $\mathcal{C}$ for this Boolean lattice. For any parameter $\theta \in \Theta_i$, there exist only one symmetric chain $C \in \mathcal{C}$ that contains $\theta$. We pick the element $\theta'$ in chain $C$ with rank $\rho(\theta') = n - \rho(\theta)$. This always possible because the chain is rank-symmetric and saturated. We then define the map $S : \theta \mapsto \theta'$. This mapping is bijective and $S(S(\theta)) = \theta$.

$$\frac{1}{|\Theta|} \sum_{\theta \in \Theta} [\boldsymbol{X}_T(\omega, \theta) - A(\omega, \widetilde{\omega}, \theta)]^2$$

$$= \frac{1}{|\Theta|} \sum_{\Theta_i} \sum_{k=0}^{n} \sum_{\substack{\theta \in \Theta_i \\ \rho(\theta)=k}} [\boldsymbol{X}_T(\omega, \theta) - A(\omega, \widetilde{\omega}, \theta)]^2$$

$$= \frac{1}{|\Theta|} \sum_{\Theta_i} \sum_{k=0}^{n} \sum_{\substack{\theta \in \Theta_i \\ \rho(\theta)=k}} \frac{1}{2}([\boldsymbol{X}_T(\omega, \theta) - A(\omega, \widetilde{\omega}, \theta)]^2 + [\boldsymbol{X}_T(\omega, S(\theta)) - A(\omega, \widetilde{\omega}, \theta)]^2)$$

$$\geq \frac{1}{|\Theta|} \sum_{\Theta_i} \sum_{k=0}^{n} \sum_{\substack{\theta \in \Theta_i \\ \rho(\theta)=k}} \frac{1}{4}[\boldsymbol{X}_T(\omega, \theta) - \boldsymbol{X}_T(\omega, S(\theta))]^2 \tag{122}$$

$$\overset{①}{\geq} \frac{1}{|\Theta|} \sum_{\Theta_i} \sum_{k=0}^{n} \sum_{\substack{\theta \in \Theta_i \\ \rho(\theta)=k}} \frac{1}{4}[C\frac{N\xi}{n^2}(n-k-k)]^2 \mathbb{I}_{\mathcal{E}}(\omega)$$

$$= \frac{1}{2^{2n}} 2^n \sum_{k}^{n} \binom{n}{k} \frac{1}{4}[C\frac{N\xi}{n^2}(n-k-k)]^2 \mathbb{I}_{\mathcal{E}}(\omega)$$

$$= \frac{1}{4} C^2 \xi^2 \frac{N^2}{n^4} \mathbb{I}_{\mathcal{E}}(\omega)$$

Inequality ① comes from lemma 7.

$$e^2_{\mathcal{A}_n, \mathcal{U}} \geq \inf_{A \in \mathcal{A}_n} \mathbb{E}_{\omega \in \mathbb{P}} \mathbb{E}_{\widetilde{\omega} \in \widetilde{\mathbb{P}}} \frac{1}{|\Theta|} \sum_{\theta \in \Theta} [\boldsymbol{X}_T(\omega, \theta) - A(\omega, \widetilde{\omega}, \theta)]^2$$

$$\geq \inf_{A \in \mathcal{A}_n} \mathbb{E}_{\omega \in \mathbb{P}} \mathbb{E}_{\widetilde{\omega} \in \widetilde{\mathbb{P}}} \frac{1}{4} C^2 \xi^2 \frac{N^2}{n^3} \mathbb{I}_{\mathcal{E}}(\omega) \tag{123}$$

$$= \frac{1}{4} C^2 \xi^2 \mathbb{P}(\mathcal{E}) \frac{N^2}{n^3}$$

Then we can optimize all free parameter including $\xi$, $C_x$ and $C_v$ introduced in appendix C.3.2 to derive final result.

$$e^2_{\mathcal{A}_n, \mathcal{U}} \geq C_2 \frac{N^2}{n^3} \tag{124}$$

According to the discussion in appendix C.3.2, $\varepsilon = \frac{NC_x\xi}{8n} \leq \frac{C_x\xi}{8}$ and we can always select parameters, so that $\varepsilon$ is small enough and $C_v$ is large enough such that $\mathbb{P}(\mathcal{E}) > 0$. Therefore, we have $C_2 > 0$.

### C.3.2  Perturbation error

For positive values $\varepsilon > 0, C_v > 0$ we define

$$\mathcal{U}_{u,\varepsilon} = \{(f_1, \ldots, f_N) \in \mathcal{U} | \|\sum_i^N \nabla f_i(x) - ux\|_\infty \leq \varepsilon\} \tag{125}$$

$$\mathcal{E} = \{\omega \in \mathbb{M} | \forall U \in \mathcal{U}_{u,\varepsilon}, \sup_{0 \leq t \leq T} \boldsymbol{X}_t(\omega, U) \geq C_x,$$
$$\inf_{0 \leq t \leq T} \boldsymbol{X}_t(\omega, U) \leq -C_x, \sup_{0 \leq t \leq T} \boldsymbol{V}_t(\omega, U) \leq C_v\} \tag{126}$$

**[10, Lemma 2.3, Lemma 3.2]**   When $\varepsilon$ is small enough and $C_v$ is large enough, the event $\mathcal{E}$ happens with a positive probability. More specifically, there exists $\overline{\varepsilon} > 0$ that only depends on $L, u, C_x, C_v$ and $\overline{C_v}$ that only depends on $L, u, C_x$, such that when $\varepsilon \leq \overline{\varepsilon}$ and $C_v \geq \overline{C_v}$, we have $\mathbb{P}(\mathcal{E}) > 0$.

**[10, Proposition 3.1]**   Given $U^{(1)}, U^{(2)} \in \mathcal{U}_{u,\varepsilon}, \mathcal{I} \subseteq [-\frac{C_x}{2}, \frac{C_x}{2}]$ as finite union of closed bounded interval, such that

$$g(x) = \sum_{i=1}^N \nabla f_i^{(1)}(x) - \sum_{i=1}^N \nabla f_i^{(2)}(x) \geq c\varepsilon \mathbb{I}_\mathcal{I}(x), \forall x \in \mathbb{R}$$

where $\mathbb{I}$ is the indicator function. Then for every $\omega \in \mathcal{E}$ we have

$$\boldsymbol{X}_T(\omega, U^{(2)}) - \boldsymbol{X}_T(\omega, U^{(1)}) \geq \overline{C}c\varepsilon\mu(\mathcal{I})$$

where $\overline{C}$ is always positive and only depends on $L, u, C_x, C_v$.

**Remark 4.** *The statement in the above proposition is not exactly the same as the original statement in [10]. Some modification is made to adapt to this paper's setup. We address these differences as follows.*

*Value $u_R$: The original statement in [10] assumes $\nabla^2 U(x) \leq u_R \leq L$. Here we just assume $u_R = L$ for simplicity.*

*Sign: The original statement in [10, Proposition 3.1] only gives lower bounds for absolute value $|\boldsymbol{X}_T(U^{(1)}, \omega) - \boldsymbol{X}_T(U^{(2)}, \omega)|$. However, [10, Lemma 3.7] already shows that $\boldsymbol{X}_T(U^{(1)}, \omega) - \boldsymbol{X}_T(U^{(2)}, \omega) \leq 0$.*

*Scale of $g(x)$: The original statement in [10, Proposition 3.1] assumes $c = \frac{1}{2}$, however, this is not necessary. In the proof of [10, Proposition 3.1], $g(x)$ is used linearly, therefore the proposition should be valid for all $c$. We always have $c \leq 2$ because $U^{(1)}, U^{(2)} \in \mathcal{U}_{u,\varepsilon}$.*

*Scale of ULD process: The SDE for ULD in [10] has different scale than this paper, this can be resolved by a linear transformation as discussed in Appendix A.3 or simply assuming $\gamma = 2$ and $L = 1$.*

**Lemma 7.** *Let $\varepsilon = \frac{NC_x\xi}{8n}$. If $\omega \in \mathcal{E}$, then for any two parameter $\theta, \theta'$ in an equivalent class $\Theta_i$, if $\theta \prec \theta'$, we have*

$$\boldsymbol{X}_T(\omega, \theta) - \boldsymbol{X}_T(\omega, \theta') \geq C\frac{N\xi}{n^2}(\rho(\theta') - \rho(\theta))$$

*where $C$ is always positive and only depends on $L, u, C_x, C_v$.*

*Proof of lemma 7.* For any $\theta$, we have $U \in \mathcal{U}_{u,\varepsilon}$.

Recall that $\Theta_i$ forms a finite Boolean lattice with order $\succ$. Therefore, we can always find a saturated chain $\theta^{(\rho(\theta))} \prec \theta^{(\rho(\theta))+1} \prec \cdots \prec \theta^{(\rho(\theta'))}$, so that $\theta = \theta^{(\rho(\theta))}$ and $\theta' = \theta^{(\rho(\theta'))}$.

For any two adjacent element in this chain $\theta^{(u)}, \theta^{(u+1)} \in 2^{[2n]}$, we know that $\theta^{(u+1)}$ is only greater than $\theta^{(u)}$ at one index $jN + i$. Therefore, the corresponding functions $f^{(u)} = \sum_{i=1}^N f_i^{(u)}(x)$ and

$f^{(u+1)} = \sum_{i=1}^{N} f_i^{(u+1)}(x)$ only differ at $f_i(x)$ within interval $I_j$. More specifically, $\nabla f^{(u+1)}(x) - \nabla f^{(u)}(x) = g(x - x_j) \geq \frac{1}{2N}\varepsilon \mathbb{I}_{\mathcal{I}}$ where $\mathcal{I} = [x_j + \frac{NC_x}{8n}, x_j + \frac{3NC_x}{8n}]$. [10, Proposition 3.1] can be applied to give

$$\boldsymbol{X}_T(\omega, U^{(u)}) - \boldsymbol{X}_T(\omega, U^{(u+1)}) \geq \overline{C}\frac{NC_x^2\xi}{64n^2}$$

Telescoping the above inequality for all $u$, we have

$$\boldsymbol{X}_T(\omega, U^{(\rho(\theta))}) - \boldsymbol{X}_T(\omega, U^{(\rho(\theta'))}) \geq \overline{C}\frac{NC_x^2\xi}{64n^2}(\rho(\theta') - \rho(\theta))$$

$\square$

## C.4  High dimension case

In this section we extend the proof in previous sections to high dimensional space.

We have already established a parameterized model for controlling both flat error and perturbation error in 1 -dimensional space. We now define a parameterized model for high dimensional space. The parameter space is just the Cartesian product of parameters for a single dimension $\Theta = \times_{i=1}^{d}\Theta^{(i)}$. Given a parameter $(\theta^{(1)}, \ldots, \theta^{(d)}) \in \Theta$, we construct a potential function as follows. For each $1 \leq i \leq d$, let $\widetilde{f^{(i)}}$ be the 1-d potential corresponding to $\theta^{(i)}$. Then we define $f(\boldsymbol{X}) = \sum_{i=1}^{d} f^{(i)}(\boldsymbol{X}) = \sum_{i=1}^{d} \widetilde{f^{(i)}}(\boldsymbol{X}^{(i)})$. Clearly, the different dimension components of ULD decouple with each other, so $\boldsymbol{X}_T^{(i)}$ is just a ULD on the 1-dimensional potential $\widetilde{f^{(i)}}$ and is only affected by $\theta^{(i)}$.

$$
\begin{aligned}
e_{\mathcal{A},\mathcal{U}}^2 &= \inf_{A \in \mathcal{A}} \sup_{U \in \mathcal{U}} \mathbb{E}_{\omega \in \mathbb{P}} \mathbb{E}_{\widetilde{\omega} \in \widetilde{\mathbb{P}}} \|\boldsymbol{X}_T(\omega, U) - A(\omega, \widetilde{\omega}, U)\|_2^2 \\
&= \inf_{A \in \mathcal{A}} \sup_{U \in \mathcal{U}} \mathbb{E}_{\omega \in \mathbb{P}} \mathbb{E}_{\widetilde{\omega} \in \widetilde{\mathbb{P}}} \sum_{i=1}^{d} [\boldsymbol{X}_T^{(i)}(\omega, U) - A^{(i)}(\omega, \widetilde{\omega}, U)]^2 \\
&\geq \inf_{A \in \mathcal{A}} \frac{1}{\prod_{i=1}^{d} |\Theta^{(i)}|} \sum_{\substack{\theta^{(i)} \in \Theta^{(i)} \\ i=1,\ldots,d}} \mathbb{E}_{\omega \in \mathbb{P}} \mathbb{E}_{\widetilde{\omega} \in \widetilde{\mathbb{P}}} \sum_{i=1}^{d} [\boldsymbol{X}_T^{(i)}(\omega, \theta) - A^{(i)}(\omega, \widetilde{\omega}, \theta)]^2 \\
&\geq \inf_{A \in \mathcal{A}} \sum_{i=1}^{d} \frac{1}{\prod_{j \neq i} |\Theta^{(j)}|} \sum_{\substack{\theta^{(j)} \in \Theta^{(j)} \\ j \neq i}} \mathbb{E}_{\omega \in \mathbb{P}} \mathbb{E}_{\widetilde{\omega} \in \widetilde{\mathbb{P}}} \\
&\qquad\qquad \frac{1}{|\Theta^{(i)}|} \sum_{\theta^{(i)} \in \Theta^{(i)}} [\boldsymbol{X}_T^{(i)}(\omega, \theta) - A^{(i)}(\omega, \widetilde{\omega}, \theta)]^2
\end{aligned}
$$
(127)

By fixing dimension index $i$, all other parameters $\theta^{(j)} \in \Theta^{(j)}$ with $j \neq i$, random events $\omega, \widetilde{\omega}$, we reduce the problem into constructing lower bounds for $\frac{1}{|\Theta^{(i)}|} \sum_{\theta^{(i)} \in \Theta^{(i)}} [\boldsymbol{X}_T^{(i)}(\omega, \theta) - A^{(i)}(\omega, \widetilde{\omega}, \theta)]^2$. This is exactly the same as the case for 1-d problem, and we can again define the mask, expanded mask, complementary mask, equivalent parameters under this setting and generate the same lower bounds. Therefore, the lower bound for error of $d$ dimensional space is just $d$ times the result from 1 dimensional case.

## C.5  Proof of corollary 2

When $\varepsilon^2 < dC_1$, according to theorem 5, we know that $n \geq N$.

Let $n' = \lceil \frac{n}{N} \rceil N$, we have $n \leq n' \leq n + N - 1$. According to theorem 6, we have $e_{\mathcal{A}_{n'},\mathcal{U}}^2 \geq dC_2\frac{N^2}{n'^3}$. According to monotonicity, we have $e_{\mathcal{A}_n,\mathcal{U}} \geq e_{\mathcal{A}_{n'},\mathcal{U}}$. We also have $\varepsilon \geq e_{\mathcal{A}_n,\mathcal{U}}$. Combining the above inequities gives us $n \geq C_2^{\frac{1}{3}} d^{\frac{1}{3}} N^{\frac{2}{3}} \varepsilon^{-\frac{2}{3}} - N + 1$.

We then combine above two lower bounds to finish the proof.

$$
\begin{aligned}
n &\geq \frac{2}{3}N + \frac{1}{3}(C_2^{\frac{1}{3}}d^{\frac{1}{3}}N^{\frac{2}{3}}\varepsilon^{-\frac{2}{3}} - N + 1) \\
&\geq \frac{1}{3}(N + C_2^{\frac{1}{3}}d^{\frac{1}{3}}N^{\frac{2}{3}}\varepsilon^{-\frac{2}{3}} + 1)
\end{aligned}
\tag{128}
$$

## D  Extra discussion on experiments

### D.1  Estimating the discretization error

We wish to calculate discretization error $\sqrt{\|x_K - X_{Kh}\|_2^2 + \|v_K - V_{Kh}\|_2^2}$ for comparing different algorithms, where $x_k, v_k$ are generated by the algorithm, and $X_t, V_t$ are the true solution.

For an arbitrary non-linear model, the true solution of ULD process might not be available in closed form. Therefore, instead of calculating the error between an algorithm and the true solution, we calculate the error between an algorithm and another algorithm which is guaranteed to have much smaller discretization error. This typically can be achieved with a full gradient RMM with small enough step size.

Naturally, a following question is how to make sure these two algorithms are approximating the same ULD process. In another word, we need to make sure that the noise terms in these two algorithms are derived from the same realization of Brownian motion. We introduce a novel technique by accumulating the noise terms in the algorithms with smaller step size to generate the noise term for algorithms with larger step size. If the two algorithms uses step size $h'$ and $nh'$ respectively with positive integer $n$, we show that the noise terms $e_{x,[0,nh']}, e_{v,[0,nh']}, e_{x,[0,anh']}$ can be represented as a combination of $e_{x,[ih',(i+1)h']}, e_{v,[ih',(i+1)h']}, e_{x,[\lfloor an \rfloor h', anh']}$. The detailed derivation can be found in appendix A.11.

Finally, we calculate an average of errors along the path instead of at last iterate to reduce variance. We call this as trajectory error.

The final method to estimate the discretization error is shown in Algorithm 3. Typically, we select algorithm $B$ as RMM with full gradient and segments number $n = 10$ to approximate discretization error for an algorithm $A$.

A following question is: how accurate this estimation is to the real trajectory error $\frac{1}{K}\sum_{i=1}^{K}\sqrt{\|x_k - X_{kh}\|_2^2 + \|v_k - V_{kh}\|_2^2}$? We know that when $n$ increases, $h' = \frac{h}{n}$ decreases, so the reference path converge to the real solution. Therefore, by selecting a large enough $n$, we can approximate the trajectory error to arbitrary accuracy. We apply Algorithm 3 to estimate the trajectory error of RMM with fixed step size under different segments number $n$. Figure 4 shows that selecting $n = 10$ could generate an accurate enough estimate.

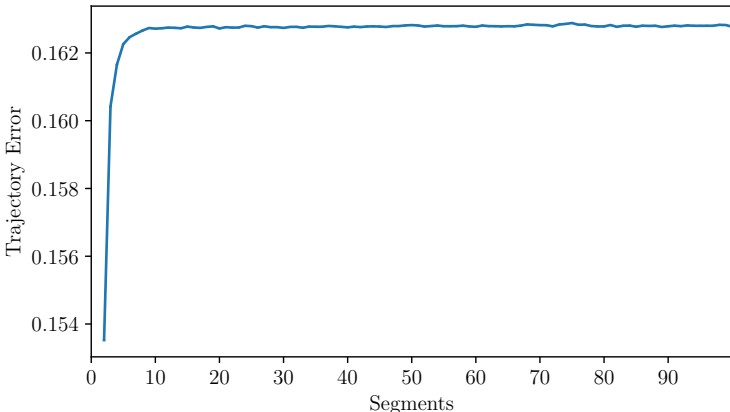

Figure 4: Estimated trajectory error of RMM with different segments number.

**Algorithm 3:** Method to estimating trajectory error.

**Input:** Initial point $(\boldsymbol{x}_0, \boldsymbol{v}_0)$, parameter $\gamma$, iteration number $K$ and step size $h > 0$.
**Input:** Two algorithms $A, B \in \{\text{LPM}, \text{RMM}, \text{ALUM}, \text{VR-ALUM}, \dots\}$, number of segments $n$.
Initialize $\boldsymbol{x}_0' = \boldsymbol{x}_0, \boldsymbol{x}_0' = \boldsymbol{x}_0$.
**for** $k = 0$ **to** $K - 1$ **do**
    $h' = \frac{h}{n}$;
    *// Generate noise terms*
    **for** $i = 0$ **to** $n - 1$ **do**
        Randomly sample $a_i$ uniformly from $[0, 1]$;
        Generate $\boldsymbol{e}_{x,[ih',(i+1)h']}$, $\boldsymbol{e}_{v,[ih',(i+1)h']}$, $\boldsymbol{e}_{x,[ih',(i+a_i)h']}$ according to appendix A.5;
    **end for**
    *// Generate reference path*
    Initialize $\boldsymbol{x}_0'' = \boldsymbol{x}_k', \boldsymbol{x}_0'' = \boldsymbol{x}_k'$.
    **for** $i = 0$ **to** $n - 1$ **do**
        Generate $\boldsymbol{x}_{i+1}''$ and $\boldsymbol{v}_{i+1}''$ by algorithm $B$ with input $\boldsymbol{x}_i''$ and $\boldsymbol{v}_i''$ and noise terms
        $\boldsymbol{e}_{x,[ih',(i+1)h']}$, $\boldsymbol{e}_{v,[ih',(i+1)h']}$, $\boldsymbol{e}_{x,[ih',(i+a_i)h']}$;
    **end for**
    $\boldsymbol{x}_{k+1}' = \boldsymbol{x}_n'', \boldsymbol{v}_{k+1}' = \boldsymbol{v}_n''$.
    *//Accumulate noise terms*
    Randomly sample $j \in \{0, \dots, n-1\}$ with uniform distribution and let $a' = j + a_j$, $a = \frac{a'}{n}$;
    $\boldsymbol{e}_{x,[0,h]} = \sum_{i=0}^{n-1} \left( \boldsymbol{e}_{x,[ih',(i+1)h']} + \psi_1((n-i-1)h') \boldsymbol{e}_{v,[ih',(i+1)h']} \right)$;
    $\boldsymbol{e}_{v,[0,h]} = \sum_{i=0}^{n-1} \psi_0((n-i-1)h') \boldsymbol{e}_{v,[ih',(i+1)h']}$;
    $\boldsymbol{e}_{x,[0,ah]} = \boldsymbol{e}_{x,[\lfloor a' \rfloor h', a'h']} + \sum_{i=0}^{\lfloor a' \rfloor - 1} \left( \boldsymbol{e}_{x,[ih',(i+1)h']} + \psi_1((a'-i-1)h') \boldsymbol{e}_{v,[ih',(i+1)h']} \right)$;
    *// Generate original path with large step size*
    Generate $\boldsymbol{x}_{k+1}$ and $\boldsymbol{v}_{k+1}$ by algorithm $A$ with input $\boldsymbol{x}_k$ and $\boldsymbol{v}_k$ and noise terms $\boldsymbol{e}_{v,[0,h]}$, $\boldsymbol{e}_{v,[0,h]}$,
    $\boldsymbol{e}_{x,[0,ah]}$;
**end for**
**Output:** $\frac{1}{K} \sum_{i=1}^{K} \sqrt{\|\boldsymbol{x}_k - \boldsymbol{x}_k'\|_2^2 + \|\boldsymbol{v}_k - \boldsymbol{v}_k'\|_2^2}$.

There also exists theoretical guarantee on the accuracy of this trajectory error estimation. Since we use RMM for algorithm $B$, we have discretization error for this reference path as $O(h'^{\frac{3}{2}}) = O(h^{\frac{3}{2}} n^{-\frac{3}{2}})$. Therefore, by selecting $n = 10$, we know the discretization error for this reference path is at least $10^{\frac{3}{2}} \approx 30$ times smaller than RMM with original step size. The accuracy can then be derived as $\|\boldsymbol{z}_k - \boldsymbol{Z}_{kh}\|_{\mathbb{L}_2} - \|\boldsymbol{z}_k' - \boldsymbol{Z}_{kh}\|_{\mathbb{L}_2} \leq \|\boldsymbol{z}_k - \boldsymbol{z}_k'\|_{\mathbb{L}_2} \leq \|\boldsymbol{z}_k - \boldsymbol{Z}_{kh}\|_{\mathbb{L}_2} + \|\boldsymbol{z}_k' - \boldsymbol{Z}_{kh}\|_{\mathbb{L}_2}$. [11]

### D.2 Detailed setup

**Datasets** For Gaussian model:

$$f_i(\boldsymbol{x}) = \frac{1}{2N}(\boldsymbol{d}_i - \boldsymbol{x})^\top \Sigma^{-1}(\boldsymbol{d}_i - \boldsymbol{x}), \tag{129}$$

we let $d = 5$ and $N = 100$. The vectors $\boldsymbol{d}_i$ are generated from $\mathcal{N}(\mathbf{2}, 2I)$, where $\mathbf{2}$ is a vector with $2$ in all its elements. $\Sigma$ is generated by QR decomposition of a random Gaussian matrix and then re normalized to have smallest and largest eigenvalue of $m = 1, L = 10$.

For logistic regression:

$$f_i(\boldsymbol{x}) = \frac{m}{2N}\|\boldsymbol{x}\|_2^2 + \sum_{i=1}^{N} \log(1 + \exp(-y_i \boldsymbol{a}_i^\top \boldsymbol{x})), \tag{130}$$

we summarize four datasets used in Table 4. We split the dataset by half randomly for training and testing model. We set parameter $m$ according to $L = \frac{1}{4}\sigma_{\max}(A^\top A) + m$ such that the final condition number $\kappa$ is $10^4$ for australian, $10^5$ for phishing dataset, $10^3$ for german dataset, and $10$ for mushromms dataset.

---

[11] Actually, this analysis is not tight. As we can see in fig. 4, $n = 2$ could achieve about $1/10$ relative accuracy, but the theory predicts about $2^{\frac{3}{2}} \approx 1/3$. Most error of RMM comes from the variance instead of bias, which affects the estimate error differently. We leave the tight analysis of trajectory error estimation for future works.

Table 4: The summary of different datasets used in our experiments.

| Dataset | australian | german | phishing | mushrooms |
|---------|-----------|--------|----------|-----------|
| $N$ | 690 | 1000 | 11055 | 8124 |
| $d$ | 14 | 24 | 68 | 112 |

**Hyperparameters**  If there is no further explanation, we use $b = 20$ for Gaussian model and $b = 40$ for Logistic regression models.

We use $\tau \approx N/b$ for SVRG. More specifically, we update the full gradient after every $N$ evaluations of the single component gradient $\nabla f_i(x)$. This means that the epoch length of different epochs can vary up to 1. This setup doesn't affect our theoretical result, as the bounded MSE property in Appendix B.3 always holds for SVRG with $\Theta = \frac{N^2}{b^3}$.

**Tasks**

1. Discussing the relationship between error and gradient evaluation number: We first transform the potential to satisfy $L = 1$, as discussed in Remark 1. Then we choose certain step size $h$, run different algorithms to estimate ULD till time $T$. Time $T$ is $T = 10$ for Gaussian model and $T = 100$ for logistic regression. For any chosen step size $h$, we can record how many gradient oracles $\nabla f_i(x)$ was evaluated for $x$-axis and record the trajectory error as $y$-axis.

2. Discussing the relationship between error and step size: We use same $T$ and other settings as above. The only difference is that we report step size in $x$-axis.

3. Discussing the relationship between error and batch size: We select a group of batch sizes $b$. For each $b$, we use the same settings as Item 1 to generate the plot.

4. Sampling: The detailed setup for sampling is shown in Appendix D.6.

### D.3   Results on other datasets

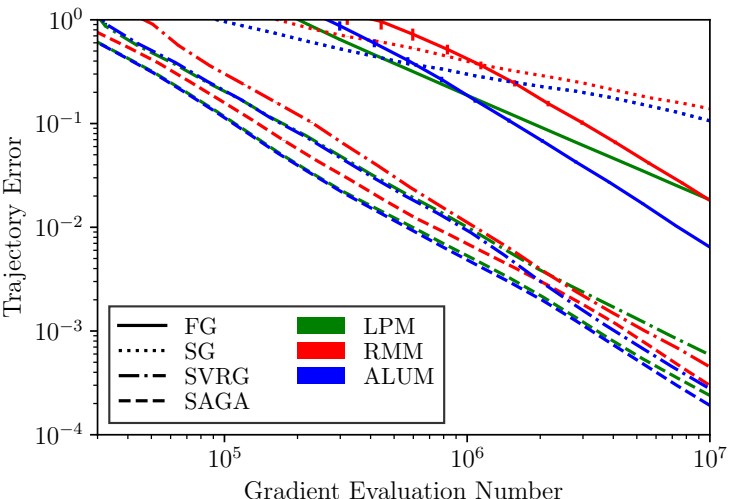

Figure 5: Logistic regression on phishing dataset.

We show discretization error on phishing, german, and mushrooms datasets in Figures 5 to 7.

We can see that SAGA-ALUM constantly outperforms all other algorithms.

Although in Figure 5, the difference between SAGA-ALUM and SAGA-LPM seems small when we only consider less than $10^7$ gradient evaluations, we know that gradient complexity of LPM has worse dependence on accuracy $\varepsilon$, therefore the difference will grow larger when we increase the budget of gradient evaluations.

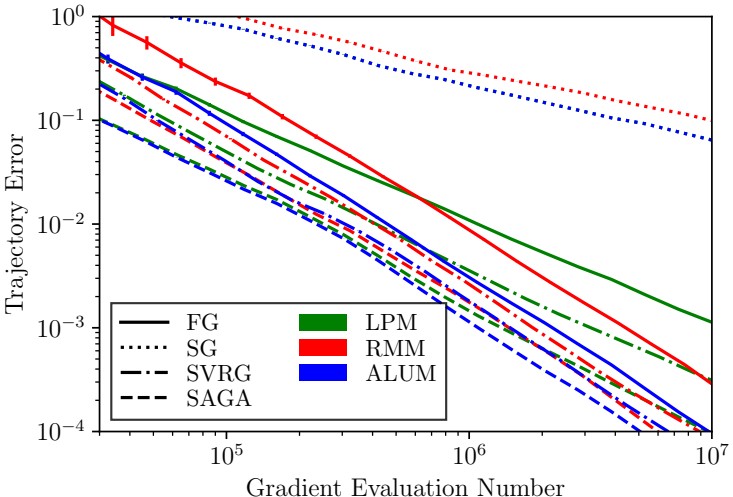

Figure 6: Logistic regression on german dataset.

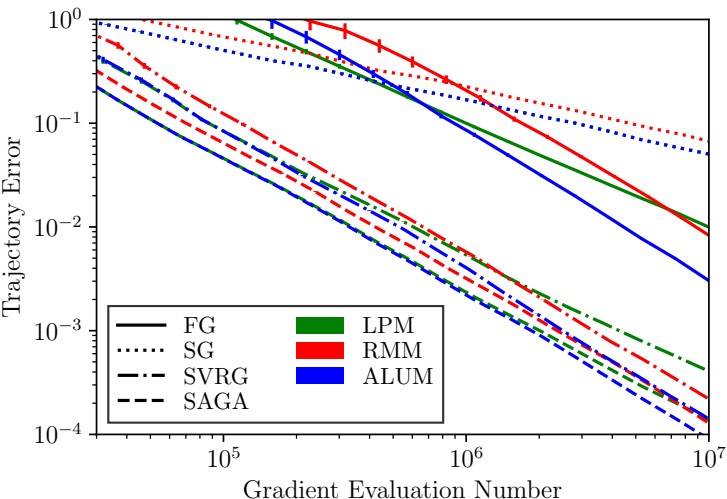

Figure 7: Logistic regression on mushrooms dataset.

## D.4 Dependency on step size

In contrast to Figure 1, we adopt a different point of view in this section by considering the relationship between accuracy and step size.

Figure 8 verifies our theoretical result that discretization error is $O(h^{\frac{3}{2}})$ for ALUM.

We can also see that SAGA-ALUM has higher discretization error than SVRG-ALUM when using same step size. However, practically, SAGA uses less gradient evaluations per iteration, therefore achieves better gradient efficiency.

We also notice that given the same step size, the SAGA-RMM and SVRG-RMM have smaller error than SAGA-ALUM and SVRG-ALUM respectively. However, ALUM uses only one gradient evaluation per iteration in contrast to RMM which takes two, therefore ALUM based algorithms achieves better gradient efficiency.

Finally, we note that our theory analysis is only valid for small enough step size. For example, Theorem 4 requires $h^3 \leq \frac{1}{2304c}b^3 mN^{-2}$ and $h \leq \frac{m}{22}$. For a given step size $h$, our theory might not be applicable if $N$ is too large or $b$ is too small. This happens on phishing and mushrooms datasets where sample size $N$ is much larger than australian and german datasets, but we use the same batch

size. Figures 8c and 8e shows that trajectory error's dependence of step size $h$ is roughly $O(h^{1/2})$ when step size is relatively large, and then becomes $O(h^{3/2})$ when step size keeps decreasing.

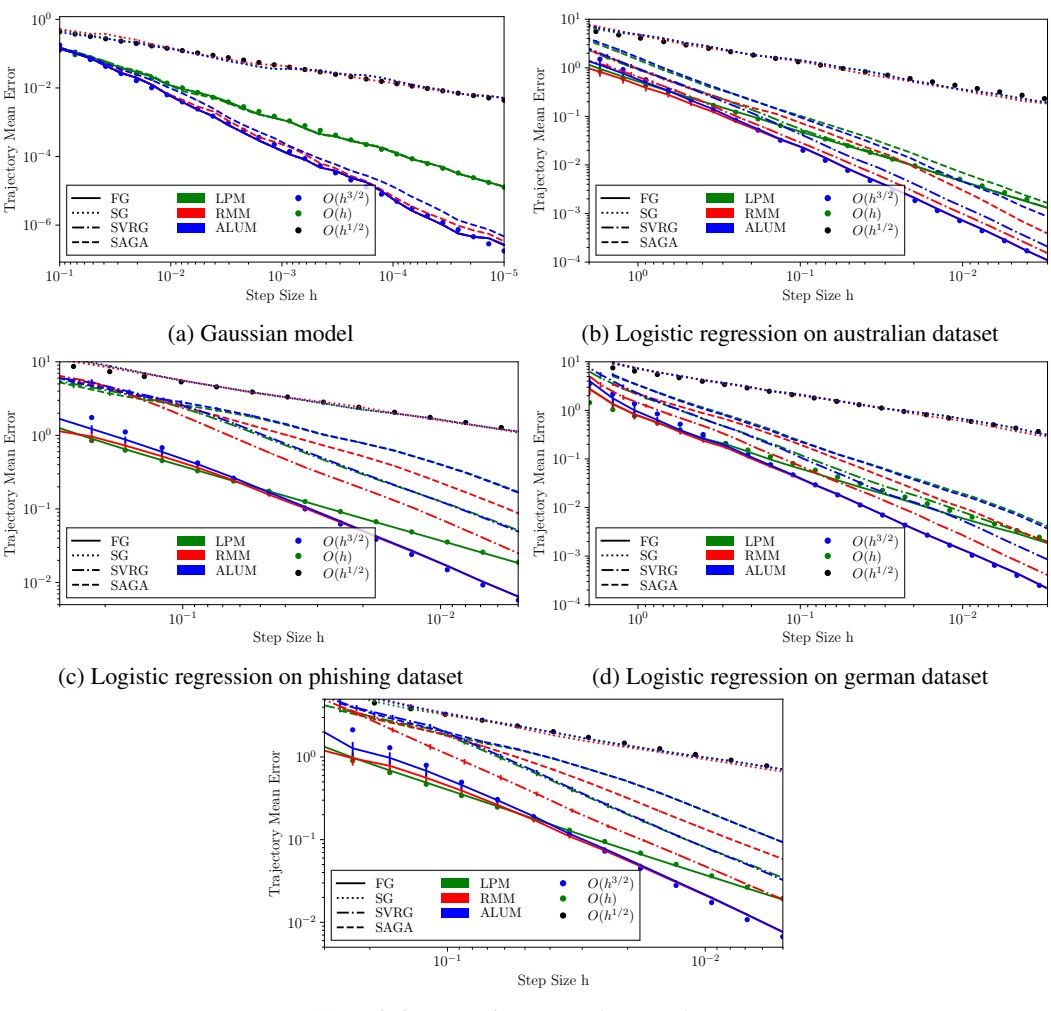

(a) Gaussian model        (b) Logistic regression on australian dataset

(c) Logistic regression on phishing dataset      (d) Logistic regression on german dataset

(e) Logistic regression on mushrooms dataset

Figure 8: Discretization error on australian dataset for different algorithms under different step size.[12]

## D.5 Dependency on batch size

Here, we give an intuition why our method is not sensitive to batch size when batch size is relatively small, and why the efficiency deteriorate for very large batch size.

We can roughly split the discretization error between the vector generated by the VR-ALUMs $z_k^{(o)\widetilde{\nabla}}$ and true solution $Z_{kh}$ as two part: (1) Difference between $z_k^{(o)\widetilde{\nabla}}$ and $z_k^{(o)}$ generated by the full gradient ALUM. We denote $E_1 = \|z_k^{(o)\widetilde{\nabla}} - z_k^{(o)}\|_{\mathbb{L}_2}^2$. (2) Difference between $z_k^{(o)}$ and $Z_{kh}$ as two part. We denote $E_2 = \|z_k^{(o)} - Z_{kh}\|_{\mathbb{L}_2}^2$.

According to analysis in Appendix B, we have $E_1 = O(h^3 \frac{N^2}{b^3})$ and $E_2 = O(h^3)$.

Next, we show that $E_1$ is irrelevant to $b$ if we fix the total number of gradient evaluation $n$. Since every iteration need $O(b)$ gradient evaluation, we can have at most $K = O(\frac{n}{b})$ iterations. Then the

---

[12]On phishing dataset and mushrooms datasets, SAGA-LPM, SVRG-LPM highly overlap with SAGA-ALUM and SVRG-ALUM respectively.

step size is $h = \frac{T}{K} = O(\frac{Tb}{n})$. Therefore, we can see $E_1 = \frac{T^3 N^2}{n^3}$ so this part of error doesn't change with respect to batch size.

On the other hand, $E_2 = h^3 = O(\frac{T^3 b^3}{n^3})$, therefore $E_2$ will increase as batch size increases.

When $b = O(N^{\frac{2}{3}})$, $E_1$ is larger than or at same order as $E_2$, and the change in $E_2$ doesn't affect the overall performance too much. Therefore, our methods are not sensitive to batch size when batch size is relatively small. When $b$ is very large, $E_2$ is dominant, and $E_2$ increases as batch size increases. Therefore, the overall efficiency deteriorates significantly for extremely large batch size.

Finally, we show Figure 9 which is larger and contains more information than Figure 3.

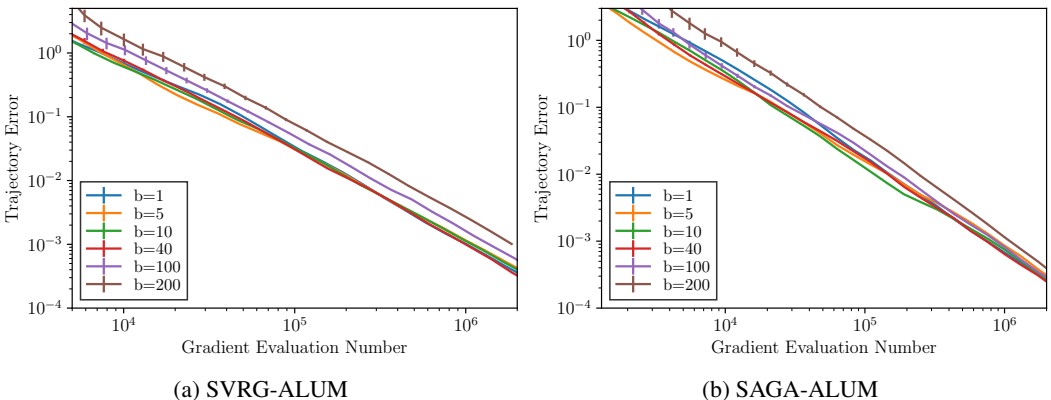

(a) SVRG-ALUM           (b) SAGA-ALUM

Figure 9: Discretization error for SVRG-ALUM and SAGA-ALUM with different batch sizes on australian dataset.

## D.6 ALUM for Sampling

In previous subsections, we discussed the accuracy of ALUM for approximating a ULD process and show that ALUM achieves smaller discretization error on estimating a ULD process than LPM and RMM with same number of gradient evaluation. When time $T$ is very large, the continuous ULD process converge to the target distribution. Therefore, any output of ALUM can be used as a sample approximately drawn from the target distribution.

In this section, we apply ALUM to sample from the posterior of a Bayesian logistic regression model. We use prior distribution as $p(\boldsymbol{x}) \sim \mathcal{N}(0, mI)$, likelihood as $p(y_i|\boldsymbol{a}_i, \boldsymbol{x}) = \frac{1}{1+\exp(-y_i \boldsymbol{a}_i^\top \boldsymbol{x})}$ for $y_i \in \{1, -1\}$ and $y_i, \boldsymbol{a}_i$ comes from training set. The posterior is just $p^*(\boldsymbol{x}) \propto \exp(\sum_{i=1}^N f_i(\boldsymbol{x}))$ where the potentials $f_i$ are defined as in Appendix D.2.

Within any gradient evaluation budgets, ALUM and VR-ALUMs can generate a sample $\boldsymbol{x}$ from a distribution that is similar to target distribution $p^*(\boldsymbol{x})$. We evaluate these sampling algorithms with the following tasks:

1. Estimating the mean potential: we sample $M = 100$ number of independent $\boldsymbol{x}_m$ by ALUM or VR-ALUMs, and report the mean potential $\frac{1}{M} \sum_{m=1}^M f(\boldsymbol{x}_m) = \frac{1}{M} \sum_{m=1}^M \sum_{i=1}^N f_i(\boldsymbol{x}_m)$.

2. Estimating the mean accuracy on test set: we sample $M = 100$ number of independent $\boldsymbol{x}_m$ by ALUM or VR-ALUMs, and report the mean accuracy $\frac{1}{M} \sum_{m=1}^M \frac{1}{N'} \sum_{i=1}^{N'} \mathbb{I}(y_i \boldsymbol{a}_i^\top \boldsymbol{x}_m > 0)$ where $N'$ is the size of test set, $y_i, \boldsymbol{a}_i$ comes from test set and $\mathbb{I}$ is the indicator function.

3. Estimating the posterior predictive distribution: we sample $M = 100$ number of independent $\boldsymbol{x}_m$ by ALUM or VR-ALUMs, and report the estimated posterior predictive distribution $\frac{1}{N'} \sum_{i=1}^{N'} \frac{1}{M} \sum_{m=1}^M \frac{1}{1+\exp(-y_i \boldsymbol{a}_i^\top \boldsymbol{x}_m)}$ where $N'$ is the size of test set and $y_i, \boldsymbol{a}_i$ comes from test set. This should approximate the true posterior predictive distribution $\frac{1}{N'} \sum_{i=1}^{N'} p(y_i|D_{\text{train}}) = \frac{1}{N'} \sum_{i=1}^{N'} \int p(y_i|\boldsymbol{a}_i, \boldsymbol{x}) p^*(\boldsymbol{x}) d\boldsymbol{x}$.

Although our theory in Theorems 1 and 3 control the sampling error in 2-Wasserstein distance, we don't conduct experiments to measure sampling error directly due to high computational cost.[13]

In order to apply ALUM for sampling, we first transform the potential to satisfy $L = 1$ as discussed in Remark 1, then we use hyperparameters $b = 1$ and $h = \frac{1}{40}$.

The results for the above tasks are shown in Figures 10 to 13, where the $x$-axis represent the number of gradient evaluations needed to generate a single sample. We can see that SAGA-ALUM and SVRG-ALUM converge much faster than full gradient version, since only a small batch is used for each iteration. SVRG-ALUM has multiple "plateau" in the plot, which represent the periodic update of full gradient.

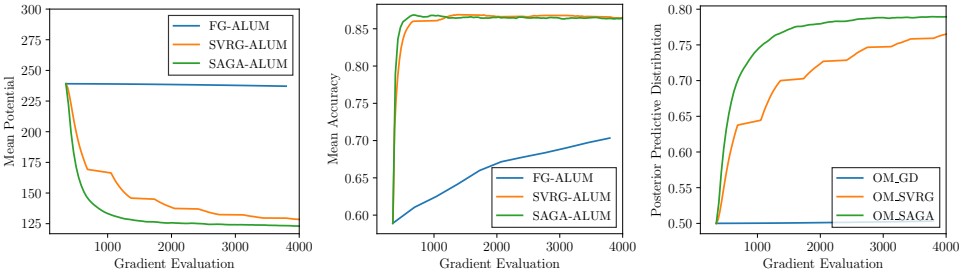

Figure 10: Sampling from posterior of Bayesian logistic regression model on australian dataset.

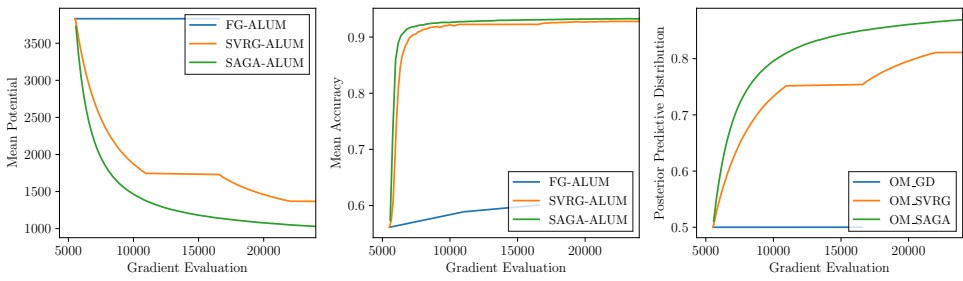

Figure 11: Sampling from posterior of Bayesian logistic regression model on phishing dataset.

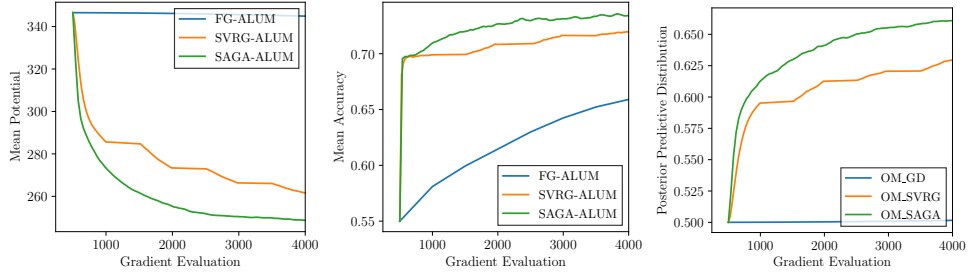

Figure 12: Sampling from posterior of Bayesian logistic regression model on german dataset.

---

[13]The 2-Wasserstein distance between these two continuous distributions cannot be calculated directly, therefore we must draw many samples from these distributions and calculate 2-Wasserstein distance between two discrete distribution as an approximation. The final stationary distribution of ALUM and VR-ALUMs could have very small sampling error (e.g. about $10^{-2}$). In order to experimentally measure this error, we must be able to approximate the 2-Wasserstein distance between two continuous distributions up to an accuracy which is at least one order of magnitude smaller than sampling error. Due to the curse of dimension, we need an exponentially large number of samples to do that. The cost of generating these samples and the cost of calculating the distance afterwards are unaffordable in high dimensional space.

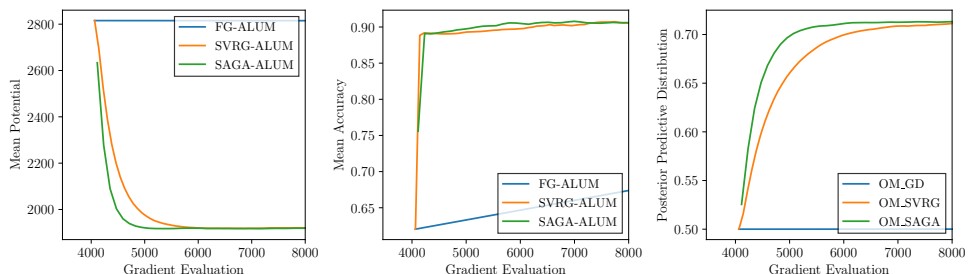

Figure 13: Sampling from posterior of Bayesian logistic regression model on mushrooms dataset.

We call the sampling error of ALUM and VR-ALUMs after running infinite steps with a fixed step size as flat sampling error. All results in this subsection illustrate how quickly the ALUM and VR-ALUMs converge with a fixed step size, but don't directly measure the flat sampling error. According to Section 5, we know both the discretization error and the flat sampling error have upper bound of $O(h^{3/2})$. We show in Appendix D.4 that this upper bound is tight for discretization error. However, it is still not clear whether the upper bound is tight for the flat sampling error.