# OpenReview forum: "Optimal Underdamped Langevin MCMC Method"
_NeurIPS.cc/2021/Conference — NeurIPS 2021 Poster_

### Official Review · Reviewer_S5uq · 2021-07-11

**Rating:** 7
**Confidence:** 2

**Summary:**

The paper makes new contributions to the literature on sampling from a strongly log-concave smooth measure. This is done by approximating the Underdamped Langevin Diffusion (ULD), and so, a fortiori, the paper also contributes to this approximation problem.

The main result is a new algorithm to approximate the ULD, which is well suited to a stochastic setting, where the gradient of the potential is replaced by certain estimators.
In the full gradient (i.e. without using an estimator for the gradient) setting the new algorithm matches the asymptotic performance of previous works while cutting the number of Gradient queries by about half.
In the stochastic setting, if the potential can be represented as a sum of $N$ smooth functions, then the authors also show how to modify their algorithm to accommodate estimators to the gradient and reduce the overall query and iteration complexity.

The results of the stochastic setting are also complemented by a lower bound which essentially show that the dependence on the parameters $N$, and $d$, the dimension are essentially optimal.


**Limitations And Societal Impact:**

The authors adequately addressed the limitations and potential negative societal impact of their work.

**Main Review:**

======POST REBUTTAL======
The authors addressed my concerns and I have raised my score accordingly.

The new algorithm, which the authors dub ALUM (with stochastic variant VR-ALUM), is based on the RMM method for discretizing the ULD, introduced by Shen and Lee in 2019.
The RMM method makes two gradient queries per iteration, and the gradient cleverly identified a way to remove one query so that the overall bias does not increase too much.
However, this does come at the cost of worsening the dependence on the condition number, which, I believe, was the main motivation in introducing the RMM algorithm.

The stochastic gradient analog introduced by the authors is a nice addition to the literature and, to my knowledge, outperforms previous algorithms. It is based on thoroughly analyzing variance reduction techniques to stochastic gradients coupled with the discretization scheme of the ULD. The supplied lower bounds further strengthen this.

While the authors supply a clear motivation and an explanation of their analysis, I tried to go over the proofs. I found them extremely technical, even when compared to previous works. This is the reason why my current score is not higher, and I am hoping that the authors can comment on that and explain the main technical difficulties and their own contributions in this sense so that I can raise my score.

Other than that, the paper has many typos and English grammar mistakes. I would ask the authors to go over the entire paper and fix such issues carefully. Such a thing could greatly improve the readability of the paper. (I want to emphasize that my score is not based on that point).

Other comments (I do not mention here typos):
Table 1: Actually, the dependency on $d$ is not shown below ($d$ is not even properly defined by this point)
Line 36: I agree that the question is natural, but perhaps there is some room for discussion about the notion of optimality here.
Optimality can be taken with respect to the number of iterations, the total number of queries, and total operations.
Also, which parameters are taken into account (this discussion appears later in the paper, but some small mention could be appropriate here.
Line 63: I could not understand which divergences are discussed. Total variation vs. something else?
Line 95: I'm not sure what the $h$ in the subscript $X_h$ stands for.
Line 126: I'm not sure what the comparison to Nesterov's momentum algorithm is supposed to demonstrate. Yes, the algorithms are similar because both of them add a momentum term to accelerate the vanilla gradient method. This similarity was already observed before (for example [CCBG2018]).
Line 211: I would suggest reconsidering the use of $U$ for a potential function and $\mathcal{F}$ for a function class. An uncareful reader can think that $\mathcal{F}$ stands for a sigma-algebra and $U$ is an open set.
Line 239: Can the authors say something about the requirement that n is a multiple of N. At first glance, it looks very artificial.
Line 727: I could not find a Lemma 7 in the paper of Dalalyan and Riou-Durand, in any of its versions.
Line 371: On the same topic, there are only two authors to this paper, no et al.

[CCBG2018] Cheng, X., Chatterji, N. S., Bartlett, P. L., & Jordan, M. I. (2018, July). Underdamped Langevin MCMC: A non-asymptotic analysis. In Conference on learning theory (pp. 300-323). PMLR.

**Time Spent Reviewing:**

8

---

> ### Author Response · Authors · 2021-08-10
> **Response to Reviewer S5uq**
>
> We thank the reviewer for the insightful comments. We answer the reviewer’s suggestions and questions as below.
>
> **Q1.** However, this does come at the cost of worsening the dependence on the condition number, which, I believe, was the main motivation in introducing the RMM algorithm.
>
> **A1:** We agree with the reviewer’s observation above. As compensation for related topics, we also note that the impact of a worse dependence on κ is limited due to following reasons.
>
> * under the high precision regime (when ε is small enough), the dependency of κ is same for both ALUM and RMM.
>
> We summarize different ULD methods as follows.
>
> | Method | Iteration complexity | maximum order dependency on $\kappa$ | high precision regime |
> | ----------- | ----------- | ----------- | ----------- |
> | LPM      | $O(\kappa^{3/2}/\varepsilon+\kappa^2)$ | $O(\kappa^2)$ | $O(\kappa^{3/2}/\varepsilon)$ |
> | RMM   | $O(\kappa/\varepsilon^{2/3}+\kappa^{7/6}/\varepsilon^{1/3})$ | $O(\kappa^{7/6})$ | $O(\kappa/\varepsilon^{2/3})$ |
> | ALUM   | $O(\kappa/\varepsilon^{2/3}+\kappa^2)$ | $O(\kappa^2)$ | $O(\kappa/\varepsilon^{2/3})$ |
>
> Under the high precision regime, only the first term in the iteration complexities is dominant, and our ALUM method has same asymptotic complexity as RMM.
>
> * Our experiment also shows that ALUM consistently outperforms RMM.
>
> **Q2.**  ...  extremely technical ... I am hoping that the authors can comment on that and explain the main technical difficulties and their own contributions
>
> **A2:** Our analysis for upper bound in appendix B is inherently technical as it addresses many degrees of complexity altogether.
> We tried our best to separate and abstract these complexities.
>
> First in order to handle variance reduction, we introduce the "Bounded MSE property" to abstract out the specific form of gradient estimator.
> This effectively covers full gradient, SVRG and SAGA.
>
> Next, we decompose the single step error between ALUM and continuous process to a series of error terms by considering a series of intermediate processes as stated in overview B.5.1. Then we control them term by term which occupies most spaces.
>
> Finally, we resolve the recurrent inequality to give the result for multiple steps.
>
> Given that our method is based on and is more complex than previous methods, it is natural that upper bound analysis is more technical.
>
> In our analysis for lower bound, we tend to make it more explicit to describe our novel construction of adversarial models clearly. This makes the proof in appendix C verbose.
>
> **Q3.**  typos and English grammar mistakes … Table 1 ... Line 36 ...
>
> **A3:** We thank the reviewer for the suggestions. All above issues will be fixed in final paper and we will amend the presentation to avoid ambiguity and make it clearer.
>
> **Q4.** Line 63: I could not understand which divergences are discussed. Total variation vs. something else?
>
> **A4:** We are deeply thankful for the opportunity of clarification. We will provide a more detailed discussion regarding the MALA and MH step as below and we will make it clearer in the final paper.
>
> On line 63, we cite the exponential convergence result of MALA under total variation (TV) distance.
> The purpose is to compare MALA with other sampling methods without MH step, which typically enjoy polynomial convergence, and suggest that function value oracle is more powerful.
>
> An extra degree of complexity involved in the argument above is that, most convergence results for sampling methods without MH step are based on distribution divergence other than TV, e.g. 2-Wasserstein distance.
> The purpose of line 65 is to remark that different divergences are not directly comparable.
> This complexity doesn't affect the intuition that MH step enables exponential convergence, since that different distribution divergences are typically linked with certain transportation inequality and are only polynomially dependent with each other.
>
> **Q5.**  Line 95: I'm not sure what the $h$ in the subscript $X_h$ stands for
>
> **A5:** We will clarify that $h$ is the step size.
>
> **Q6.**  Line 126 ... the comparison to Nesterov's momentum algorithm ...
>
> **A6:** Thanks for mentioning [CCBG2018]. Our comparison focuses on a different aspect as [CCBG2018].
>
> The ALUM and NAG are similar because:
> * Both of them add a momentum term: This is noticed by [CCBG2018].
> * Both of them use an aggressive gradient evaluation: More specifically, they take a big jump along current momentum direction to calculate the gradient instead of directly computing gradient at current iterate. We believe this observation is new.
>
> The aggressive gradient evaluation is the difference between vanilla momentum and Nesterov momentum.
> This is also the difference between LPM in [CCBG2018] and our ALUM.
>
> **Q7.**  Line 211 ... reconsidering the use of $U$ for a potential function and $\mathcal{F}$ for a function class.
>
> **A7:** We thank the reviewer for the suggestion. We will change $\mathcal{F}$ to $\mathcal{G}$.
>
> However we think it's common practice to use the letter $U$ to denote potential energy function, thus we would like to keep this notation.
>
> **Q8.**  Line 239 ...  it looks very artificial
>
> **A8:** We fully agree.
> This condition is indeed artificial to facilitate the construction of perturbation models.
> As shown in lines 855-862, we spread $2n$ small perturbations evenly to N components, such that each component could have $2n/N$ perturbations.
>
> It might be possible to generalize to $n$ that is not multiple of $N$.
> However, we think it is not necessary as it doesn't affect the final result in Corollary 2.
> For more details, note that Corollary 2 doesn’t assume that $n$ to be a multiple of $N$ because we apply a small trick when combining Theorems 5-6 in appendix C.5.
>
> **Q9.**  Line 727 ... Lemma 7 ...
>
> **A9:** We made a mistake in the citation.
> It should be Lemma 7 in "User-friendly guarantees for the Langevin Monte Carlo with inaccurate gradient" by Dalalyan and Karagulyan.

---

> > ### Comment · Reviewer_S5uq · 2021-08-23
> > **Response to Rebuttal**
> >
> > Thank you very much for the clarifications.
> >
> > The authors have managed to address most of my concerns.
> >
> > While I am still not convinced that minimizing the number of gradient queries is actually a desirable property by itself, the authors do show that their algorithm is applicable in a wide range of settings.
> >
> > Perhaps more importantly, after reading the author's explanation and reading the proof of the upper bound, I think that the mathematical ideas underlying the paper are interesting and are not trivial.
> > I will raise my score to reflect my new opinion.
> >
> > One comment:
> > - First in order to handle variance reduction, we introduce the "Bounded MSE property" to abstract out the specific form of gradient estimator. This effectively covers full gradient, SVRG and SAGA.
> > Next, we decompose the single step error between ALUM and continuous process to a series of error terms by considering a series of intermediate processes as stated in overview B.5.1. Then we control them term by term which occupies most spaces.
> > Finally, we resolve the recurrent inequality to give the result for multiple steps.
> > Given that our method is based on and is more complex than previous methods, it is natural that upper bound analysis is more technical.
> > In our analysis for lower bound, we tend to make it more explicit to describe our novel construction of adversarial models clearly. This makes the proof in appendix C verbose."
> >
> > Although short, I found this note to be quite illuminating. I encourage the authors to find a way to incorporate it in the paper.

---

### Official Review · Reviewer_BMbv · 2021-07-14

**Rating:** 5
**Confidence:** 4

**Summary:**

The paper proposes a new way of discretizing the underdamped Langevin dynamics, by modifying the Randomized midpoint method, proposed by Shen and Lee 2019, which reduces the number of gradient queries by a constant factor. In the case when the log potential of the target distribution is a sum of many terms, the paper produces a variance-reduced variant of their discretization method, which they show achieves the optimal query complexity for approximating the underdamped Langevin dynamics.

**Limitations And Societal Impact:**

Yes

**Main Review:**

My main criticism for the paper is that the problems investigated are not well motivated. The setting is overly specific to be interesting to a practitioner, so the main contribution of the paper is a technical one. However, although the analysis is impressive, there does not seem to be genuinely new proof ideas introduced, so it seems unlikely that the paper will generate much interest among theoreticians. The lower bound that the paper proves is also not the most interesting one: it proves a lower bound for the point-wise approximation error; however, in sampling it is much more natural to consider distributional approximation error.

**Time Spent Reviewing:**

2

---

> ### Author Response · Authors · 2021-08-10
> **Response to Reviewer BMbv**
>
> We thank the reviewer for the time spent and valuable comments.
>
> **Q1.**  The problems investigated are not well motivated
>
> **A1:** We believe the sampling problem is well motivated. Sampling has important applications in Bayesian inference, multi-arm bandit optimization, reinforcement learning, etc.
> Our pursuit of the optimal ULD MCMC method is also well motivated as sampling efficiency could be improved by a better MCMC method.
>
> We are willing to hear further justification from the reviewer on the above claim.
>
> **Q2.** The setting is overly specific to be interesting to a practitioner, so the main contribution of the paper is a technical one.
>
> **A2:** We believe that Bayesian logistic regression is indeed interesting to a practitioner.
>
> Our contribution is also practical, as our novel SAGA-ALUM method achieves at least 10 times smaller error than RMM with the same computation budget and on two relatively large datasets (mushroom:N=8124 and phishing: N=11055), we even achieve a 100 times smaller error.
>
> **Q3.**  However, although the analysis is impressive, there does not seem to be genuinely new proof ideas introduced
>
> **A3:** Our proof for the upper bound is based on the novel idea that dropping one gradient term from RMM doesn't increase bias too much.
>
> Our proof for lower bound is based on the novel construction of adversarial models.
>
> **Q4.**  It seems unlikely that the paper will generate much interest among theoreticians
>
> **A4:** Both our upper and lower bound is novel and important. Closing the gap makes our theoretical result even more interesting.
>
> **Q5.** The lower bound that the paper proves is also not the most interesting one: it proves a lower bound for the point-wise approximation error; however, in sampling it is much more natural to consider distributional approximation error.
>
> **A5:**  We agree that a lower bound for distributional approximation error is also interesting.
> However, currently there is no existing method to derive that lower bound under our setting.
> Therefore we turn to the approximation error for accessible results.
> Please note that the lower bound for approximation error is still novel and useful, as identified by the Reviewer VSLc.
>
> We hope now we explained our motivation, novelty, and significance of our contribution a little better, and we would deeply appreciate it if the reviewer could kindly re-evaluate the contribution of our work.

---

### Official Review · Reviewer_NwCb · 2021-07-17

**Rating:** 5
**Confidence:** 4

**Summary:**

This paper studies sampling from strongly convex potential in the form of finite summation of N smooth components. The authors show that using ALUM to simulate ULD needs $O(N+d^{1/3}N^2/3/\epsilon^{2/3})$ gradient evaluations.

**Limitations And Societal Impact:**

Some limitations are discussed in sec 7.

**Main Review:**

This paper proposes a new algorithm ALUM and combines it with SVRG and SAGA. The algorithms and theorems are clearly written. The claims are supported by theoretical analysis and experimental results.

However, this paper does not discuss some of the important works studying the same problem, e.g. [1,2].  The significance of this work is unclear given previous works [1,2]. [1] shows it is possible to achieve better $\kappa$ and $N$ dependence than ALUM using much simpler algorithms based on similar ideas. The only advantage of ALUM over [1] is that ALUM can improve eps dependence from $1/\epsilon$ to $1/\epsilon^{2/3}$, but [2] shows that with a simple MH step, it’s possible to achieve $\log 1/\epsilon$ and much better $\kappa$ and $N$ dependence at the same time. For this reason, it’s unclear why improving $1/\epsilon$ to $1/\epsilon^{2/3}$ is important.

Moreover, the algorithm ALUM seems very tailored to sampling from potential in the form of finite summation. It’s based on the idea of RMM, a previous sampler, but its performance on sampling general log-concave distributions is worse than that of RMM. As a result, the contribution of ALUM to understanding broader sampling problems is limited as well.

[1] Chatterji, Niladri, et al. "On the theory of variance reduction for stochastic gradient Monte Carlo." International Conference on Machine Learning. PMLR, 2018.

[2] Lee, Yin Tat, et al. "Structured Logconcave Sampling with a Restricted Gaussian Oracle." arXiv preprint arXiv:2010.03106 (2020).



**Time Spent Reviewing:**

4

---

> ### Author Response · Authors · 2021-08-10
> **Response to Reviewer NwCb**
>
> We thank the reviewer for the time spent and valuable comments as it points out two relevant papers.
>
> However, we note that [1,2] use different settings than our paper, therefore the result is not directly comparable.
> * [1] analyze SAGA-LD and SVRG-LD based on an extra Hessian Lipschitz assumption, which makes the potential function smoother and makes the sampling problem easier. Our method doesn't rely on this assumption but still achieves better dependency on ε.
> * [2] uses the MH step which requires the potential function value oracle. This is more informative than the gradient oracle as discussed in line 65.
> However, the MH step requires calculating the full function value at each step, which could be expensive on a large dataset.
> Our VR-ALUM method avoids that and significantly reduces the computation needed.
>
> Once this confusion is cleared up, we see that our setting depends on less assumptions than [1] and uses cheaper oracle than [2]. We propose novel methods and show that VR-ALUM achieves optimal dependency on d,N,ε under this setting.
>
> We hope now we explained our contribution and position in the literature a little better, and we would deeply appreciate it if the reviewer could kindly re-evaluate the contribution of our work.
>
> **Q.**  But its performance on sampling general log-concave distributions is worse than that of RMM
>
> **A:** We don't believe that is true. On the contrary, all our experiments show that ALUM outperforms RMM.
>
> We are willing to hear further justification from the reviewer on this claim.

---

> > ### Comment · Reviewer_NwCb · 2021-08-21
> > **Response to Authors**
> >
> > Thank you for your response. However, I don’t agree with your claims about [1,2]. First, not all bounds in [1] are based on an extra Hessian Lipschitz assumption. For example, Theorem 4.3 does not assume hessian lipschitzness. Second, I agree that [2] requires function value, but it does not require full function value at each step. It uses an approximate MH step.
> >
> > Compared to RMM, I agree that ALUM reduces the number of gradients needed from two to one at each step, but it’s not clear what the $\kappa$ dependence is. The factor 2 gain can be easily dominated by a worse $\kappa$ dependence. ALUM might be of interest to practitioners since the experiments in the paper show that ALUM outperforms RMM, but that needs more experiments on real datasets to verify. Based on the current version, I think this paper's main contribution is an improved $\epsilon$ dependence in the sum-decomposable setting when zeroth order oracle is unavailable. I am still not convinced that this contribution is significant enough. I think this paper will be much stronger if it can show a better $\kappa$ bound.

---

> > > ### Author Response · Authors · 2021-08-22
> > > **Response to Reviewer NwCb**
> > >
> > > We appreciate the reviewer for the time and thought-provoking discussion.
> > >
> > > For concerns of [1], this paper contains multiple algorithms under different settings. When the reviewer originally commented that:
> > > > “[1] shows it is possible to achieve better $\kappa$ and $N$ dependence than ALUM”
> > >
> > > We thought the reviewer was referring to Theorem 4.1, 4.2, because Theorem 4.3 clearly shows a worse dependence of $\kappa$.
> > >
> > > We clarify that, Theorem 4.1, 4.2 in [1] achieve better dependence than ALUM because they rely on extra Hessian Lipschitz assumptions. Theorem 4.3 in [1] uses a similar setting as ours, and their dependence of $\kappa$, $N$ and $\varepsilon$ is worse than our paper.
> > >
> > > Comparison between Theorem 4.3 in [1] and our result of VR-ALUM is not straightforward because of one difference between the setting of Theorem 4.3 in [1] and our paper when deriving the asymptotic results.
> > > [1] defines mixing time as the number of steps to achieve $W_2(p_k,p^\ast)\leq\varepsilon$ and assumes gradient Lipschitz constant $L$ ([1] actually uses letter $M$) and strongly convex constant $m$ both scale linearly with the number of samples N. Our paper defines Iteration complexity as the number of steps to achieve $W_2(p_k,p^\ast)\leq\sqrt{\frac{d}{m}}\varepsilon$.
> > > After adapting Theorem 4.3 [1] into our setting, we can derive a computational complexity (mixing time $\times$ gradient evaluations per step) as $\widetilde{O}(N+\varepsilon^{-3})$. This is worse than our computation complexity $\widetilde{O}(N+N^{2/3}\varepsilon^{-2/3})$. In order to show that our computational complexity is not worse than the result of Theorem 4.3 in [1], we only need to show that $\frac{7}{9} N+\frac{2}{9} \varepsilon^{-3} \geq N^{7/9} (\varepsilon^{-3})^{2/9} \geq N^{2/3} \varepsilon^{-2/3}$, where the first inequality is an AM-GM inequality.
> > > Please note that the last inequality directly relaxes the order of N, therefore the inequality is asymptotically not tight, i.e. our computational complexity is better than the result of Theorem 4.3 in [1].
> > >
> > > > “[2] requires function value, but it does not require full function value at each step.”
> > >
> > > Our paper simply doesn't use function value at all, therefore [2] uses a different setting.
> > > Our matching lower and upper bounds are established under our setting, and are not directly comparable with other works that use different oracles.
> > >
> > > > “it’s not clear what the $\kappa$ dependence is”
> > >
> > > The iteration complexity for sampling by ALUM is $\widetilde{O}(\text{max}(\kappa/\varepsilon^{2/3},\kappa^2))$ which is clearly shown in Table 3 in page 6.
> > > This obviously contains the $\kappa$ dependence.
> > > Please kindly check Table 2,3 and Section 7, lines 257-272 in our main paper for more discussion.
> > >
> > > > “The factor 2 gain can be easily dominated by a worse $\kappa$ dependence.”
> > >
> > > This doesn’t happen in any of our experiments. Instead, all our experiments show that ALUM outperforms RMM.
> > > Our theoretical analysis also justifies this phenomenon: the iteration complexity of ALUM is $\widetilde{O}(\text{max}(\kappa/\varepsilon^{2/3},\kappa^2))$, and under the high precision regime (when $\varepsilon$ is small enough), the κ dependence is $O(\kappa)$ which is the same as RMM.
> > >
> > > > “ALUM might be of interest to practitioners since the experiments in the paper show that ALUM outperforms RMM, but that needs more experiments on real datasets to verify”
> > >
> > > We will add additional experiments on real datasets to further verify the speedup of ALUM over RMM practically.
> > >
> > > > “I think this paper will be much stronger if it can show a better $\kappa$ bound.”
> > >
> > > We fully agree it is an important future extension. However, we note that it doesn’t change the fact that we propose VR-ALUM which is the first method that achieves optimal dependence on $d$, $N$ and $\varepsilon$.

---

> > > ### Author Response · Authors · 2021-08-22
> > > **Addition Experiments Verifying the Speedup of ALUM over RMM**
> > >
> > > Per reviewer’s request, we provide additional experiments to further verify that ALUM outperforms RMM.
> > >
> > > We extend our experiments to all binary classification problems in UCI datasets.
> > > The specific setups are exactly the same as what we used for generating Figure 1(b) in the paper.
> > > The only difference is that we only report a single point (ALUM and RMM with a fixed computational budget) instead of a curve (results for a range of computational budget) because of limitations of pure text comment.
> > >
> > > The results are shown as below. Note that with the same number of gradient evaluation, ALUM achieves smaller trajectory error than RMM on all datasets. This observation justifies the benefit of ALUM over RMM and shows that ALUM is indeed of interest to practitioners.
> > >
> > > | dataset | # gradient | ALUM | RMM |
> > > | ----------- | ----------- | ----------- | ----------- |
> > > | monks-2 | 8500 | 6.39e-1±5.78e-2 | 1.46e0±1.85e-1 |
> > > | spect | 4000 | 1.34e0±7.82e-2 | 3.15e0±2.76e-1 |
> > > | breast-cancer | 14300 | 2.10e0±1.09e-1 | 4.31e0±3.34e-1 |
> > > | breast-cancer-wisc | 35000 | 9.50e-1±1.58e-1 | 1.84e0±3.76e-1 |
> > > | parkinsons | 9800 | 6.69e-1±4.11e-2 | 1.41e0±1.35e-1 |
> > > | blood | 37400 | 7.57e-1±6.34e-2 | 1.53e0±1.76e-1 |
> > > | planning | 9100 | 1.66e0±9.91e-2 | 3.72e0±3.86e-1 |
> > > | echocardiogram | 6600 | 1.72e0±9.54e-2 | 3.53e0±2.55e-1 |
> > > | balloons | 800 | 1.38e0±9.17e-2 | 3.15e0±3.15e-1 |
> > > | statlog-german-credit | 50000 | 4.34e0±2.53e-1 | 8.76e0±7.94e-1 |
> > > | breast-cancer-wisc-prog | 9900 | 1.23e0±7.22e-2 | 2.58e0±2.18e-1 |
> > > | molec-biol-promoter | 5300 | 3.62e0±1.74e-1 | 7.39e0±5.98e-1 |
> > > | acute-nephritis | 6000 | 8.42e-1±8.83e-2 | 1.98e0±2.47e-1 |
> > > | cylinder-bands | 25600 | 2.98e0±1.61e-1 | 5.72e0±4.84e-1 |
> > > | oocytes_merluccius_nucleus_4d | 51100 | 7.15e-1±3.93e-2 | 1.57e0±1.57e-1 |
> > > | spectf | 4000 | 4.75e-1±3.68e-2 | 1.15e0±1.73e-1 |
> > > | monks-3 | 6100 | 4.92e-1±5.26e-2 | 1.24e0±1.69e-1 |
> > > | bank | 226100 | 3.77e0±2.52e-1 | 7.58e0±7.68e-1 |
> > > | conn-bench-sonar-mines-rocks | 10400 | 1.68e0±1.12e-1 | 3.67e0±3.60e-1 |
> > > | hepatitis | 7800 | 1.52e0±8.13e-2 | 2.89e0±2.42e-1 |
> > > | horse-colic | 15000 | 2.02e0±1.17e-1 | 4.31e0±4.06e-1 |
> > > | acute-inflammation | 6000 | 1.12e0±9.37e-2 | 2.30e0±2.77e-1 |
> > > | ringnorm | 370000 | 8.28e0±8.60e-1 | 1.63e1±2.23e0 |
> > > | pima | 38400 | 1.73e0±1.44e-1 | 3.58e0±4.33e-1 |
> > > | heart-hungarian | 14700 | 1.56e0±9.99e-2 | 3.21e0±3.06e-1 |
> > > | mushroom | 406200 | 5.63e0±8.49e-1 | 9.56e0±2.23e0 |
> > > | twonorm | 370000 | 5.67e0±8.37e-1 | 9.73e0±2.13e0 |
> > > | congressional-voting | 21800 | 1.32e0±7.35e-2 | 2.94e0±2.52e-1 |
> > > | fertility | 5000 | 2.25e0±1.14e-1 | 4.73e0±3.95e-1 |
> > > | vertebral-column-2clases | 15500 | 7.81e-1±7.45e-2 | 1.73e0±2.16e-1 |
> > > | credit-approval | 34500 | 1.98e0±1.81e-1 | 4.16e0±4.91e-1 |
> > > | statlog-australian-credit | 34500 | 2.29e0±1.28e-1 | 4.56e0±4.08e-1 |
> > > | statlog-heart | 13500 | 1.65e0±1.33e-1 | 3.23e0±3.51e-1 |
> > > | oocytes_trisopterus_nucleus_2f | 45600 | 1.54e0±9.55e-2 | 3.21e0±3.10e-1 |
> > > | musk-1 | 23800 | 1.34e0±8.41e-2 | 3.02e0±3.45e-1 |
> > > | monks-1 | 6200 | 5.42e-1±5.49e-2 | 1.28e0±1.77e-1 |
> > > | trains | 500 | 6.31e-1±4.14e-2 | 1.31e0±1.06e-1 |
> > > | hill-valley | 30300 | 4.59e-1±3.30e-2 | 1.01e0±1.29e-1 |
> > > | haberman-survival | 15300 | 1.38e0±9.33e-2 | 2.62e0±2.51e-1 |
> > > | pittsburg-bridges-T-OR-D | 5100 | 1.67e0±8.70e-2 | 3.41e0±3.14e-1 |
> > > | chess-krvkp | 159800 | 7.13e0±6.27e-1 | 1.25e1±2.04e0 |
> > > | magic | 951000 | 3.37e0±4.56e-1 | 5.82e0±1.44e0 |
> > > | tic-tac-toe | 47900 | 3.85e0±2.27e-1 | 7.97e0±7.36e-1 |
> > > | spambase | 230100 | 6.16e0±6.86e-1 | 1.06e1±1.89e0 |
> > > | ozone | 126800 | 1.28e0±7.60e-2 | 2.61e0±2.22e-1 |
> > > | adult | 1628100 | 9.34e0±1.06e0 | 1.89e1±2.73e0 |
> > > | ionosphere | 17600 | 1.55e0±1.04e-1 | 3.03e0±3.82e-1 |
> > > | mammographic | 48100 | 1.37e0±1.59e-1 | 2.61e0±3.79e-1 |
> > > | titanic | 110100 | 1.22e0±1.40e-1 | 2.35e0±3.80e-1 |
> > > | musk-2 | 329900 | 2.61e0±1.90e-1 | 5.42e0±6.63e-1 |
> > > | breast-cancer-wisc-diag | 28500 | 1.23e0±1.36e-1 | 2.20e0±3.64e-1 |
> > > | connect-4 | 3377900 | 8.63e0±6.08e-1 | 1.80e1±2.13e0 |
> > > | miniboone | 6503200 | 6.63e0±4.61e-1 | 1.28e1±1.59e0 |
> > > | ilpd-indian-liver | 29200 | 1.43e0±8.50e-2 | 3.12e0±3.05e-1 |

---

### Official Review · Reviewer_VSLc · 2021-07-17

**Rating:** 8
**Confidence:** 4

**Summary:**

This paper proposes a randomized algorithm for approximating a trajectory of underdamped Langevin dynamics, similar to the RMM method cited in the paper. It further uses variance reduction method to improve the error over sum-decomposable problems. An information theoretic lower bound is established, which the proposed method matches.

**Limitations And Societal Impact:**

Yes.

**Main Review:**

This paper is interesting and technically sound. Even tough the lower bounds are only in terms of approximating a specific trajectory---instead of for sampling a posterior distribution---they are still useful for numerical methods with Monte Carlo strategies. I quickly skimmed through the proofs, which seem to make sense.

More specific comments below:

In remark 1, the authors stated that the function can be rescaled for arbitrary L. It is worthwhile to point out how does the step size h scale with L as well.

The notation of equations (2) and (3) causes some confusion: I first thought that the authors are analyzing an algorithm with a shriking step size. It turns out that the step size is a constant. I would suggest chaning t to something else (perhaps \eta).

In variance reduction, m has been double used as both the epoch length and the strong convexity. For brevity, the authors can also choose a suitable (if not optimal) epoch length and define it in the algorithm.

It seems that the improvement of N dependence provided by the current method is due to the variance reduction method. It seems to also work on the RMM method. Is that the take away from the experiments? Also it is mentioned that the error for ALUM, LPM, and RMM with full gradient, stochastic gradient, SVRG, and SAGA are compared. I can't seem to find the curve for LPM with stochastic gradient.

The overall structure of the paper is clear. However, the literary quality of the paper can be improved. The authors should also be aware of grammatical errors. For example, in the abstract, "consists of finite summation" should be changed to "consisting of finite summation".

**Time Spent Reviewing:**

5

---

> ### Author Response · Authors · 2021-08-10
> **Response to Reviewer VSLc**
>
> We thank the reviewer for the insightful comments. We answer the reviewer’s suggestions and questions as below.
>
> > It is worthwhile to point out how does the step size h scale with L as well ... The notation of equations (2) and (3) causes some confusion ... m has been double used as both the epoch length and the strong convexity ... grammatical errors
>
> Thanks for the great suggestions. We will change as suggested or add explanations to avoid confusion and make it clearer.
> In remark 1, we will clarify that we only scale the position $x$, and don't scale time $t$ therefore the step size is still the same.
>
> > It seems that the improvement of N dependence provided by the current method is due to the variance reduction method. It seems to also work on the RMM method.
>
> We fully agree, as all our experiments justify the use of VR-RMM over RMM.
>
> However, currently there is no theoretical guarantee on the convergence of VR-RMM.
> We also didn't provide non-asymptotic theoretical analysis of VR-RMM because VR-RMM uses two gradient estimations in each iterate.
> The variance of previous gradient estimation could propagate to the bias of second gradient estimation. That complicates the analysis a lot.
>
> > Is that the take away from the experiments?
>
> We agree that "VR-RMM is better than RMM" is one take away from experiments.
> As compensation for the discussion, we believe the most significant experimental result is that SAGA-ALUM is better than any other algorithms considered in the experiments.
>
> > the curve for LPM with stochastic gradient.
>
> The plot of stochastic gradient LPM highly overlaps with stochastic gradient ALUM.

---

### Official Review · Reviewer_aKpK · 2021-07-26

**Rating:** 7
**Confidence:** 4

**Summary:**

The paper studies the underdamped Langevin diffusion for strongly-convex potential consists of finite sum of smooth components. The authors propose ALUM, which achieves optimal asymptotic complexity in the full gradient setting, and only requires one gradient at each iteration compared to RMM. For the stochastic gradient setting, VR-ALUM methods are proposed which improve over the previous methods in gradient complexity. The authors also accompany with a lower bound showing that the proposed methods are optimal in related parameters for the approximation problem.


**Limitations And Societal Impact:**

The authors have adequately addressed the limitations and potential negative societal impact of their work.

**Main Review:**

Overall, I believe this work is a nice complement to existing underdamped Langevin MCMC literature. The presentation is good with clear upper and lower bounds and fair experiments. My comments are summarized as below:

1. The paper studies two problems: sampling in $W_2$ distance and approximation in Euclidean distance. It is known that in the full gradient setting, the algorithm matches RMM, which achieves optimal asymptotic complexity for sampling problem, and in the stochastic setting, the proposed algorithm achieves optimal complexity for the approximation problem. Since sampling problem is considered an important target in MCMC, it would be great if the authors can further comment on the optimality of the proposed algorithm for sampling problem in the stochastic setting beyond the paragraph around line 252: is the gap conjectured to be only logarithmic? What are the lower bounds that one can achieve so far?

2. Given that there are two different problems (sampling vs. approximation) and two different settings (full gradient vs stochastic gradient), the authors are encouraged to be explicit on the optimality w.r.t. which problem in the introduction and be careful with overclaim in e.g. line 57. Furthermore, the rate $\Theta(N+d^{1/3}N^{2/3}/\epsilon^{2/3})$ for approximation never appears in the introduction, which to me is one of the core contributions of this paper.

3. The theoretical analysis relies on the assumption that the initialization is not too far away from the target distribution. How does this assumption compare to that in other algorithms? Is this the exact same assumption used in other algorithms? Appendix A.6 also appears to be too brief to make the initialization condition clear.


** Update on the reply:

Thanks for the detailed reply. I'm satisfied with the responses to the comments and remain my score.



**Time Spent Reviewing:**

4

---

> ### Author Response · Authors · 2021-08-10
> **Response to Reviewer aKpK**
>
> We thank the reviewer for the insightful comments. We will answer each point in the review as follows.
> 1. We fully agree that lower bounds for sampling problems are very important.
>
> However, we are not aware of lower bound or method to derive lower bound for sampling problem in our setting.
> Therefore we don't know whether the gap is just logarithmic.
>
> Before we proceed the discussion, we suggest distinguishing "sum-decomposable setting" and "stochastic gradient setting".
> Sum decomposable setting, which we use in the paper, gives a finite sum structure, such that the full gradient can still be calculated.
> However, stochastic gradient setting is harder in the sense that the precise gradient cannot be obtained with a finite number of evaluations.
>
> We note that there is a lower bound for stochastic gradient setting in "Oracle lower bounds for stochastic gradient sampling algorithms" by Chatterji et al..
> However, we don't know any lower bound for sampling problems under a sum-decomposable setting.
>
> 2. We thank the reviewer for the suggestions as it prompts us to clarify more in the final paper.
> The statement in line 57 is too abstract and needs to be incorporated with the previous sentence on line 55.
> We will append an explanation that VR-ALUM is optimal to ULD approximation problem and specific rate to make it clearer in the final paper.
>
> 3. We thank the reviewer for the opportunity of clarification.
>
> We first clarify that our non-asymptotic result doesn't use the assumption that the initialization is not too far away from the target distribution.
> Therefore our methods don't require a warm start, and non-asymptotic guarantees always hold regardless of what initialization we chose.
>
> For asymptotic results, this assumption simply prevents the initialization to be arbitrarily away from target distribution.
>
> Previous works use several different ways to prevent initialization error to be arbitrarily large.
> (Lee et al. 2019) starts from the global minimum.
> "Theoretical guarantees for approximate sampling from smooth and log-concave densities" by Dalalyan starts from a Gaussian distribution around the global minimum.
> (Cheng et al. 2018) assumes the distance between start point and global optimum to be smaller than a value $D$.
>
> We hope to separate our algorithm and specific initialization. Therefore, we introduce the form of assumption shown in the paper. Then we show that the assumption could be guaranteed by two initialization methods in appendix A.6.
> If the reviewer thinks the result in A.6 is not clear enough, we will add more derivations of 2-Wasserstein distance in A.6.

---

### Official Review · Reviewer_gwq5 · 2021-07-31

**Rating:** 7
**Confidence:** 3

**Summary:**

The paper considers methods for approximating underdamped Langevin diffusion processes with sum-decomposable strongly convex potential. The main contribution in the paper is the proposal of the Accelerated ULD-MCMC algorithm and its variance-reduced variants. This is followed by a detailed analysis of convergence for each of the algorithms, and the derivation of an information-based lower bound for gradient complexity in the task of estimating ULD that matches the upper bound (in dimensionality, component number, and target accuracy). The authors conclude with a discussion of the optimality (or lack thereof) of ALUM with respect to all the variables. Comprehensive experiments are presented to compare the performance of various algorithms on estimation ULD processes, showing that ALUM and its variants achieve better performance than previous algorithms.

**Main Review:**

** Update: The additional discussion and context provided in the rebuttal addressed my concerns and I have adjusted the score accordingly. **

The paper is interesting and well-written overall with good contribution to the algorithms for ULD processes. The authors offer a clear context about ULD and existing algorithms such as LPM and RMM (mostly in appendix, some of which I believe might better fit in the intro if space allows). I find it interesting and a bit surprising that dropping the $-\psi_2(at)\nabla f(X_0)$ term in RMM does not hinder the accuracy by much. The authors also offer a very comprehensive and detailed discussion on the experiments on both synthesized and real-world data.

However, my main concern is that I’m not fully convinced that the newly proposed algorithm and its performance improvement are significant enough. The difference between ALUM and RMM might seem subtle: I did not completely get the motivation behind ALUM or its intuition (e.g., why the extra gradient term can be drop - such intuition could potentially be of value in other problems); the factor-two reduction in gradient evaluation could also use more justification regarding why it is significant (e.g., is there some applications that can critically benefit from the factor-two reduction?). For the experiment results, while I agree that the proposed algorithms outperform the existing ones (e.g., Fig 1), I would again suggest that the authors argue its significance by, e.g., relating the improvement to some application.

Comparison of ALUM with RMM and NAG is also interesting, but I would hope to see deeper insights, e.g., whether there may be underlying intuition for why ALUM might be desirable in its current form. I would expect this theme to re-appear in later discussion if the authors believe there may be more interesting insights.

I did not go over all the proofs. Overall, the theoretical analysis is sound and I believe all the claims are correct. As a minor point I believe some of the proofs in the appendices could use more polishing with explanations around the equations (for easier reading).

To sum up, I believe the paper can be stronger if the authors could demonstrate better the significance of the improvement and/or offer some motivation or intuition behind the ALUM and RMM distinction.

### Other comments:

Should probably mention $I$ is identity matrix with dimension inferred from context (since $I_n$ is also used in some discussion for other things).

Given the tiny error bars (e.g., in Fig 1), I would expect the curves to be smoother/straighter. Is there certain artifact that contributes to the curviness or maybe I missed something in my thinking?

Apart from the main assumptions in Section 3 (Assumptions 1-3), there seem to be a number of sporadic assumptions in the theorems/lemmas, e.g. bounded 2-Wasserstein distance of initialization (Section 5, L156, and I did not fully get how Appendix A.6 addresses this), and assumption on step size $h$ in Theorem 3 (L185). I would hope that a quick comment can be added to address how strong/realistic these assumptions are.

The paper could benefit from proof-reading for typos and minor English style/grammar errors.

Some typos spotted:

L74: studies -> studied

L132: comes -> come, is -> are

L184: delete “and”

L521: Eqn (20), don’t think $\gamma=\bar{\gamma}$ is needed

L609: Eqn (35) seems incomplete (I think a summation is missing)

**Time Spent Reviewing:**

18

---

> ### Author Response · Authors · 2021-08-10
> **Response to Reviewer gwq5**
>
> We thank the reviewer for the time spent and valuable comments.
>
> We first emphasize that our contribution is indeed significant in both practice and theory.
> * In all experiments, our SAGA-ALUM achieves at least 10 times smaller error than RMM with the same computation budget. On two relatively large datasets (mushroom:N=8124, and phishing: N=11055), we even achieve a 100 times smaller error than RMM. We believe that an improvement of two orders of magnitude is enough to be called significant.
> * Moreover, our theory is also significant as it shows that our VR-ALUM is optimal in the way that any algorithm with better dependency of d,N,ε doesn't exist.
>
> As compensation for the discussion, we comment on some wrong ways to evaluate the significance of our contributions.
> 1. Only comparing full gradient ALUM and RMM.
>
> This is not sufficient to judge the significance of our contributions, as ALUM is just part of our contribution. The optimal result is actually achieved by VR-ALUM.
>
> We suspect that the reviewer didn't take VR-ALUM into account when saying that:
> > I’m  not fully convinced that the newly proposed algorithm and its performance improvement are significant enough. The difference between ALUM and RMM might seem subtle
>
> According to our experiment, the difference between VR-ALUM and RMM is indeed significant (e.g. two orders of magnitude improvement on two benchmark datasets).
>
> 2. Only comparing variance reduced versions: VR-ALUM and VR-RMM.
>
> VR-RMM was not proposed before, and there is no theoretical convergence guarantee for VR-RMM.
> Although we evaluate VR-RMM as a reference in experiments, VR-RMM should not be taken as an existing method for comparison.
>
> Next we comment on "deeper insight" that the reviewer asks for.
> > Comparison of ALUM with RMM ... whether there may be underlying intuition for why ALUM might be desirable in its current form.
>
> The construction of ALUM is rooted in the separated analysis of bias and variance.
>
> We find that RMM achieves much smaller bias than variance, and dropping certain terms only slightly increases bias, thus the bottleneck is still variance. However, dropping a gradient term could have practical interest such that only half gradient evaluations are needed. That gives rise to the final form of ALUM.
>
> We note the current form of ALUM could be changed while keeping the asymptotic convergence rate as discussed in lines 124-125.
>
> > I would expect this theme to re-appear in later discussion if the authors believe there may be more interesting insights.
>
> We do believe asymptotic analysis of bias and variance is important to understand our methods and inspire new methods.
> We will add more discussions on the insights in the final paper.
>
> > Comparison of ALUM with ... and NAG
>
> We find both ALUM and NAG use an aggressive gradient evaluation. More specifically, they take a big jump along current momentum direction to calculate the gradient instead of directly computing gradient at current iterate.
>
> We are not sure how important this aggressive gradient evaluation is and whether it is necessary to achieve optimal convergence rate.
>
> We also emphasize here that the construction and analysis of our ALUM algorithm is fully based on analysis of bias and variance but is not based on the aggressive gradient evaluation.
>
> Finally, we respond to specific comments or questions of the reviewer.
>
> >I’m not fully convinced that the newly proposed algorithm and its performance improvement are significant enough
>
> Please refer to our discussion at the beginning where we summarize the significance of our work.
>
> >why the extra gradient term can be drop
>
> This is based on the analysis of bias and variance.
> Please refer to our comment on "deeper insight" above or lines 114-125 for detail.
>
> >is there some applications that can critically benefit from the factor-two reduction?
>
> We first note that factor-two reduction for ALUM is only part of our contribution. VR-ALUM actually achieves more significant acceleration.
>
> Next, our experiments show that all Bayesian logistics regression tasks considered in our paper benefit from ALUM over RMM.
> More specifically, with the same computation budget, ALUM can achieve smaller errors than RMM. Equivalently, in order to achieve the same accuracy, ALUM uses less computation than RMM.
>
> >I would again suggest that the authors argue its significance by, e.g., relating the improvement to some application.
>
> We emphasize here that our SAGA-ALUM method achieves two orders of magnitude smaller error than RMM method in Bayesian logistics regression for two benchmark datasets.
>
> >some of the proofs in the appendices could use more polishing with explanations around the equations (for easier reading).
>
> We will improve the explanation of the proof.
>
> > mention  $I$  is identity matrix
>
> Thanks for your suggestion.
>
> > I would expect the curves to be smoother/straighter
>
> There are two possible reasons.
> * First, there is no theoretical guarantee that the plot should be straight.
> * Second, the samples within a chain typically are autocorrelated, which makes us underestimate the variance.
>
> > I would hope that a quick comment can be added to address how strong/realistic these assumptions are.
>
> * > bounded 2-Wasserstein distance of initialization
>
> We first note that this assumption is only introduced to derive asymptotic results. Therefore, our non-asymptotic analysis, e.g. Theorems 1-4 have nothing to do with this assumption.
>
> For asymptotic results, this assumption simply prevents the initialization to be arbitrarily away from target distribution. Practically, this could be guaranteed by multiple initialization methods. We give two of them in appendix A.6. More specifically, we derive upper bounds of 2-Wasserstein distance between initialization distribution and target distribution to show that these two initializations indeed satisfy the assumption.
>
> We emphasize again that our methods don't require a warm start, and non-asymptotic guarantee always holds regardless of what initialization we chose.
>
> * > assumption on step size  $h$  in Theorem 3
>
> This extra condition is easy to satisfy.
>
> For example, we can select $b=\Theta(N^{2/3})$, so that this condition becomes $h= O(m^{1/3})$. This is much weaker than another upper bound $h=O(m)$ firstly introduced in line 166. Therefore, by selecting $b=\Theta(N^{2/3})$, the maximum step size for VR-ALUM is at most constant times smaller than maximum step size for full gradient ALUM.
>
> > typos and minor English style/grammar errors
>
> Thanks. We will fix all typos and proofread the whole paper in the final version.
>
> Finally, we thank the reviewer for this detailed review and hopefully now we explained our significance of our contribution, novelty and intuition a little better.

---

### Official Review · Reviewer_5bz1 · 2021-08-02

**Rating:** 7
**Confidence:** 3

**Summary:**

In the context of sum-decomposable, smooth, and strongly convex potential functions the authors propose algorithms to improve the upper bound on the approximation error for estimating a Underdamped Langevin Diffusion (ULD) process and on the sampling error of the strongly-log-concave sampling problem. The continuous time ULD process has the strongly-log-concave distribution as its invariant distribution so approximating this process and sampling from this distribution are closely related. In the full gradient setting, the authors present a novel algorithm, ALUM which has optimal iteration complexity in $d$, $N$ and $\epsilon$ for the ULD approximation problem. Here, $N$ is the number of terms in the sum-decomposable function, $d$ is the input dimension and $\epsilon$ is the accuracy parameter. In comparison to the RMM algorithm (Lee et al), the dependence on $d$, $N$ and $\epsilon$ in the iteration complexity are matched, but the number of gradient evaluations are reduced by a factor of 2.
Next the stochastic gradient oracle setting is considered: here the authors propose algorithms (SVRG-ALUM and SAGA-ALUM) and analyze the corresponding sampling/discretization error. Finally an improvement in the lower bound for the gradient complexity of the ULD approximation problem to $\Omega(N + d^{1/3}N^{2/3}/\epsilon^{2/3})$ is presented. This shows that the gradient complexity of the presented Variance reduction modifications of ALUM match the lower bound in $d$, $N$ and $n$ in the sum-decomposable setting with a stochastic gradient oracle.

**Ethical Concerns:**

None to the best of my knowledge.

**Limitations And Societal Impact:**

Yes, I believe the authors have.

**Main Review:**

Overall the authors do a good job in conveying the content of the paper. The body of the paper abstracts the lower level details of the paper fairly effectively, and the relationship of the presented algorithms with the rest of the literature is comprehensive (case in point: section 4.1).

However, I should point out that I find the main contribution of the paper in the gradient oracle setting questionable. In each iteration, the algorithm saves a single gradient computation, but this is at the cost of worse dependence on $\kappa$ which can often be $10^2$ or larger in practice. Thus, overall the results seem to suggest that the gradient complexity may even be worse when the error parameter $\epsilon$ is moderately large.

- Space permitting, I would recommend moving the first paragraph from appendix A.1 into the preliminaries section of the paper. I think it sets up a nice general introduction to the setting and the relation between the sampling and ULD approximation problems.

- It should be made more clear that the gradient complexity upper bounds proved in the paper are not optimal in all parameters. The dependence on the condition number $\kappa$ can be improved (as with the RMM algorithm of Lee et al).

- In Section 7, the authors mention that the dependence on the condition number $O(\kappa^2)$ can be reduced to $O(\kappa^{3/2})$. It would help to be more explicit about the approach here and what are the challenges in improving the guarantee.

- The intuition in Section 4.1, lines 118-123 can be made more clear. Namely, it would be more clear to write the bias accumulated over $T/h$ steps as $O(Th^2)$ (instead of $O(h^2)$). Further, it is mentioned that the variance term is $O(h^{3})$, so I fail to see how the bias is of smaller order than the induced standard deviation. It would also help to point out how the calculation for the variance is carried out.

- According to the first two plots in the experiments section, the ALUM algorithm appears to have the same curve as the RMM algorithm but shifted by what appears to be a constant factor for all ranges of the error parameter. Seeing that the dependence on $\kappa$ ($\gg 1$ here) is presumably worse for ALUM compared to RMM in the proved results, does this plot seem to indicate that the analysis of with respect to $\kappa$ can be improved for ALUM? If the precision level $\epsilon$ is increased further, would the plots eventually diverge?

- An important point to address in the paper is the case when Assumption 2 is relaxed so that only the total potential function $f$ is L-Smooth and the uniform smoothness assumption on the $f_i$'s is dropped. Is there any intuition about whether ALUM / SVRG-ALUM continue to achieve a similar iteration complexity under the gradient and stochastic gradient oracles?

- This is an important point to address at least in the special case when $f$ takes the form of an empirical risk under L2 loss: the smoothness of each $f_i$ corresponds to $\| x_i \|_2^2$ where $x_i$ is the $i^{\text{th}}$ input point. The current smoothness assumption implies that all the $x_i$'s lie in an L2 ball of radius $\sqrt{L}$. Without the uniform smoothness assumption, but assuming that $f$ is $L$ smooth relaxes the condition on the points lying strictly in an L2 ball and imposes an eigenvalue constraint on the covariance matrix which is of more practical significance. Do the guarantees carry over here?


**Time Spent Reviewing:**

4

---

> ### Author Response · Authors · 2021-08-10
> **Response to Reviewer 5bz1**
>
> We deeply thank the reviewer for the time spent and valuable comments.
>
> **Q1.** Main contribution of the paper in the gradient oracle setting questionable ... saves a single gradient computation ... at the cost of worse dependence on κ
>
> **A1:** We thank the reviewer for the opportunity to clarify our contribution and comparison to the existing literature.
>
> First, we'd like to clear two confusions in the above statement.
> * The "gradient oracle" should be clarified into two cases: full gradient oracle and single component gradient oracle.
> We proposed two methods: ALUM and VR-ALUM to address both of them, where ALUM is the precursor of VR-ALUM.
> * Our main contribution is not only "saves a single gradient computation". Our novel method VR-ALUM achieves the best oracle complexity in terms of dependency of d,N, and ε.
> Our matching lower bound and upper bound are the most novel and salient contributions of this paper.
>
> Next, if we restrict the discussion to the full gradient case, we believe we have addressed most issues regarding dependency κ in section 7, lines 257-272. If the reviewer could propose a new question that we didn’t answer in the paper, we’d like to address it in the discussion.
>
> We emphasize here that the impact of a worse dependence on κ is limited since: 1. under the high precision regime, the dependency of κ is actually the same as SOTA. 2. In all experiments, our novel method ALUM performs better than previous SOTA full gradient method RMM, which shows that the benefits from ALUM outweigh the worse dependency on κ.
>
> **Q2.** Space permitting, I would recommend moving ... into the preliminaries ... it sets up a nice general introduction to the setting …
>
> Thank you for your suggestion. The only reason for us to move these contents into the appendix is the space limit. We are willing to move them back to main paper in the final paper.
>
> **A2:** It should be made more clear that the gradient complexity upper bounds proved in the paper are not optimal in all parameters.
>
> We fully agree that the word “optimal” should be handled carefully.
>
> As stated in the abstract, "our method is optimal in dependency of N , ε, and d". We also have a whole section devoted to explaining "in what sense our ALUM is optimal (or not)" which covers an extensive discussion of the dependency on κ.
>
> We thank the reviewer for prompting us to clarify more and we will make sure additional explanations are clearly stated at any occurrence of the word "optimal".
>
> **Q3.**... dependence on the condition number  $O(\kappa^2)$  can be reduced to $O(\kappa^{3/2})$. It would help to be more explicit…
>
> **A3:** We first provide intuition on how $O(\kappa^2)$ is derived.
> In each step, the approximation error $A_k$ decreases with a multiplier $e^{-\frac{m}{\gamma} h}$ and increases with noise $h^2 A_k$.
> In order to balance the decrease and increase in each step to ensure a decrease of error, we require $h= O(m)$. This upper bound makes the final iteration complexity $K=\Omega(\kappa^2)$.
>
> If we just need to balance same decrease and increase as $h^3 A_k$, the upper bound of step size becomes  $h=O(m^{1/2})$, and the final iteration complexity is $K= \Omega(\kappa^{3/2})$.
> We believe a different way to split bias and variance could achieve the above condition. However it complicates the analysis a lot, and has no effect on our main result, which is optimal dependency on d,N,ε. Therefore we only mention it without giving rigorous theoretical guarantees.
>
> **Q4.**... lines 118-123 can be made more clear …
>
> **A4:** Thank you for your suggestion.
> We will make it clearer in Section 4.1 by keeping value T.
>
> **Q5.** I fail to see how the bias is of smaller order than the induced standard deviation
>
> **A5:** We kindly clarify that we didn’t claim “bias is of smaller order than the induced standard deviation”. We are willing to hear from the reviewer which part of our writing needs to be changed to avoid the misunderstanding.
>
> For more details, according to discussion in lines 118-123, bias term is $O(h^2)$,  and square root of variance/standard deviation is $O(h^{3/2})$. Therefore the bias term has higher order dependency on step size $h$ than induced standard deviation.
>
> When h decreases to 0, bias decreases to 0 faster than square root of variance.
> Therefore, we say bias "has better dependency on $h$ than the square root of variance" in our paper.
>
> **Q6.** It would also help to point out how the calculation for the variance is carried out
>
> **A6:** The variance terms for both RMM and ALUM are $O(h^3)$.
>
> The variance of RMM is firstly derived in (Lee et al.). In our paper, an intuitive explanation is provided in appendix A.1 lines 502-505. A rigorous analysis is also written in appendix B.5.4
>
> Compared to RMM, ALUM only increases the variance by a very small value, so the final asymptotic rate is still $O(h^3)$. A rigorous analysis can be found in appendix B.5.5.
>
> **Q7.** Does this plot seem to indicate that the analysis of with respect to  κ  can be improved for ALUM?
>
> **A7:** All experiments show that our novel algorithm ALUM performs better than previous state of the art full gradient method RMM.
> We believe our plots demonstrate that the benefits from ALUM outweigh the worse dependency on κ.
> However, that doesn’t necessarily mean analysis can be improved.
>
> **Q8.** If the precision level  ϵ  is increased further, would the plots eventually diverge?
>
> **A8:** We first clarify several concepts.
> * We don't control error ε directly. Instead, we fix a computation budget (shown as x axis), then run algorithms to measure the error.
> *"Diverge" normally means that error becomes too large.
>
> Then we answer the reviewer’s question.
> If the error ε increases further (possibly achieved by reducing computation budget), then by definition, it diverges.
> If the precision increases, or equivalently the error decreases, then by definition, it doesn't diverge.
> Please note that the precision or error of algorithm is controlled by the computation budget.
>
> **Q9.**... Assumption 2 is relaxed so that only the total potential function $f$ is L-Smooth and the uniform smoothness assumption on the  $f_i$ 's is dropped.
>
> **A9:** This relaxation of the uniform smoothness assumption is interesting.
> However we’d like to point out that this assumption is too weak such that the gradient variance for any stochastic gradient method is unbounded.
> All methods can only use full gradient to achieve convergence guarantee.
> Therefore, the problem the reviewer proposed is essentially the same as it's full gradient version and has not much independent interest.
>
> **Q10.** any intuition about whether ALUM / SVRG-ALUM continue to achieve a similar iteration complexity under the gradient and stochastic gradient oracles?
>
> **A10:** ALUM utilizes full gradient and is not affected by uniform smoothness assumption on $f_i$. Therefore, all results for ALUM still hold.
>
> The gradient variance of SVRG is not bounded anymore without uniform smoothness assumption, thus SVRG-ALUM cannot be used under new setting.
>
> **Q11.**... in the special case ... an empirical risk under L2 loss ... Without the uniform smoothness assumption … Do the guarantees carry over here?
>
> **A11:** As explained above, ALUM can still be used.
>
> We hope we have explained more clearly the significance of our work. We also hope that the reviewer kindly re-evaluate how the dependency on κ affects the significance of our methods and matching upper bound and lower bound.

---

> > ### Comment · Reviewer_5bz1 · 2021-08-20
> > **Response to authors**
> >
> > I should clarify, I mean "full gradient oracle" and not the stochastic oracle gradient setting in my first comment. While the contribution in the stochastic gradient setting is interesting and novel, I am not convinced by how the result for ALUM improves on the existing state SOTA in the full gradient setting, RMM, for which guarantees apply even beyond the sum-decomposable setting. Reading the introduction of the paper and the response to Reviewer gwq5, it seems that the algorithm may have practical benefits? However I do not see this in the plots explicitly compared to RMM. There is an improvement of a constant factor in the achieved accuracy, which can be attributed to the number of gradient calls being halved (compared to RMM). Is this that significant compared to the performance of VR-ALUM. Mentioning the algorithm in this much detail dilutes the most important 2 contributions of the paper - the tight lower and upper bound on # of gradient evaluations in the sum-decomposable setting.
> >
> > I think the discussion on $\kappa$ is sufficient in the paper as clarified in the response, but I think the discussion surrounding the improvement of $\kappa$ from $\kappa^2$ to $\kappa^{3/2}$ can be amended or removed. The current writing in the paper does not make any explicit claims about how the improvement is possible, and as the authors claim is not the main focus of the paper. I would suggest keeping the statement only if an abridged version of the above argument can be included in the paper.
> >
> > I am satisfied with the remaining responses of the authors.
> >
> > I will amend my score for the paper if (1) any misconception I have about the contribution of ALUM is clarified, or (2) if none exists, the authors will consider reorganizing the paper so that the important contributions appear sooner in the paper (and using ALUM in the appendix as a vessel to motivate the actual optimal algorithm VR-ALUM).

---

> > > ### Author Response · Authors · 2021-08-20
> > > **Response to Reviewer 5bz1**
> > >
> > > We appreciate the reviewer for the time spent and valuable comments. Glad to hear the acknowledgement of the interest and novelty of our algorithm for stochastic gradient setting. We will answer specific comments or questions of the reviewer as follows.
> > >
> > > > how the result for ALUM improves on the existing state SOTA in the full gradient setting, RMM
> > >
> > > As we discussed in the introduction of the main paper, lines 25-28, lines 44-47, "RMM already has optimal asymptotic complexity in full gradient setting", therefore ALUM can only achieve constant speedup. The constant speedup is achieved by saving almost half of the gradient evaluation, as discussed in lines 101-125, section 4.1, where we proposed ALUM algorithm.
> > >
> > > > RMM, for which guarantees apply even beyond the sum-decomposable setting
> > >
> > > We are not aware of any work that gives guarantee for RMM in any setting other than full gradient setting. We don't know what is "beyond the sum-decomposable setting" referred to by the reviewer. We are willing to know the setting or work referred to by the reviewer.
> > >
> > > > Reading the introduction of the paper and the response to Reviewer gwq5, it seems that the algorithm may have practical benefits?
> > >
> > > In introduction, lines 44-47, we clarify that ALUM could achieve constant speedup compared to RMM.
> > > In response to reviewer gwq5, we mainly discuss how VR-ALUM achieves significant acceleration.
> > >
> > > > it seems that the algorithm may have practical benefits? However I do not see this in the plots explicitly compared to RMM.
> > >
> > > All experiments show that ALUM has practical benefits over RMM.
> > >
> > > From figure 1, we can see that ALUM achieves smaller errors than RMM with the same number of gradient evaluations.
> > >
> > > From figure 2, we can see that ALUM achieves almost the same error as RMM with the same step size. Note that with the same step size, ALUM only uses half gradient evaluations than RMM.
> > >
> > > From all figures in the appendix, we can see similar results.
> > >
> > > We are willing to hear why the reviewer got the above misunderstanding and how we should amend the main paper to avoid the misunderstanding.
> > >
> > > > Is this that significant compared to the performance of VR-ALUM.
> > >
> > > No. ALUM is only part of our contribution. We believe that our most significant contribution is algorithm VR-ALUM and the matching lower and upper bound.
> > >
> > > > I think the discussion surrounding the improvement of $\kappa$ from $\kappa^2$ to $\kappa^{3/2}$ can be amended or removed ... I would suggest keeping the statement only if an abridged version of the above argument can be included in the paper.
> > >
> > > We will clarify that this is not an explicit claim and provide additional explanation.
> > >
> > > > reorganizing the paper so that the important contributions appear sooner in the paper.
> > >
> > > We will append explanations in introduction to emphasize the significance of VR-ALUM and matching lower and upper bound and we thank the reviewer for prompting us to clarify more in the final paper.
> > >
> > > > using ALUM in the appendix as a vessel to motivate the actual optimal algorithm VR-ALUM
> > >
> > > We thank the reviewer for the suggestion to improve the presentation.
> > > However we would like to keep the discussion about ALUM in the main paper because:
> > >
> > > * One of our novel ideas in this paper is that dropping certain terms doesn't affect the accuracy too much. This is rooted in the separated analysis of bias and variance and using ALUM as an example could help to better demonstrate how the bias and variance term is analyzed.
> > > * By introducing ALUM as a precursor of VR-ALUM, readers could better understand how the algorithm evolves from existing method RMM to a better one.
> > > * In the experiment, we compare ALUM and RMM, therefore we need to define ALUM before that.
> > > * Although ALUM only achieves constant speedup compared to RMM, it still could be interesting for practitioners.
> > >
> > > We hope now we clarified the misconception about ALUM and explained the connection of it with other parts of the paper a little better.

---

> > > > ### Comment · Reviewer_5bz1 · 2021-08-21
> > > > **Response to the authors**
> > > >
> > > > One point: the RMM algorithm does not require the potential function $f$ to be sum-decomposable. It applies even beyond the sum decomposable setting. Whereas ALUM applies only in the sum-decomposable setting.
> > > >
> > > > Anyway, to summarize my review, I think the writing of the paper needs some work in order to properly bring out the contributions of the paper. There is too much of a focus on ALUM in the current write-up when it only brings in a constant factor improvement in the number of gradient evaluations (which is anyways known to be optimal). I think it should not be listed as a significant contribution of the paper, and instead, as a means to motivate the optimal algorithm VR-ALUM which has far superior performance. I am not convinced by the discussion in the paper as well the discussion above that this algorithm is more useful to practitioners compared to RMM, in contrast with the significant improvement provided by VR-ALUM.
> > > >
> > > > As a reader, the contributions of the paper would be less confusing instead of spending so much effort discussing ALUM, if there was a single remark/section showing that (i) the extra gradient step in RMM can be removed, and (ii) why can it be removed, leading into the next section introducing the optimal algorithm VR-ALUM. I think this would make the paper significantly more concise to read and appreciate. If the authors agree to introduce modifications to this effect, I will amend my score for the paper. Otherwise, my score is unchanged.

---

> > > > > ### Author Response · Authors · 2021-08-21
> > > > > **Response to Reviewer 5bz1**
> > > > >
> > > > > We appreciate the reviewer for the time and thought-provoking discussion.
> > > > > We first hope to clarify the following misconception about ALUM:
> > > > >
> > > > > > Whereas ALUM applies only in the sum-decomposable setting
> > > > >
> > > > > Both ALUM and RMM use full gradients. ALUM is not limited to the sum-decomposable setting, just like RMM.
> > > > >
> > > > > Once the above misconception is resolved, we could address the following concern.
> > > > >
> > > > > > I am not convinced by the discussion in the paper as well the discussion above that this algorithm is more useful to practitioners compared to RMM, in contrast with the significant improvement provided by VR-ALUM.
> > > > >
> > > > > We fully agree that VR-ALUM is a better choice than ALUM under the sum-decomposable setting.
> > > > >
> > > > > However, there are circumstances where only full gradient is accessible or full gradient is necessary to achieve the convergence guarantee. The reviewer actually has mentioned one example in the original review, please kindly check Q9,Q10 in the above discussion.
> > > > > For these problems, the practitioners could still benefit from ALUM even if VR-ALUM is not applicable.
> > > > > ---------------
> > > > >
> > > > > We thank the reviewer for continuously motivating us to improve our presentation.
> > > > >
> > > > > > spending so much effort discussing ALUM
> > > > >
> > > > > Per the reviewer's suggestion, we will shorten the length of discussion about ALUM and move part of the discussion to appendix so that it doesn’t dilute other important contributions. We will only leave necessary discussion of ALUM in the main paper to help non-expert readers understand the article more smoothly.
> > > > >
> > > > > > if there was a single remark/section showing that (i) the extra gradient step in RMM can be removed, and (ii) why can it be removed, leading into the next section introducing the optimal algorithm VR-ALUM. I think this would make the paper significantly more concise to read and appreciate
> > > > >
> > > > > There is already a section (section 4.1) in our paper to explain the above two points mentioned by the reviewer.
> > > > > Per the reviewer's suggestion, we will amend this section to make it clearer.
> > > > > We will also ensure all details on the construction of ALUM happens only in this section so that it doesn’t dilute other important contributions.
> > > > >
> > > > > >  If the authors agree to introduce modifications to this effect, I will amend my score for the paper. Otherwise, my score is unchanged.
> > > > >
> > > > > We will revise the paper as the reviewer’s  suggestion and we thank the reviewer for the specific and thought-provoking suggestions again. If the reviewer thinks more modifications are needed in the final paper, we are willing to listen.

---

> > > > > > ### Comment · Reviewer_5bz1 · 2021-08-24
> > > > > > **Response to the authors**
> > > > > >
> > > > > > Thank you for the response. I am satisfied with the discussion and have updated my score to reflect the same.

---

### Decision · Program_Chairs · 2021-09-27

**Decision:**

Accept (Poster)

**Comment:**

This paper proposes a randomized algorithm for approximating a trajectory of underdamped Langevin dynamics. It also uses variance reduction to improve the error over decomposable problems. It also establishes an information-theoretic lower bound. There have been extensive discussions between the authors and the reviewers, and I have gone through them carefully. I agree with the majority of the reviewers that the paper has made interesting technical contributions to the field. I am happy to recommend acceptance.